**RESEARCH**

# Target residence of Cas9-sgRNA influences DNA double-strand break repair pathway choices in CRISPR/Cas9 genome editing

Si-Cheng Liu[1,2†], Yi-Li Feng[1,2,3†], Xiu-Na Sun[1,2], Ruo-Dan Chen[1,2,3], Qian Liu[1,2], Jing-Jing Xiao[1,2], Jin-Na Zhang[2,4], Zhi-Cheng Huang[1,2], Ji-Feng Xiang[1,2,5], Guo-Qiao Chen[1,2], Yi Yang[2], Chao Lou[6], Hao-Dan Li[6], Zhen Cai[4], Shi-Ming Xu[2], Hui Lin[1] and An-Yong Xie[1,2*]

†Si-Cheng Liu and Yi-Li Feng contributed equally to this work.

*Correspondence:
anyongxie@zju.edu.cn

[2] Institute of Translational Medicine, Zhejiang University School of Medicine and Zhejiang University Cancer Center, Hangzhou, Zhejiang 310029, People's Republic of China
Full list of author information is available at the end of the article

## Abstract

**Background:** Due to post-cleavage residence of the Cas9-sgRNA complex at its target, Cas9-induced DNA double-strand breaks (DSBs) have to be exposed to engage DSB repair pathways. Target interaction of Cas9-sgRNA determines its target binding affinity and modulates its post-cleavage target residence duration and exposure of Cas9-induced DSBs. This exposure, via different mechanisms, may initiate variable DNA damage responses, influencing DSB repair pathway choices and contributing to mutational heterogeneity in genome editing. However, this regulation of DSB repair pathway choices is poorly understood.

**Results:** In repair of Cas9-induced DSBs, repair pathway choices vary widely at different target sites and classical nonhomologous end joining (c-NHEJ) is not even engaged at some sites. In mouse embryonic stem cells, weakening the target interaction of Cas9-sgRNA promotes bias towards c-NHEJ and increases target dissociation and reduces target residence of Cas9-sgRNAs in vitro. As an important strategy for enhancing homology-directed repair, inactivation of c-NHEJ aggravates off-target activities of Cas9-sgRNA due to its weak interaction with off-target sites. By dislodging Cas9-sgRNA from its cleaved targets, DNA replication alters DSB end configurations and suppresses c-NHEJ in favor of other repair pathways, whereas transcription has little effect on c-NHEJ engagement. Dissociation of Cas9-sgRNA from its cleaved target by DNA replication may generate three-ended DSBs, resulting in palindromic fusion of sister chromatids, a potential source for CRISPR/Cas9-induced on-target chromosomal rearrangements.

**Conclusions:** Target residence of Cas9-sgRNA modulates DSB repair pathway choices likely through varying dissociation of Cas9-sgRNA from cleaved DNA, thus widening on-target and off-target mutational spectra in CRISPR/Cas9 genome editing.

**Keywords:** CRISPR-Cas9 genome editing, DNA double-strand break repair pathway choice, Editing heterogeneity, Off-target effect, Target residence

## Introduction

CRISPR/Cas9 genome editing relies on the binding of the Cas9 nuclease, in complex with a single-guide RNA (sgRNA), to a DNA target to induce a site-specific DNA double-strand break (DSB) and its subsequent repair [1, 2]. Upon DSB induction by Cas9, different repair pathways compete for DSB repair, generating the desired DNA edits including substitutions, insertions, deletions, or translocations among varieties of repair products [3]. The two major DSB repair mechanisms in mammalian cells include non-homologous end joining (NHEJ) and homology-directed repair (HDR). While classical NHEJ (c-NHEJ) is the primary NHEJ pathway, alternative end joining (a-EJ) could also be employed to re-ligate the ends of DSBs if either of the core NHEJ factors including DNA-PKcs, Ku70/Ku80, and XRCC4/DNA ligase 4 is deficient or not engaged [4, 5]. If the ends of DSBs are readily ligatable, such as Cas9-induced blunt ends and I-SceI-induced 3′-overhanging ends, c-NHEJ generates largely accurate end-joining products whereas a-EJ remains mostly mutagenic [6–9]. Additionally, using homologous sequences as a template, HDR is the preferred pathway for accurate substitutions and insertions in CRISPR/Cas9 genome editing.

The DSB repair pathway choice is governed by a host of factors, including cell cycle stage, DNA end configurations, surrounding chromatin context, and local DNA metabolism [10]. Uniqueness in DSB induction by CRISPR/Cas9 may also participate in this regulation [11, 12]. In CRISPR/Cas9 genome editing, targeting Cas9 to a given site is mediated by several interactions, including the contacts between Cas9 and the proto-spacer adjacent motif (PAM) of the target, the base pairing of the sgRNA spacer with target strand and non-specific interactions between Cas9 and target DNA [2]. In vitro and in vivo studies have indicated that these interactions entail strong and persistent binding of the Cas9-sgRNA complex to its target and help maintain its target residence for hours (h) even after Cas9-induced DNA cleavage [13–17]. Repair kinetics reveals that repair of Cas9-induced DSBs is generally slow and often lasts for more than 20 h in mammalian cells; this is likely due to the concealing of DSBs by the Cas9-sgRNA complex retained at the cleaved DNA [17, 18]. Owing to intrinsic disparity in the interactions that mediate the binding of Cas9-sgRNA to its target, the binding affinity of Cas9-sgRNA varies at different sites along with altered target residence. It is likely that Cas9-sgRNA could be spontaneously released from its target or may encounter local DNA replication, transcription, or chromatin remodeling, leading to release of Cas9-sgRNA from cleaved DNA and exposure of Cas9-induced DSBs [11, 12, 19–22]. These DSBs are subsequently recognized and engaged with repair factors that determine a pathway choice. Therefore, Cas9-sgRNA target residence may regulate DSB repair pathway choices in CRISPR/Cas9 genome editing, as this residence can persist even after DNA cleavage. However, this hypothesis has yet to be tested. Even if target residence of Cas9-sgRNA affects repair of Cas9-induced DSBs, it is unclear what effect it has on repair of Cas9-induced DSBs and how.

Here, we find that the extent of c-NHEJ involvement varies between different target sites in repair of Cas9-induced DSBs in a population of asynchronous mammalian cells. We demonstrate that weakening target interaction of Cas9-sgRNA promotes the repair bias towards c-NHEJ at the same Cas9-induced DSBs. In vitro binding assays reveal that the weakened target interaction increases target dissociation and shortens target

residence of Cas9-sgRNA at cleaved and uncleaved DNA. The c-NHEJ inhibition, which is often used to increase HDR-mediated CRISPR/Cas9 genome editing, elevates off-target effects of CRISPR/Cas9, as the interaction between Cas9-sgRNA and off-target sites is weaker. Local DNA replication, not transcription, suppresses c-NHEJ and promotes a-EJ and HDR by dislodging Cas9-sgRNA that remains bound to its cleaved target and generating three-ended DSBs unsuitable for c-NHEJ. Repair of three-ended DSBs could result in palindromic fusion of sister chromatids, a key step in chromosomal breakage-fusion-bridge cycles and a potential source for on-target gross chromosomal rearrangements in CRISPR/Cas9 genome editing. As CRISPR/Cas9 genome editing generates highly heterogeneous repair products, the effects of Cas9-sgRNA target residence on DSB repair pathway choices at both on-target sites and off-target sites may significantly contribute to this mutational heterogeneity.

## Results

### Inactivation of c-NHEJ induces varying stimulation of Cas9-induced HDR among targets

Like any other DSBs, Cas9-induced DSBs are repaired by c-NHEJ, a-EJ, and HDR (Fig 1a). Thus, inactivation of the predominant NHEJ pathway c-NHEJ is expected to channel more Cas9-induced DSBs towards HDR for repair, increasing the usage of HDR [23–25] (Fig. 1a and Additional file 1: Fig. S1a). If target interaction of Cas9-sgRNA influences DSB repair pathway choice after DNA cleavage at its targets, the involvement of c-NHEJ in repair of Cas9-induced DSBs would change between targets with different target interaction for Cas9-sgRNA. Inactivation of c-NHEJ would thus lead to varying degrees of HDR stimulation at these sites. To test this hypothesis, we used *Streptococcus pyogenes* Cas9 (SpCas9) in complex with its sgRNA partner (Cas9-sgRNA) to induce site-specific DSBs at different sites in a single-copy HDR reporter integrated at the *Rosa26* locus in the genome of mESC and analyzed the impact of c-NHEJ inactivation on Cas9-induced HDR (Fig. 1a). This HDR reporter contains two inactivated *GFP* copies, *TrGFP* truncated at the 5′-end and *I-SceI-GFP* interrupted with an 18-bp recognition site for the rare cutting endonuclease I-SceI [26]. Using *TrGFP* of the sister chromatid as a template,

(See figure on next page.)
**Fig. 1** Involvement of c-NHEJ varies widely in repair of Cas9-induced DSBs. **a** Schematic of the HDR reporter with 5 sgRNAs. Repair of Cas9-induced DSBs by HDR between sister chromatids can generate $GFP^+$ cells. Inhibition of c-NHEJ is expected to promote HDR. **b** Effects of DNA-PKcs inhibition on SpCas9-induced HDR in mESC transfected with individual Cas9-sgRNA. Left: Frequencies of SpCas9-induced $GFP^+$ cells; Right: Relative HDR after normalizing DMSO treatment to 1.0. **c** Effects of *DNA-PKcs*, *Ku80*, or *Xrcc4* deficiency on SpCas9-induced HDR in mESC transfected with individual Cas9-sgRNA. Left: Frequencies of SpCas9-induced $GFP^+$ cells; Right: Relative HDR after normalizing both WT cells and $Xrcc4^{+/+}$ cells to 1.0. **d** Schematic of the NHEJ reporter with 6 sgRNAs and their target sites indicated. Repair of Cas9-induced DSBs by c-NHEJ or a-EJ generates accurate NHEJ (accNHEJ) products indistinguishable from undamaged targets and mutagenic NHEJ (mutNHEJ) products represented by $GFP^+$ cells. Inhibition of c-NHEJ promotes a-EJ. **e** Effect of DNA-PKcs inhibition and *Xrcc4* deletion on SpCas9-induced NHEJ in mESC transfected with individual Cas9-sgRNA. Left: Frequencies of SpCas9-induced $GFP^+$ cells; Right: Relative NHEJ after normalizing both DMSO treatment and $Xrcc4^{+/+}$ cells to 1.0. **f**, **g** Cells were transfected with SpCas9-sgRNA expression plasmids and treated with DMSO or NU7441. Four different sites of the *Cola1* (**f**) and *Rosa26* (**g**) locus were targeted by 4 sgRNAs indicated. The efficiency of SpCas9-induced genome editing (left) was calculated as ratios of edited reads to total reads from targeted Illumina sequencing and normalized by transfection efficiency. Relative SpCas9-induced NHEJ (right) was calculated by normalizing the editing efficiency with DMSO treatment to 1.0. Each circle indicates one independent experiment, each in triplicates. Columns indicate the mean ± S.E.M of at least three independent experiments. Significance was detected by two-tailed Student's *t* test and indicated by * for $P<0.05$, ** for $P<0.01$ and *** for $P<0.001$

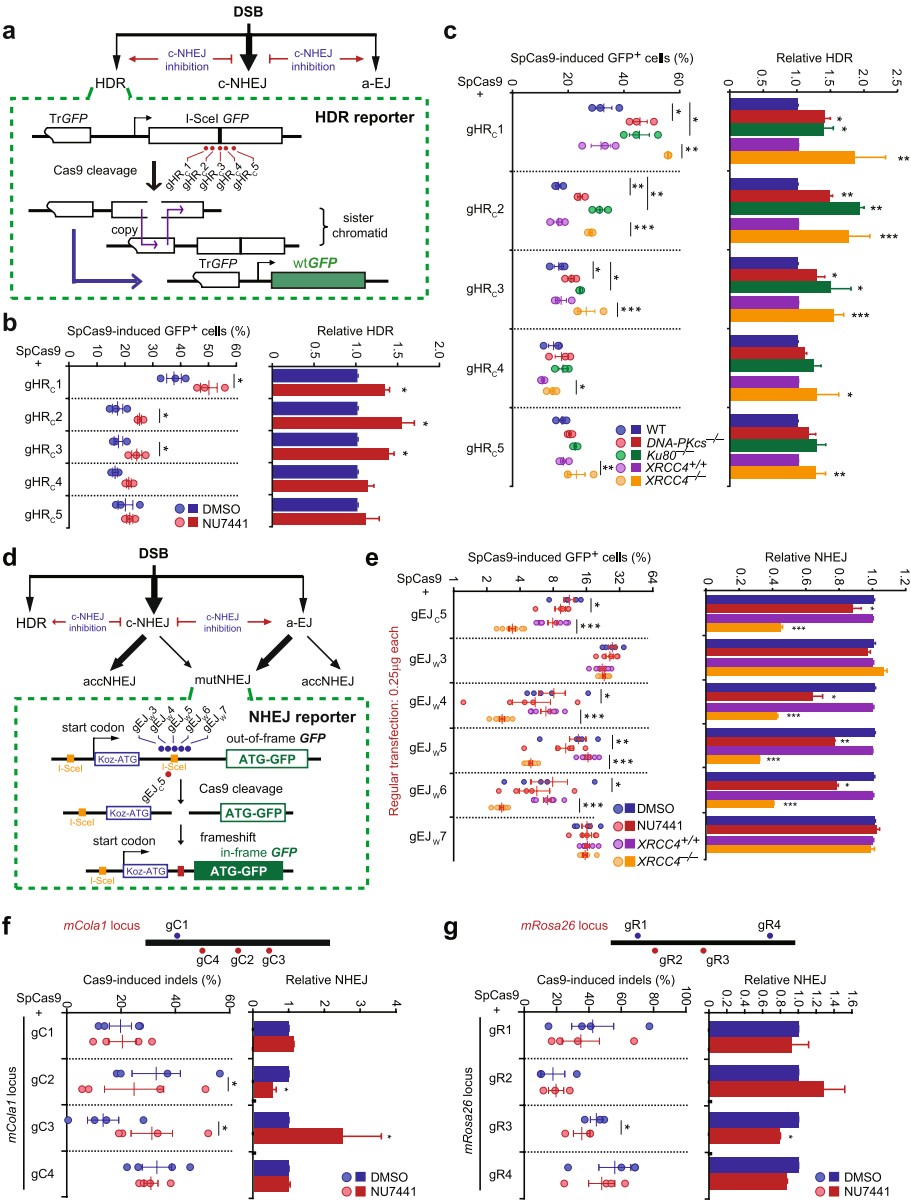

**Fig. 1** (See legend on previous page.)

HDR of a site-specific chromosomal DSB induced by I-SceI or CRISPR nucleases generates a wild-type *GFP* copy and thereby *GFP*$^+$ cells (Fig. 1a). The frequency of *GFP*$^+$ cells induced by I-SceI or CRISPR nucleases reflects the level of HDR. Like I-SceI-induced HDR, Cas9-induced HDR was increased by NU7441 at the sites targeted by gHR$_C$1, gHR$_C$2, and gHR$_C$3 and the extent of this stimulation was different among these three targets (Fig. 1b). Surprisingly, DNA-PKcs inhibition did not elevate HDR induced by Cas9-gHR$_C$4 and Cas9-gHR$_C$5 (Fig. 1b), suggesting a possibility of little c-NHEJ involvement in DSB repair at either the gHR$_C$4 site or the gHR$_C$5 site. We also used CRISPR/Cas9 gene editing to generate isogenic wild-type, *DNA-PKcs*$^{-/-}$, and *Ku80*$^{-/-}$ mESC clones containing the HDR reporter (Additional file 1: Fig. S1b, c). Clonal variation of each genotype was small in I-SceI-induced HDR (Additional file 1: Fig. S1d). Using one

of these clones, along with isogenic $Xrcc4^{+/+}$ and $Xrcc4^{-/-}$ HDR reporter mESC previously established [27], we found that deletion of *DNA-PKcs*, *Ku80*, or *Xrcc4* significantly enhanced HDR induced by $gHR_C1$, $gHR_C2$, or $gHR_C3$ in complex with SpCas9, as well as HDR induced by I-SceI (Fig. 1c and Additional file 1: Fig. S1d). However, deletion of *DNA-PKcs* or *Ku80* stimulated no HDR at the $gHR_C4$ and $gHR_C5$ sites whereas deletion of *Xrcc4* caused limited degrees of HDR stimulation at these two sites (Fig. 1c). Therefore, the extents of HDR stimulation by c-NHEJ inactivation varied among these five different targets from little stimulation at the $gHR_C4$ and $gHR_C5$ sites to stimulation by 90.7% at the $gHR_C2$ target (Fig. 1c). It is possible that c-NHEJ is engaged to different extents among targets where Cas9-induced HDR is stimulated to varying degrees by inactivation of c-NHEJ, and not even engaged at all at the targets where Cas9-induced HDR is not stimulated by inactivation of c-NHEJ in mESC.

**Repair of Cas9-induced DSBs involves c-NHEJ to varying degrees at different targets**

To directly analyze the extent of c-NHEJ involvement in repair of Cas9-induced DSBs at different target sites, we used Cas9-sgRNA to induce site-specific DSBs in an NHEJ reporter integrated in the genome of mESC as done before [28] and analyzed the effect of c-NHEJ inactivation on the frequencies of Cas9-induced insertion or deletion mutations (indels) (Fig. 1d and Additional file 1: Fig. S2a). In this NHEJ reporter, no wild-type *GFP* is translated due to an upstream, out-of-frame translation start site (Koz-ATG), which is flanked by two I-SceI sites sequentially positioned [29]. When a DSB is induced by Cas9-sgRNA at a site between "Koz-ATG" and the *ATG-GFP* coding region, repair by either c-NHEJ or a-EJ can generate indels at the repair junction. In general, because of a 34-bp interval between "Koz-ATG" and ATG for *GFP*, indels with net addition of "3n+2" bp or net loss of "3n-1" bp can change the 34-bp frame-shift to in-frame, leading to production of $GFP^+$ cells. The frequency of Cas9-induced $GFP^+$ cells thus represents the relative efficiency of Cas9-induced indels [28] (Fig. 1d and Additional file 1: Fig. S2a). As c-NHEJ and a-EJ generate different proportions of accurate NHEJ (accNHEJ) products and indel-based mutagenic NHEJ (mutNHEJ) products [4], inactivation of c-NHEJ would channel more Cas9-induced DSBs towards error-prone a-EJ in addition to HDR, increasing the frequencies of mutNHEJ. We found that neither DNA-PKcs inhibition by NU7441 nor *Xrcc4* deletion changed the frequencies of mutNHEJ represented by Cas9-induced $GFP^+$ cells at the two sites targeted by the sgRNA $gEJ_W3$ or $gEJ_W7$, suggesting little involvement of c-NHEJ at these two sites (Fig. 1e). However, inactivation of c-NHEJ inhibited the level of Cas9-induced $GFP^+$ cells at the four sites targeted by $gEJ_C5$, $gEJ_W4$, $gEJ_W5$, and $gEJ_W6$ to different extents, varying from 16.6 to 69.2% (Fig. 1e). Consistently, junction analysis by targeted PCR amplicon deep sequencing revealed that NU7441 had little or modest effect on the editing efficiency, the length distribution of deletions, and the MH usage at the gEJw7 site, but significantly altered the editing efficiency, the length distribution of deletions, and the MH usage at the gEJc5 site (Additional file 1: Fig. S2b-d). As a dominant type of Cas9-induced indels [6, 18, 30–32], 1-bp templated insertions (TIs), which do not generate GFP$^+$ cells in NHEJ reporter cells, occurred much more frequently at the gEJw7 site than at the gEJc5 site (Additional file 1: Fig. S2e). Together, these results indicate that the participation of c-NHEJ varies in repair of Cas9-induced DSBs at different targets in mESC.

Using targeted PCR amplicon deep sequencing, we also measured the frequencies of Cas9-induced indels at two natural genome loci *Cola1* and *Rosa26* in mESC. We found that NU7441 reduced the editing efficiency at the sites targeted by *Cola1* gC2 and *Rosa26* gR3, stimulated by more than 2-fold at the sites by *Cola1* gC3, and had minimal effect at the rest of the sites including gC1 and gC4 for *Cola1* and gR1, gR2, and gR4 for *Rosa26* (Fig. 1f, g). Generally, DNA-PKcs inhibition had little or modest effect on the length distribution of deletions and the MH usage at the target sites where c-NHEJ involvement is limited, but increased deletion length and the MH usage at the target sites where c-NHEJ is engaged (Additional file 1: Fig. S3a-c). Together with varying stimulation of Cas9-induced HDR at different targets by inactivation of c-NHEJ, these results suggested variable involvement of c-NHEJ in CRISPR/Cas9 genome editing at different sites or even no involvement of c-NHEJ at some sites in mESC.

### Target recleavage by Cas9 amplifies the mutagenicity of c-NHEJ

Like I-SceI, CRISPR nucleases generate DSBs with directly ligatable ends. Previous studies have demonstrated that c-NHEJ is intrinsically accurate for these ends [6, 7, 29]. In each round of repair during CRISPR/Cas9 genome editing, about a half of NHEJ products are accurate in repair of Cas9-induced DSBs and the remaining half generate indels [6, 8, 9]. Thus, inactivation of c-NHEJ would increase the use of a-EJ in each round of CRISPR/Cas9 genome editing. Since a-EJ is more error prone, inactivation of c-NHEJ would elevate Cas9-induced indels. It is unexpected that the frequency of Cas9-induced indels was instead inhibited at many Cas9-sgRNA target sites by inactivation of c-NHEJ (Fig. 1e–g). To determine whether this was unique to repair of Cas9-induced DSBs, we used the same NHEJ reporter cells but with the first I-SceI site being deleted to ensure that I-SceI induces single cleavage as Cas9 does and compared the effect of c-NHEJ inactivation on the frequency of Cas9- and I-SceI-induced indels represented by *GFP*$^+$ cells (Additional file 1: Fig. S4a). In consistent with previous findings that inactivation of c-NHEJ stimulates production of I-SceI-induced *GFP*$^+$ cells [28, 33], inhibition of c-NHEJ with NU7441 increases I-SceI-induced *GFP*$^+$ cells by more than 2-fold (Additional file 1: Fig. S4b,c). Given the fact that inactivation of c-NHEJ suppresses Cas9-induced indels at many Cas9-sgRNA target sites, this appears to suggest a difference between Cas9- and I-SceI-NHEJ.

We then wondered what the difference is. While c-NHEJ of both I-SceI- and Cas9-induced DSBs generates a significant level of accurate end-joining products in each round of repair at their respective targets, regenerating the target sites for recleavage, the recleavage by Cas9 may be much more efficient than I-SceI [6, 8, 9, 34, 35]. Thus, in cells expressing abundant Cas9-sgRNA, these target sites could be efficiently recleaved and repaired until indels are introduced and accumulated (Fig. 2a). As a result, c-NHEJ appeared mostly mutagenic for Cas9-induced DSBs and inactivation of c-NHEJ would reduce Cas9-induced indels (Fig. 1e). To test this possibility, we reduced the transfection amount of Cas9 or sgRNA into the NHEJ reporter mESC to limit the Cas9 recleavage in the cells and determined whether Cas9-induced *GFP*$^+$ cells would be stimulated by DNA-PKcs inhibition after Cas9 recleavage is restricted (Fig. 2a). We found that overall Cas9-induced *GFP*$^+$ cells were reduced with a low amount of Cas9-gEJ$_W$6 in the absence of c-NHEJ inhibition (Fig. 2b). This could be explained by either less initial Cas9 cutting,

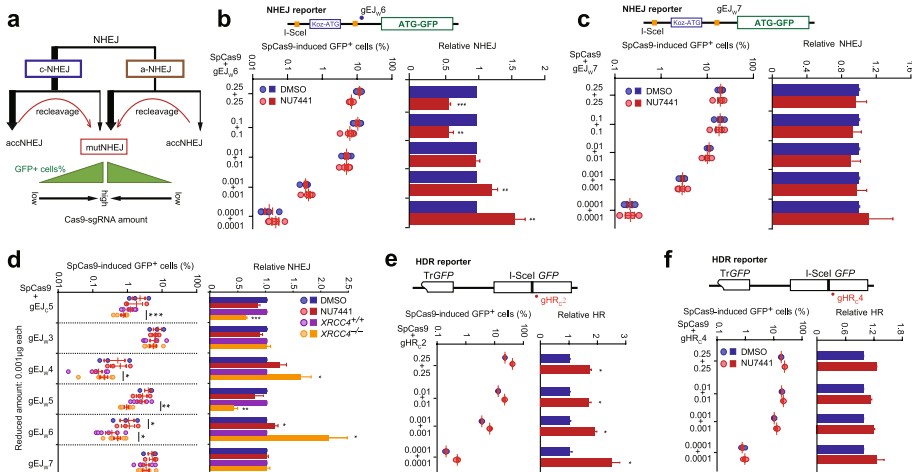

**Fig. 2** Cas9 recleavage increases c-NHEJ-mediated mutations. **a** Model for enrichment of mutNHEJ products promoted by frequent SpCas9 recleavage with increased amount of Cas9-sgRNA transfected. With sufficient amount of SpCas9-sgRNA, accNHEJ products could be recleaved until mutNHEJ products are generated, resulting in enrichment of mutNHEJ products. **b,c** Effect of DNA-PKcs inhibition on NHEJ induced by varying amount of SpCas9-sgRNA. NHEJ reporter mESC were transfected with varying amounts of expression plasmids for SpCas9-gEJ$_W$6 (**b**) or SpCas9-gEJ$_W$7 (**c**) as indicated and treated with DMSO or NU7441. Frequencies of SpCas9-induced GFP$^+$ cells were measured by FACS at 3 days post-transfection and relative SpCas9-induced NHEJ was calculated by normalizing DMSO treatment to 1.0. **d** Frequencies of *GFP*$^+$ cells (left) and relative NHEJ (right) induced by SpCas9-sgRNA at 0.001 μg each, 1/250 of the regular amount (0.25 μg each) transfected into mESC. Each circle indicates one independent experiment, each in triplicates, and the mean of at least three independent experiments is also indicated. **e,f** Effect of DNA-PKcs inhibition on HDR induced by varying amount of SpCas9-sgRNA. HDR reporter mESC were transfected with varying amounts of expression plasmids for SpCas9-gHR$_C$2 (**e**) or SpCas9-gHR$_C$4 (**f**) as indicated and treated with DMSO or NU7441. Frequencies of SpCas9-induced GFP$^+$ cells were measured by FACS at 3 days post-transfection and relative SpCas9-induced NHEJ was calculated by normalizing DMSO treatment to 1.0. For NHEJ and HDR assays in **b–f**, more than 200,000 cells were usually harvested to ensure at least 100 *GFP*$^+$ cells counted for reliable calculation. Columns indicate the mean ± S.E.M. Statistical significance was detected by two-tailed Student's *t* test: \**P*<0.05; \*\**P*<0.01; and \*\*\**P*<0.001

less Cas9 recleavage of accurate repair products, or both. While NU7441 suppressed production of *GFP*$^+$ cells induced by a high amount of Cas9-gEJ$_W$6 at 0.25μg each, the inhibitor started to stimulate production of *GFP*$^+$ cells when the amount of Cas9 and gEJ$_W$6 was both reduced to 0.001 μg (Fig. 2b). In contrast, at the gEJ$_W$7 target, NU7441 did not alter the frequency of *GFP*$^+$ cells induced by Cas9 and gEJ$_W$7 at an amount ranging from 0.25 to 0.0001 μg (Fig. 2c). This further confirms that c-NHEJ is not involved in repair of Cas9-induced DSB at the gEJ$_W$7 target after the interference of target recleavage is minimized.

We then reassessed the c-NHEJ engagement at the 6 Cas9-sgRNA target sites when Cas9 recleavage of the regenerated target is prevented by lowering the transfection amount of Cas9-sgRNA. At the two sites targeted by gEJ$_W$4 and gEJ$_W$6 with the transfection amount of Cas9-sgRNA at 0.001 μg, Cas9-induced indels were also reduced as expected (Fig. 2d). DNA-PKcs inhibition and *Xrcc4* deletion did not suppress production of Cas9-induced *GFP*$^+$ cells any more or even reversed to stimulation at the gEJ$_W$4 and gEJ$_W$6 targets but remained to exert no effect on the level of Cas9-induced *GFP*$^+$ cells at the gEJ$_W$3 or gEJ$_W$7 site (Fig. 2d). In fact, *Xrcc4* deletion elevated the frequency of Cas9-induced *GFP*$^+$ cells by 59.6 ± 14.2% (*P*<0.05) at the gEJ$_W$4 target and 81.5 ± 24.5% (*P*<0.05) at the gEJ$_W$6 target with 0.001 μg of Cas9-sgRNA (Fig. 2d), a reverse from

reduction of Cas9-induced $GFP^+$ cells by 58.1 ±3% and 60.4 ± 2.4% respectively at these two targets with 0.25 μg of Cas9-sgRNA (Fig. 1e). These results again indicate that limiting Cas9 recleavage could elicit the stimulatory effect of c-NHEJ inactivation on Cas9-induced indels in mESC.

Differently, at the $gEJ_C5$ or $gEJ_W5$ target, with the transfection amount of Cas9-sgRNA at 0.001 μg, DNA-PKcs inhibition and *Xrcc4* deletion still inhibited the generation of Cas9-induced $GFP^+$ cells; but this inhibition was reduced (Fig. 2d). At the $gEJ_C5$ target, *Xrcc4* deletion reduced Cas9-induced $GFP^+$ cells by 37.0 ±3.2% (*P*<0.001) with 0.001 μg of Cas9-sgRNA, a smaller reduction than 55.8 ± 2.6% (*P*<0.001) with 0.25 μg of Cas9-sgRNA (Figs. 2d vs. 1e). At the $gEJ_W5$ target, this reduction of $GFP^+$ cells by *Xrcc4* deletion is 57.6 ± 7.5% (*P*<0.001) with 0.001 μg of Cas9-sgRNA but 69.2 ±1.5% (*P*<0.01) with 0.25 μg of Cas9-sgRNA (Figs. 2d vs. 1e). This suggests that Cas9 recleavage could still abrogate the stimulatory effect of c-NHEJ inactivation on Cas9-induced indels at the $gEJ_C5$ or $gEJ_W5$ target sites where limiting Cas9 recleavage does not fully abolish the suppression of Cas9-induced indels by c-NHEJ inactivation. Similar to the $gEJ_W7$ target, no effect by c-NHEJ inactivation was detected at the $gEJ_W3$ target with neither 0.001 nor 0.25 μg of Cas9-sgRNA (Figs. 2d and 1e), suggesting no engagement of c-NHEJ at these two sites. Taken together, these results not only indicate that target recleavage by Cas9 amplifies the mutagenicity of c-NHEJ in CRISPR/Cas9 genome editing in mESC, but also confirm that the involvement of c-NHEJ varies significantly at different targets in repair of Cas9-induced DSBs after target recleavage by Cas9 is partially or fully prevented. Of note, DNA-PKcs inhibition by NU7441 elicited weaker manifestation of c-NHEJ engagement than deletion of *Xrcc4*. This is possible due to different functions of these two factors in NHEJ.

Since minimizing Cas9 recleavage helps maintain a higher proportion of accurate NHEJ products among NHEJ events if c-NHEJ is engaged in repair of Cas9-induced DSBs, inactivation of c-NHEJ would stimulate HDR more strongly. Indeed, at the $gHR_C2$ site, the HDR stimulation by NU7441 was enhanced after the transfection amount of Cas9-sgRNA was reduced (Fig. 2e), confirming c-NHEJ engagement in repair of Cas9-induced DSBs at this site. In contrast, at the $gHR_C4$ site, HDR was not stimulated by NU7441 regardless of the transfection amount of Cas9-sgRNA (Fig. 2f), again indicating that c-NHEJ is little involved at this site.

## Weakening target interaction of Cas9-sgRNA biases repair of Cas9-induced DSBs towards c-NHEJ

To further determine whether c-NHEJ repair of Cas9-induced DSBs is influenced by target interaction of Cas9-sgRNA, we compare the c-NHEJ engagement at the same target by changing the interaction between Cas9-sgRNA and target DNA. In this setting, the effects of DNA sequences or chromatin structures are fixed and only target interaction is allowed to change. We mutated either sgRNA or SpCas9 for two sites targeted by $gEJ_C5$ and $gEJ_W7$ in the NHEJ reporter to reduce Cas9-sgRNA target interaction. In consistent with previous observation that reducing Cas9-sgRNA target interaction generally lowered the efficiency of genome editing [36–39], induction of Cas9-induced $GFP^+$ cells was less efficient with mismatched or truncated sgRNA variants (i.e., the C2A mismatch, the T15A mismatch, and the truncated 16nt for $gEJ_C5$, and A1T, A4C, and T15A for $gEJ_W7$)

and with SpCas9 variants eSpCas9 and SpCas9-HF1, both of which were engineered to have less target interaction (i.e., lower binding affinity) and higher specificity to target DNA (Fig. 3a). The sequences of the sgRNA variants are listed in Additional file 1: Fig. S5a. As in Fig. 1e, DNA-PKcs inhibition and *Xrcc4* deletion reduced Cas9-induced *GFP*$^+$ cells respectively by 30.1% and 62.4% at the site targeted with SpCas9-gEJ$_C$5, again suggesting significant DNA recleavage by Cas9 (Fig. 3a). In contrast, at the same target, the gEJ$_C$5 variants C2A and T15A alleviated or even reversed this NU7441-mediated reduction, and the gEJ$_C$5 variant 16nt and SpCas9-HF1 strongly reversed the reduction by *Xrcc4* deletion as the fold changes of NHEJ stimulation induced by DNA-PKcs inhibition or *Xrcc4* deletion between these Cas9-sgRNA variants and the SpCas9-gEJ$_C$5 20nt control were more than 1 and up to 5.1 (Fig. 3a). At the site targeted by gEJ$_W$7, neither DNA-PKcs inhibition nor *Xrcc4* deletion had effect on the frequency of Cas9-induced *GFP*$^+$ cells as shown in Fig. 1a (Fig. 3a), indicating no engagement of c-NHEJ at this site. However, the gEJ$_W$7 mismatch variant T15A and SpCas9-HF1 allowed significant NU7441-mediated stimulation of Cas9-induced *GFP*$^+$ cells (Fig. 3a). T15A, eSpCas9, and SpCas9-HF1 also elicited stimulatory effect of *Xrcc4* deletion on Cas9-induced *GFP*$^+$ cells at the gEJ$_W$7 target site as the fold changes of this NHEJ stimulation between the Cas9-sgRNA variants and the SpCas9-gEJ$_W$7 20nt control were up to 3.5-fold (Fig. 3a). This suggests that in repair of Cas9-induced DSBs, the weaker the Cas9-sgRNA target interaction is, the more preferentially c-NHEJ is engaged in mESC.

Using endogenous genomic loci, we also found that the editing efficiency with the mismatch variants of *Cola1* gC4 (i.e., C1T and G16C) and *Rosa26* gR4 (i.e., A1C and A16T) was reduced due to weaker target interaction (Fig. 3b, c and Additional file 1: Fig. S5b). Consistently, DNA-PKcs inhibition by NU7441 had minimal effect on Cas9-induced indels at the sites targeted by *Cola1* gC4 and *Rosa26* gR4 (Fig. 1f, g), but stimulated Cas9-induced indels with the gC4 variant G16C and the gR4 variants A1C and A16T (Fig. 3b, c and Additional file 1: Fig. S5b). This again indicates that reducing Cas9-sgRNA target interaction promotes c-NHEJ. Taken together, these results suggest that

(See figure on next page.)

**Fig. 3** Reduced target binding affinity of Cas9-sgRNA shifts the pathway bias towards c-NHEJ in repair of Cas9-induced DSBs. **a** Effects of DNA-PKcs inhibition and *Xrcc4* deletion on SpCas9-induced NHEJ in mESC transfected with individual SpCas9-sgRNA and its variants as indicated. Left: Frequencies of SpCas9-induced *GFP*$^+$ cells; Right: Fold change of NHEJ alteration induced by NU7441 or *Xrcc4* deletion relative to the SpCas9-20nt control, i.e., the ratio of NHEJ change induced by NU7441 or *Xrcc4* deletion for each SpCas9-sgRNA variant to that for the SpCas9-20nt control. **b**, **c** Effects of DNA-PKcs inhibition on NHEJ-mediated genome editing at endogenous loci *Cola1* (**b**) and *Rosa26* (**c**) in mESC transfected with individual SpCas9-sgRNA and its variants as indicated. The frequencies of Cas9-induced indels (left) were calculated as ratios of edited reads to total reads from targeted Illumina sequencing and normalized by transfection efficiency. The fold change of NHEJ alteration induced by NU7441 relative to the SpCas9-20nt control (right) was calculated as the ratio of NHEJ change induced by NU7441 for each SpCas9-sgRNA variant to that for the SpCas9-sgRNA control. **d**, **e** Effects of DNA-PKcs inhibition and *DNA-PKcs*, *Ku80*, or *Xrcc4* deficiency on Cas9-induced HDR in mESC transfected with Cas9-gHR$_C$4 (**d**), Cas9-gHR$_C$2 (**e**), and its variants as indicated. Left: Frequencies of Cas9-induced *GFP*$^+$ cells; Right: Fold change of HDR stimulation induced by NU7441 or deletion of *DNA-PKcs*, *Ku80*, or *Xrcc4* relative to the SpCas9-20nt control. This fold change was calculated as the ratio of HDR stimulation induced by NU7441 or deletion of *DNA-PKcs*, *Ku80*, or *Xrcc4* for each SpCas9-sgRNA variant to that for the SpCas9-20nt control. Each circle indicates one independent experiment, each in triplicates, and the mean of at least three independent experiments is also indicated. Columns indicate the mean ± S.E.M. Statistical significance was detected by two-tailed Student's paired *t* test for frequencies of Cas9-induced *GFP*$^+$ cells or Cas9-induced indels and by one-way ANOVA followed by post hoc Dunnett's test for fold changes of NHEJ alteration or HDR stimulation: *$P<0.05$; **$P<0.01$; and ***$P<0.001$

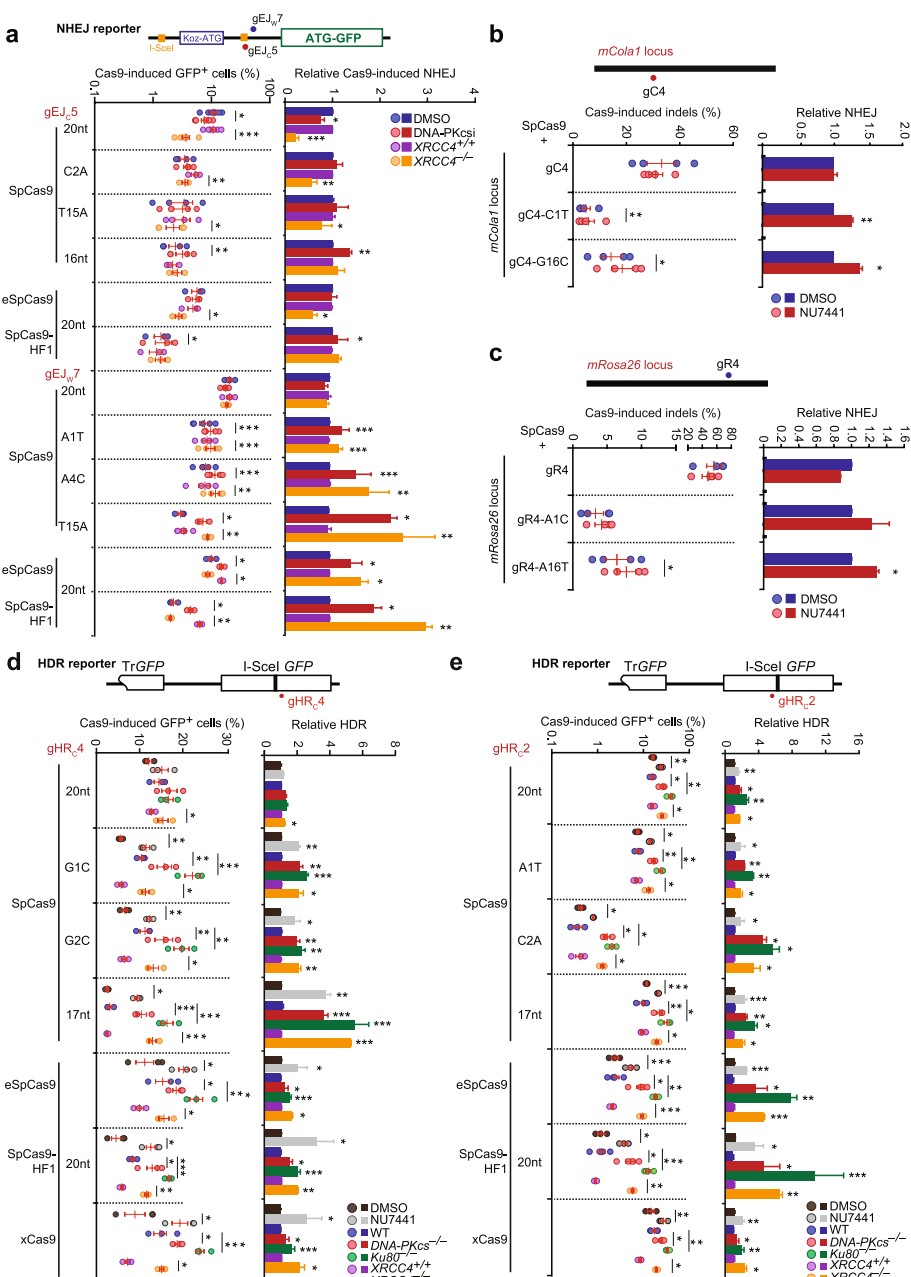

**Fig. 3** (See legend on previous page.)

weakened target interaction of Cas9-sgRNA increase bias towards c-NHEJ in repair of Cas9-induced DSBs in mESC.

## Weakening target interaction of Cas9-sgRNA enhances stimulatory effect of c-NHEJ inactivation on Cas9-induced HDR

Consistently with previous studies [23–25], inactivation of c-NHEJ stimulates HDR induced by CRISPR nucleases as well as I-SceI (Fig. 1b, c and Additional file 1: Fig. S1d). We expected that this stimulatory effect would be further enhanced if HDR were induced by Cas9-sgRNA variants with reduced target interaction, because reducing Cas9-sgRNA

target interaction promotes c-NHEJ. We thus compared HDR induced by mutated Cas9-sgRNA between cells proficient and deficient in c-NHEJ. Due to reduced efficiency of DNA cutting, Cas9-induced HDR was generally less efficient with mismatched or truncated sgRNA variants (i.e., G1C, G2C, and 17nt for gHR$_C$4, and A1T, C2A, and 17nt for gHR$_C$2) and SpCas9 variants eSpCas9, SpCas9-HF1, and xCas9 (i.e., xCas9-3.7), except eSpCas9-gHR$_C$4, SpCas9-gHR$_C$2 17nt, and xCas9-gHR$_C$2 20nt (Fig. 3d, e and Additional file 1: Fig. S5c).

At the site targeted by gHR$_C$4, as in Fig. 1c, Cas9-induced HDR was not affected by DNA-PKcs inhibition, *DNA-PKcs* deletion, or *Ku80* deletion, but modestly stimulated by deletion of *Xrcc4*, Cas9-induced HDR with the sgRNA variants G2C and 17nt was elevated by NU7441 (Fig. 3d). Similarly, deletion of *DNA-PKcs* or *Ku80* elicited stimulatory effect on Cas9-induced HDR with gHR$_C$4 G1C and 17nt, as well as with SpCas9-HF1 (Fig. 3d). In addition, *Xrcc4* deletion stimulated Cas9-induced HDR with the gHR$_C$4 variants (i.e., G1C, G2C, and 17nt) and the SpCas9 variants SpCas9-HF1 and xCas9 by up to 4.3-fold (Fig. 3d). At the site targeted by gHR$_C$2, where Cas9-induced HDR was increased by DNA-PKcs inhibition or deletion of *DNA-PKcs*, *Ku80*, or *Xrcc4* as in Fig. 1c, stimulation of Cas9-induced HDR by NU7441 was further enhanced with the SpCas9 variants such as eSpCas9 and SpCas9-HF1 (Fig. 3e). This HDR stimulation for the SpCas9 variants increased by 1.2- to 2.2-fold as compared to the SpCas9 control (Fig. 3e). Deletion of *DNA-PKcs*, *Ku80*, or *Xrcc4* caused more stimulation of Cas9-induced HDR for SpCas9-gHR$_C$2 C2A, eSpCas9-20nt, and SpCas9-HF1-20nt as this HDR stimulation were enhanced by up to 4.9-fold (Fig. 3e).

However, neither DNA-PKcs inhibition nor genetic inactivation of c-NHEJ by deletion of *DNA-PKcs*, *Ku80*, or *Xrcc4* stimulated more HDR induced by eSpCas9-gHR$_C$4 20nt, SpCas9-gHR$_C$2 A1T, SpCas9-gHR$_C$2 17nt, or xCas9-gHR$_C$2 20nt than that induced by their respective SpCas9-20nt controls (Fig. 3d, e). It appeared that HDR induced by these Cas9-sgRNA variants is as efficient as that by their SpCas9-20nt controls at their target sites (Fig. 3d, e). It is possible that little is changed in the strength of target interaction or the efficiency of target cleavage between the SpCas9-20nt control and Cas9-sgRNA variants at these sites despite modification of SpCas9 or sgRNA. Taken together, these results above confirm that reducing target interaction of Cas9-sgRNA promotes c-NHEJ in mESC, providing the basis for the enhanced stimulatory effect of c-NHEJ inactivation on Cas9-induced HDR.

## Mismatched or truncated sgRNAs reduce Cas9-sgRNA target binding and residence

To determine whether reducing target interaction of SpCas9-sgRNA by mismatched or truncated sgRNAs affect its target binding and target residence at uncleaved and cleaved DNA, we performed in vitro SpCas9-sgRNA target cleavage reaction and electrophoretic mobility shift assay (EMSA) at the four target sites by gHRc2, gHRc4, gEJc5, and gEJw7 and compared the effects of mismatched or truncated sgRNAs on SpCas9-sgRNA target cleavage and target dissociation to those of fully matched 20-nt sgRNA. ~620-bp dsDNA surrounding the gHRc2, gHRc4, gEJc5, and gEJw7 sites in the HDR or NHEJ reporter was amplified by PCR with fluorescently labeled primers and used as a substrate to test the efficiency of DNA cleavage at either of these four target sites by SpCas9-sgRNAs. SpCas9 with fully matched 20-nt sgRNA (SpCas9-20nt) rapidly cleaved a significant

portion of its targets into two fragments within 1 min of the reaction whereas SpCas9 with most of mismatched or truncated sgRNAs started a high level of target cleavage at 5 min or 1 h (Fig. 4a). Compared to SpCas9-20nt, it took a longer time for SpCas9-sgRNA variants to fully cleave their targets (Fig. 4a). At 24 h, all four target sites were cleaved by SpCas9-20nt and SpCas9-sgRNA variants (Fig. 4a). However, consistent with previous study [14], cleaved DNA was little released from the SpCas9-20nt sgRNA-DNA ternary complex at 24 h of reaction, indicating that SpCas9-20nt remained bound to cleaved DNA (Fig. 4b). In contrast, cleaved DNA was more released from the SpCas9-sgRNA variants-DNA ternary complex at 24 h (Fig. 4b), suggesting that reducing target

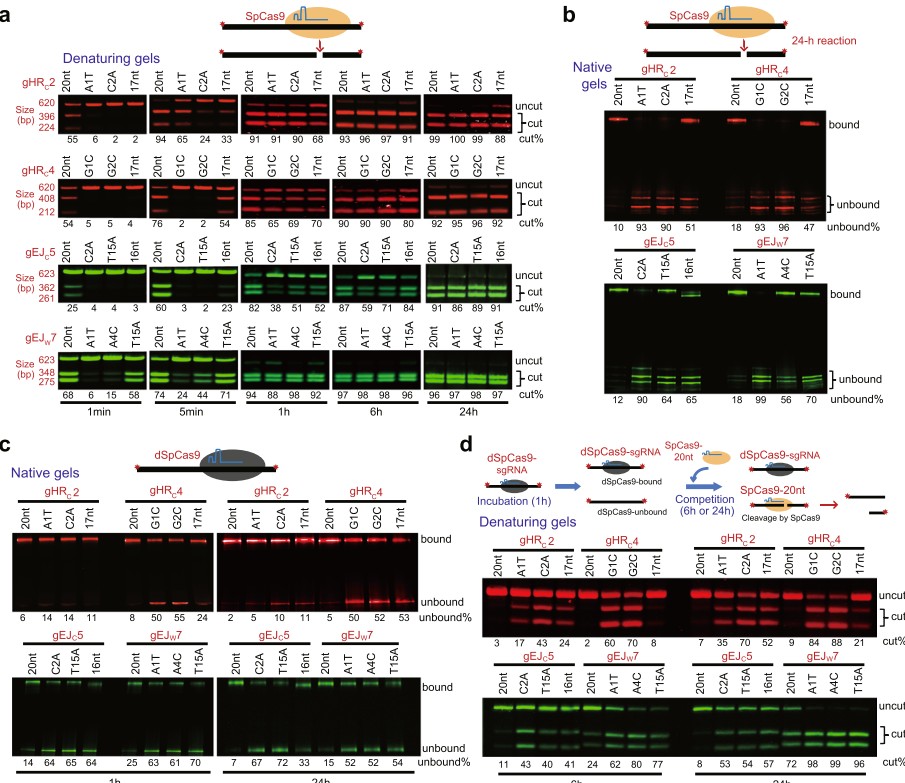

**Fig. 4** Mismatched or truncated sgRNAs reduce target residence of Cas9-sgRNA in vitro at the gHR$_C$2, gHR$_C$4, gEJ$_C$5, and gEJ$_W$7 sites. The schematics for the target binding, cleavage, and dissociation reaction were shown on the top. **a** DNA cleavage by SpCas9-20-nt perfectly matched sgRNAs and SpCas9-sgRNA variants at different time points of the reaction as indicated. Fluorescence-labeled DNA targets were 620 bp or 623 bp as shown. DNA cleavage from the reaction was detected on 2% denaturing agarose gel. **b** Residence of SpCas9-sgRNAs including SpCas9-20-nt control and its variants on cleaved DNA. SpCas9-sgRNAs were incubated with fluorescence-labeled target DNAs from **a** for 24 h. DNA bound with SpCas9-sgRNAs and unbound DNA were resolved by 4–20% native PAGE gel. **c** Target residence of dSpCas9-sgRNAs including dSpCas9-20nt control and its variants. dSpCas9-sgRNAs were incubated with fluorescence-labeled target DNAs from **a** for 1 and 24 h. DNA bound with dSpCas9-sgRNAs and unbound DNA were resolved by 4–20% native PAGE gel. **d** Cleavage of DNA released from the dSpCas9-sgRNA-DNA ternary complex by competing SpCas9-20nt sgRNA. The preassembled dSpCas9-20-nt sgRNA and dSpCas9-sgRNA variant complexes were incubated with fluorescence-labeled target DNAs from **a** for 1 h. The preassembled SpCas9-20nt perfectly matched sgRNA complex was added to compete for binding to DNA targets released from the dSpCas9-sgRNA-DNA complex and cleave DNA for 6 and 24 h. DNA cleavage from the reaction was detected on 2% denaturing agarose gel. The efficiency of target cleavage and target dissociation was calculated as the intensity ratio of cut DNA to total DNA in **a** and **d** and the intensity ratio of unbound DNA to total DNA in **b** and **c**, respectively. The values of these ratios were shown in percentages under each DNA gel

interaction of Cas9-sgRNA increases spontaneous dissociation of SpCas9-sgRNA from its target and, in other words, shorten SpCas9-sgRNA target residence.

For fully matched 20-nt sgRNAs, dSpCas9, sgRNA, and its target DNA at the molar ratio of 5:5:1 assembled nearly all target DNA into dSpCas9-sgRNA-DNA ternary complex in vitro at 1 h and 24 h of the reaction, leaving little unbound DNA (Fig. 4c). By comparison, a significant level of target DNA remained unbound by dSpCas9-sgRNA variants at 1 h and 24 h of the in vitro assembly (Fig. 4c). This again suggests that reducing target interaction could either lessen initial target binding or facilitate target dissociation of SpCas9-sgRNA. To determine whether uncleaved target DNA could be released from the dSpCas9-sgRNA-DNA ternary complex, we added SpCas9-20nt into the mix of the in vitro assembly reaction immediately after the dSpCas9-sgRNA-DNA complex was assembled for 1 h, allowing the competing SpCas9-20nt to cleave the initially unbound DNA or the DNA newly released from the dSpCas9-sgRNA-DNA complex for 6 h and 24 h (Fig. 4d). Little or modestly some target DNA from the dSpCas9-20nt-DNA complex was cleaved by SpCas9 at all 4 target sites, i.e., the gHRc2, gHRc4, gEJc5, and gEJw7 sites, indicating target protection by persistent target residence of dSpCas9 complexed with fully matched 20-nt sgRNA (Fig. 4d). In contrast, a portion of target DNA from the in vitro dSpCas9-sgRNA-DNA ternary complex assembly containing mismatched or truncated sgRNAs was cleaved by SpCas9 at 6 h of the cleavage reaction (Fig. 4d). The level of cleavage by SpCas9 increased further at 24 h for all mismatched or truncated sgRNAs (Fig. 4d), indicating continuous dissociation of target DNA from the dSpCas9-sgRNA-DNA complex with mismatched or truncated sgRNAs. These results again suggest that reducing target interaction of Cas9-sgRNA promote target dissociation and shorten target residence of SpCas9-sgRNA.

### Inactivation of c-NHEJ increases off-target activity of CRISPR/Cas9

As mismatches in base pairing between sgRNA and off-target sites weaken target interaction and shorten target residence of Cas9-sgRNA at off-target sites, it is anticipated that c-NHEJ would be engaged proportionally more at off-target sites than at on-target sites. In addition, target recleavage occurs less at off-target sites. Thus, inactivation of c-NHEJ would increase the engagement of a-EJ at off-target sites. As a-EJ is more error prone even for directly ligatable ends, inactivation of c-NHEJ leads to proportionally more mutNHEJ events and exacerbates off-target effects in CRISPR/Cas9 genome editing. To test this hypothesis, we analyzed the effects of DNA-PKcs inhibition and *Xrcc4* deletion on off-target activities of Cas9 at 7 potential off-target sites for gPnpla3 and 6 potential off-target sites for gMertk in mESC and calculated the fold change of off-target effect due to DNA-PKcs inhibition and *Xrcc4* deletion. We found that both NU7441 and *Xrcc4* deletion slightly reduced on-target editing by Cas9-gPnpla3 and Cas9-gMertk by about 15–21%, suggesting significant on-target DNA recleavage. In contrast, the frequencies of Cas9-induced indels at off-target sites were not reduced by either DNA-PKcs inhibition or *Xrcc4* deletion, but increased at many of these sites (Fig. 5a,b). The fold change of off-target effect was more than 1 and even over 2 at some sites by c-NHEJ inactivation (Fig. 5a,b). This suggests that inactivation of c-NHEJ aggravates off-target effect in CRISPR/Cas9 genome editing in mESC.

Chemical inhibition and genetic inactivation of c-NHEJ are often used to increase the efficiency of Cas9-induced HDR-mediated gene knock-in or replacement [40–45]. Given that DNA-PKcs inhibition by NU7441 stimulated Cas9-induced HDR in the HDR reporter at the targets by $gHR_C1$ and $gHR_C2$ (Fig. 1b), we also performed off-target analysis for 6 potential off-target sites for Cas9-$gHR_C1$ and Cas9-$gHR_C2$ in mESC, respectively. After NU7441 treatment, the frequencies of on-target indels induced by Cas9-$gHR_C1$ and Cas9-$gHR_C2$ were slightly lowered by 20–40%, again indicating significant on-target DNA recleavage. Unlike on-target editing, the frequencies of Cas9-induced indels at the 6 off-target sites were not reduced by NU7441. Instead, these frequencies were stimulated by DNA-PKcs inactivation or *Xrcc4* deletion (Fig. 5c,d), and the fold change of off-target effect was elevated up to 2.5 (Fig. 5c,d). This again suggests that both chemical inhibition and genetic inactivation of c-NHEJ exacerbate off-target effects in CRISPR/Cas9 genome editing in mESC.

### Local transcription does not prevent c-NHEJ engagement in repair of Cas9-induced DSBs

Since the Cas9-sgRNA complex remains bound to its target after DNA cleavage, it is possible that DNA ends are buried in the complex and do not fully elicit the DNA damage response (DDR) or engage any repair pathways before DNA end exposure [13–18]. While some ends are exposed by spontaneous dissociation of Cas9-sgRNA from cleaved target DNA and readily engage c-NHEJ, the others may require local transcription machinery to dislodge the target-bound Cas9-sgRNA complex [21]. If more time is taken for transcription to dislodge Cas9-sgRNA from its target at a given site, spontaneous dissociation at the site would occur less and the catalytically dead Cas9 (dCas9)-sgRNA bound to the site would disrupt transcription more strongly. It is possible that the target dissociation of Cas9-sgRNA by local transcription machinery may generate DNA end configurations unsuitable for binding c-NHEJ factors preventing c-NHEJ engagement. Given different time required for transcription-mediated target dissociation of Cas9-sgRNA at different sites, this might help explain why c-NHEJ engagement varies at different sites in repair of Cas9-induced DSBs. If this was the case, we reasoned that the gene silencing activity (i.e., the transcription-blocking capability) of dCas9-sgRNA at a given target would be negatively correlated with the extent of c-NHEJ participation in repair of Cas9-induced DSBs at the same site. Thus, using the single-copy *GFP* gene expression cassette integrated at the *ROSA26* locus in the genome of mESC, we induced

(See figure on next page.)

**Fig. 5** DNA-PKcs inhibition and *Xrcc4* deletion aggravate off-target effect in CRISPR/Cas9 genome editing. $Xrcc4^{+/+}$ HDR reporter mESC were used for transfection with SpCas9 in complex with gPnpla3 targeting *Pnpla3* (**a**), gMertk targeting *Mertk* (**b**), $gHR_C1$ (**c**), and $gHR_C2$ (**d**) both targeting the HDR reporter. At 6 h post-transfection, cells were treated with DMSO or NU7441. At 72 h post-transfection, gDNA was isolated and the indel frequency at on-target and selected off-target sites was measured by amplicon deep sequencing and calculated as the ratio of edited reads to total reads normalized by transfection efficiency. In an independent set of experiments, isogenic $Xrcc4^{+/+}$ and $Xrcc4^{-/-}$ HDR reporter mESC were transfected and the indel frequency at on-target and selected off-target sites was similarly measured. Fold change of off-target effect after treatment of NU7441 or deletion of *Xrcc4* was calculated as the ratio of the indel frequency with treatment of NU7441 or in $Xrcc4^{-/-}$ cells to that with DMSO or in $Xrcc4^{+/+}$ cells at each off-target site, respectively. Each circle indicates one independent experiment, and the mean of these independent experiments is also indicated. Error bars indicate S.E.M. Statistical analysis was performed by two-tailed Student's paired *t* test for frequencies of Cas9-induced indels and by one-way ANOVA followed by post hoc Dunnett's test for fold changes of off-target effect between NU7441 and DMSO, and between $Xrcc4^{+/+}$ and $Xrcc4^{-/-}$. \*$P<0.05$ and \*\*$P<0.01$

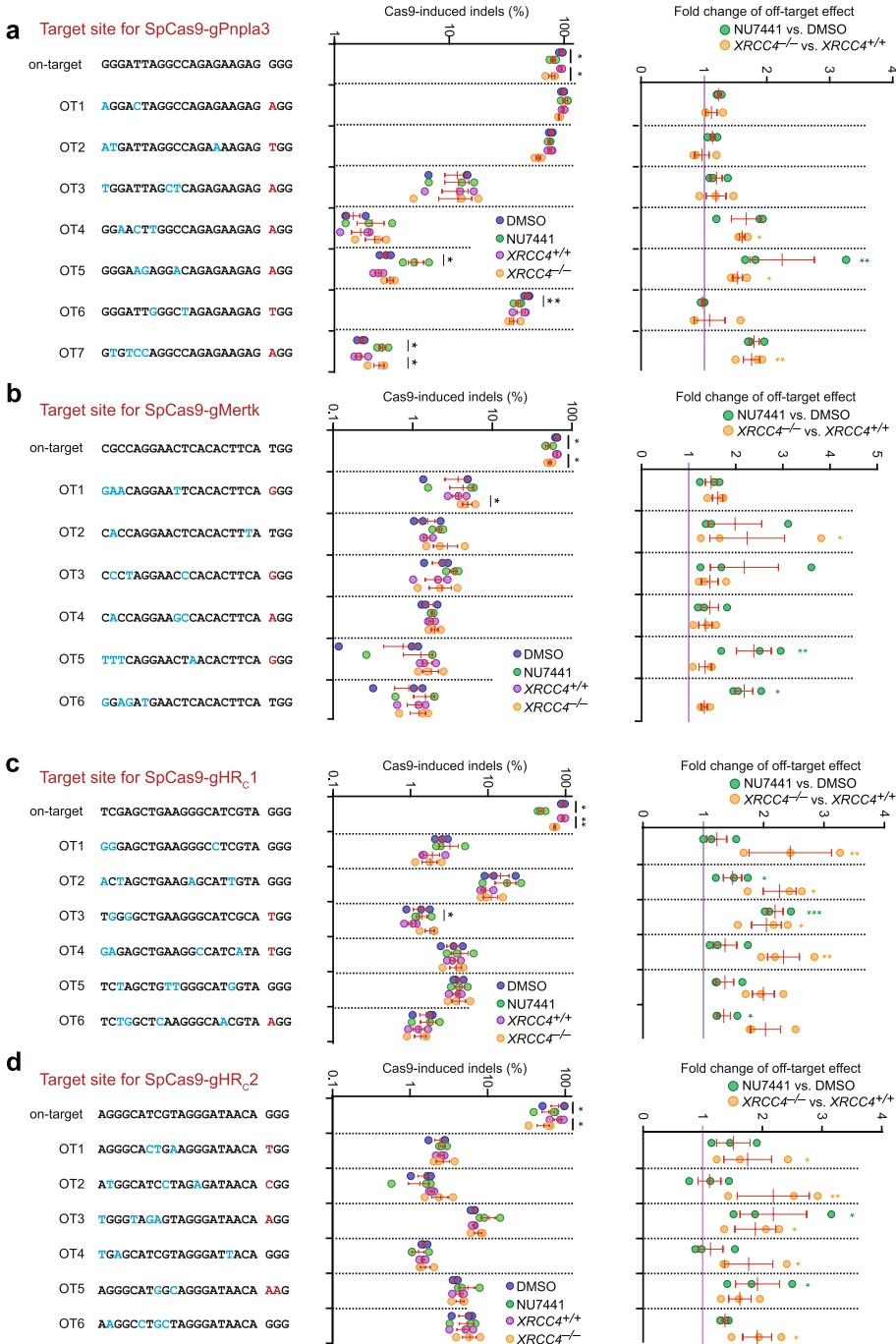

**Fig. 5** (See legend on previous page.)

the *GFP* gene silencing at various sites by catalytically dead SpCas9 (dSpCas9) and also generated *GFP⁻* cells by SpCas9-induced *GFP* knock-out (KO) editing at these sites (Additional file 1: Fig. S6a). We examined any potential correlation between dSpCas9-mediated gene silencing and c-NHEJ involvement in SpCas9-induced *GFP* KO at the same sites. While dSpCas9-sgRNA exhibited variable gene silencing activities at many of these sites (Fig. 6a), the effect of DNA-PKcs inhibition on *GFP* KO varied from no effect

for $gG_C1$, $gG_C4$, $gG_C7$, $gG_C10$, $gG_C14$, $gG_C15$, and $gG_W5$ to about 4-fold stimulation for $gG_C9$ and $gG_W2$ among targets (Fig. 6b and Additional file 1: Fig. S6b). No apparent bias towards either template strand of transcription or non-template strand was detected in both transcription silencing by dSpCas9-sgRNA and DNA-PKcs involvement reflected by stimulation of SpCas9-induced *GFP* KO by NU7441 (Fig. 6c). Importantly, no correlation was observed between transcription silencing by dSpCas9-sgRNA and stimulation of SpCas9-induced *GFP* KO by DNA-PKcs inhibition in mESC (Fig. 6d; $P=0.78$), indicating the possibility that a blocking collision with local transcription have little control over c-NHEJ engagement in repair of Cas9-induced DSBs.

To further determine whether lack of c-NHEJ engagement in repair of Cas9-induced DSBs at some sites is caused by transcription-mediated Cas9-sgRNA target dissociation generating ends unsuitable for c-NHEJ, we analyzed the effect of local transcription blockage on the state of c-NHEJ engagement at a given site where c-NHEJ is little involved. We thus used catalytically dead *Staphylococcus aureus* Cas9 (dSaCas9)-sgRNA to block the translocating RNA polymerase (RNAP), preventing downstream dissociation of SpCas9-sgRNA from its cleaved target (Fig. 6e). Among 6 sgRNAs tested for transcriptional blockage, only $gSaG_W1$ and $gSaG_W2$, in complex with dSaCas9, efficiently reduced gene expression by $26.7\pm4.5\%$ ($P<0.05$) and $47.4\pm 7.3\%$ ($P<0.01$) respectively, indicating a strong capability of blocking RNAP (Fig. 6e). As in Fig. 6b, the frequency of *GFP*⁻ cells induced by SpCas9-$gG_C4$ and SpCas9-$gG_W5$ at a transfection amount of 0.125 or 0.0005 µg for each plasmid was not altered by DNA-PKcs inhibition with NU7441, indicating little c-NHEJ involvement in repair of Cas9-induced DSBs at these two sites (Fig. 6f). This lack of c-NHEJ engagement was little changed by co-transfection with either dSaCas9-$gSaG_W1$ or dSaCas9-$gSaG_W2$ that could block local transcription (Fig. 6f). This indicates that transcription blockage upstream by dSaCas9-sgRNA (e.g., dSaCas9-$gSaG_W1$ and dSaCas9-$gSaG_W2$) would not promote c-NHEJ engagement

(See figure on next page.)

**Fig. 6** Transcription has no effect on c-NHEJ in repair of Cas9-induced DSBs. **a** dSpCas9-mediated transcriptional silencing in mESC containing pPGK-GFP expression cassette. The mean fluorescence intensity of GFP indicates relative transcription. **b** Involvement of c-NHEJ in SpCas9-induced DSBs in mESC containing *pPGK-GFP* expression cassette. Cells were transfected with a low amount of individual SpCas9-sgRNA expression plasmids (0.001 µg SpCas9, 0.001 µg sgRNA, 1/500 of total DNA each) as shown and treated with NU7441 at 6 h post-transfection. Relative NHEJ ($n=3$) was calculated by normalizing DMSO treatment to 1.0. **c** Analysis of strand bias in transcriptional silencing (top) and c-NHEJ involvement (bottom) between transcription template strand and transcription non-template strand targeted by dSpCas9-sgRNA or SpCas9-sgRNA. Transcriptional silencing and c-NHEJ involvement were defined as the percentage of GFP fluorescence intensity reduced by dSpCas9-sgRNA and the percentage of SpCas9-induced NHEJ stimulated by NU7441, respectively. **d** Correlation between dSpCas9-mediated transcriptional silencing and c-NHEJ involvement in repair of SpCas9-induced DSBs. Each circle indicates the level of dSpCas9-mediated transcriptional silencing and stimulation of SpCas9-induced NHEJ by NU7441 at the same target. Two sgRNAs $gG_W5$ and $gG_C4$ are indicated by arrows for their strong effect on transcriptional silencing. **e** dSaCas9-mediated transcriptional silencing in mESC containing *pPGK-GFP* expression cassette. The mean fluorescence intensity of GFP indicates relative transcription. Transcription blockage by dSaCas9-$gSaG_W1$ and dSaCas9-$gSaG_W2$ induced significant transcription silencing. Positions of the targets by dSaCas9-sgRNAs are indicated in the reporter. **f** Little effect of transcription blockage by dSaCas9-$gSaG_W1$ and dSaCas9-$gSaG_W2$ on DNA-PKcs involvement in SpCas9-mediated *GFP* gene editing. *GFP*⁺ cells were co-transfected with SpCas9-sgRNA (SpCas9 and sgRNA at 0.125 µg and 0.0005 µg respectively) and dSaCas9-sgRNA (0.125 µg each), and frequencies of SpCas9-induced *GFP*⁻ cells measured by FACS at 4 days post-transfection. Each circle indicates one independent experiment, and the mean of these independent experiments is also indicated. Error bars indicate S.E.M. Two-tailed Student's paired or unpaired *t* test is indicated by * for $P<0.05$, ** for $P<0.01$, *** for $P<0.001$, and n.s. for not significant. Correlation between transcription silencing and the NHEJ increase was determined by linear regression analysis

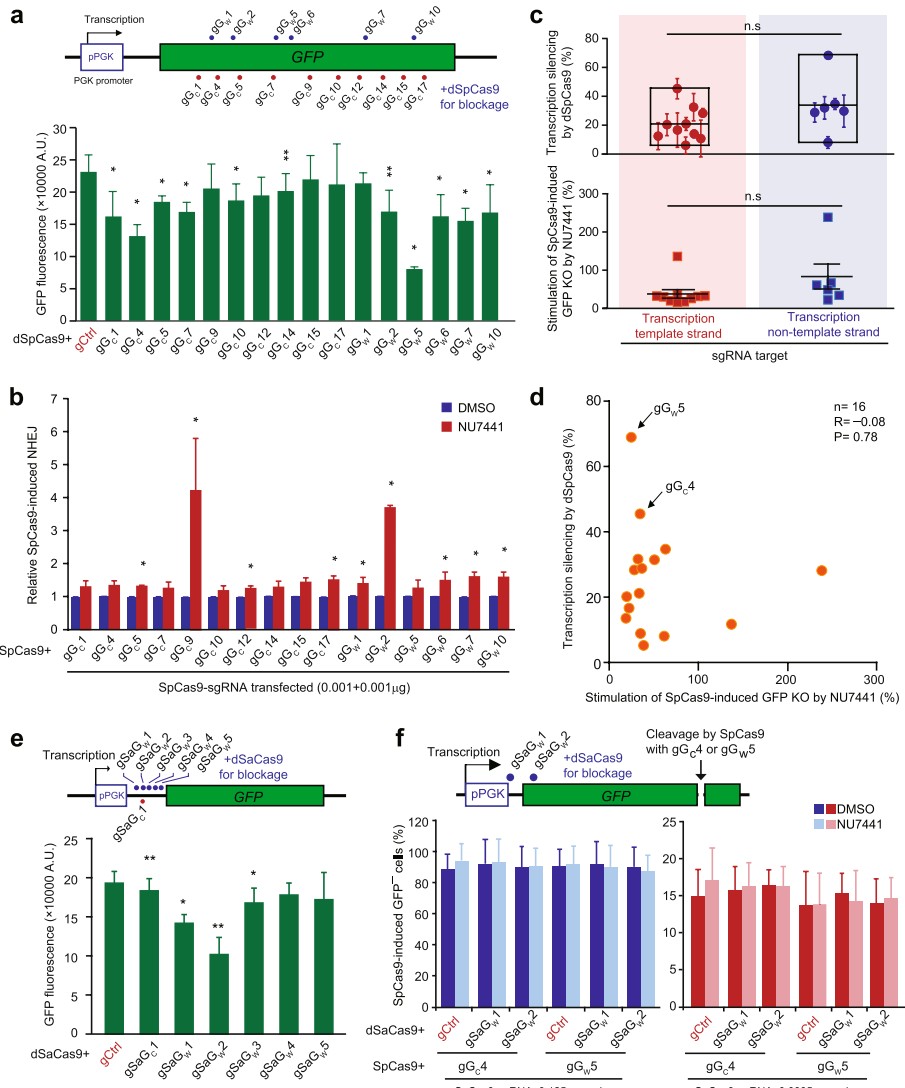

**Fig. 6** (See legend on previous page.)

in repair of SpCas9-induced DSBs at sites where c-NHEJ is little engaged and further excludes the possibility that transcription-mediated Cas-sgRNA target dissociation prevent c-NHEJ engagement in repair of Cas9-induced DSBs in mESC.

### Local replication abolishes c-NHEJ engagement at Cas9-induced DSBs

Like transcription, local DNA replication could also collide with Cas9-sgRNA that remains bound to the cleaved target and dislodge Cas9-sgRNA from the cleaved DNA, generating end configurations that may not be suitable for engaging c-NHEJ. Additionally, the collision with the replication fork occurs in S phase, where HDR is favored for replication-coupled DSB repair. Thus, to investigate whether collision with local DNA replication underlies the biased disengagement of c-NHEJ in repair of Cas9-induced DSBs at some target sites, we transfected HEK293 cells with a plasmid containing an SV40 origin-ATG-GFP-P2A-FLuc NHEJ reporter cassette, together with expression

plasmids for SV40 large T antigen (LT), I-SceI or the SpCas9-gEJ$_W$10 complex, and the *Renilla luciferase* (*RLuc*) gene as internal control. The expression of SV40 *LT* drives bidirectional DNA replication via the SV40 origin, and the expression of I-SceI or SpCas9-gEJ$_W$10 induces a site-specific DSB between the "Koz-ATG" and the "ATG-GFP-P2A-FLuc" (Fig. 7a). Repair of I-SceI- or Cas9-induced DSBs mostly by c-NHEJ generate indels that can proportionally reframe the originally out-of-frame *firefly luciferase* (*FLuc*) gene in the NHEJ reporter plasmids to in-frame in the cells and induce synthesis of active firefly luciferase. The frequency of I-SceI- or Cas9-induced indels can thus be measured as a relative ratio of FLuc to RLuc by luminescence assays. Treatment with NU7441 reduced I-SceI-induced indels by 62.0±7.3% in this assay, but the level of this reduction was similar at 58.0±6.1% with the expression of SV40 *LT* (Fig. 7a), suggesting little effect of local DNA replication on I-SceI-induced indels. However, while Cas9-induced indels were also suppressed by 83.0±2.3% with NU7441, DNA replication initiated by SV40 LT significantly attenuated this repressive effect to 33.0±5.8% (Fig. 7a). This suggests that local DNA replication driven by SV40 LT might inhibit the involvement of c-NHEJ in repair of Cas9-induced DSBs in HEK293 cells.

We also wondered whether a collision with local DNA replication would favor HDR over c-NHEJ in repair of Cas9-induced DSBs by blocking c-NHEJ engagement, thus removing the stimulatory effect of DNA-PKcs inhibition on Cas9-induced HDR. Using U2OS cells containing an integrated single-copy HDR reporter (Fig. 7b), in which an SV40 origin is located between *TrGFP* and I-*SceI-GFP*, we analyzed the effect of DNA-PKcs inhibition by NU7441 on HDR induced by I-SceI and SpCas9. In consistent with the results from mESC (Fig. 1b), NU7441 stimulated HDR induced by SpCas9 in complex with gHR$_C$1, gHR$_C$2, gHR$_C$3, gHR$_C$4, and gHR$_C$5 to different degrees, as well as by I-SceI (Fig. 7b and Additional file 1: Fig. S7a), indicating variable but detectable engagement of the competing c-NHEJ pathway in repair of these I-SceI- or Cas9-induced DSBs. After expression of SV40 *LT*, HDR induced by I-SceI, Cas9-gHR$_C$2, Cas9-gHR$_C$3, and Cas9-gHR$_C$4 were repressed in a gradual and dose-dependent manner (Fig. 7b and

(See figure on next page.)

**Fig. 7** Replication adjacent to targets of Cas9-sgRNA suppresses c-NHEJ in repair of Cas9-induced DSBs. **a** Impact of local replication on DNA-PKcs involvement in NHEJ. SV40 LT can bind to the SV40 origin in a Firefly luciferase-based NHEJ reporter (Luc: Firefly luciferase; SV40 ori: SV40 DNA replication origin) to initiate replication during DNA cleavage by I-SceI or SpCas9-gEJ$_W$10 in 293 cells. NHEJ is represented as relative luciferase activity (i.e., ratio of Firefly luciferase activity to Renilla luciferase activity). Percentage of NHEJ reduction is indicated above each column. **b** Impact of local replication on DNA-PKcs involvement in HDR induced by I-SceI (left) or SpCas9-gHR$_C$2 (right) in HDR reporter U2OS cells. SV40 LT expressed can bind to the SV40 origin in the HDR reporter to initiate replication. SV40 *LT* was titrated as indicated. The fold of the increase is shown above each column. **c** Analysis schematic for SpCas9-induced HDR and NHEJ at the same site of the HDR reporter. HDR bias: HDR reads/ (HDR and NHEJ reads). **d** Effect of Cas9-sgRNA target binding on HDR bias in repair of Cas9-induced DSBs in HDR reporter mESC (left) and U2OS cells (right). **e** Detection schematic for three ends generated by a collision between a DNA replication fork and Cas9-sgRNA at cleaved target. Three primers with different distance to the end, TF1, TF2, and TF3, were screened in pairs for PCR as indicated. **f** PCR detection of palindromic sister chromatid ligation in HDR reporter mESC and U2OS cells. Expression of SpCas9-gHR$_C$4, empty vector control, and SV40 *LT* is indicated. PCR was performed with the primer pair of TF1 and TF2 on gDNA. **g** Repair junction of sister chromatid ligation by subcloning of PCR products and Sanger sequencing. Only two types of products (#1 and #2) were detected with the size and position of deletion (del) and insertion (ins) as indicated. *T: insertion of a thymidine nucleotide. **h** Impact of Cas9-sgRNA target residence on local repair pathway choice. Each circle indicates one independent experiment, each in triplicates, and the mean of at least three independent experiments is also indicated. Columns indicate the mean ± S.E.M. Statistical significance was detected by two-tailed Student's *t* test: \**P*<0.05; \*\**P*<0.01; and \*\*\**P*<0.001

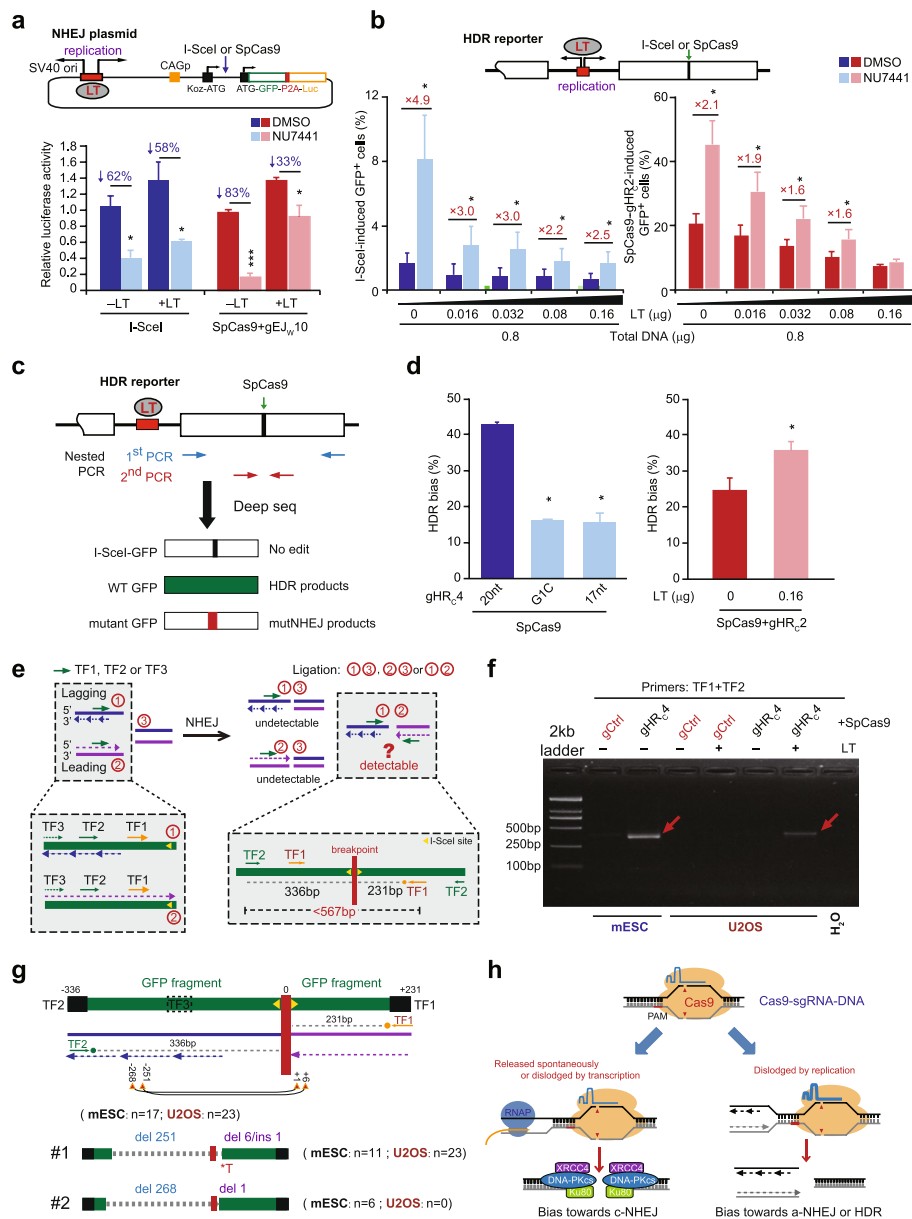

**Fig. 7** (See legend on previous page.)

Additional file 1: Fig. S7b,c). NU7441 stimulated I-SceI- or Cas9-induced HDR, and the expression of SV40 *LT* attenuated this stimulation of I-SceI-induced HDR from 4.9-fold to 2.5-fold or even abolished the NU7441-induced stimulation of Cas9-induced HDR at a transfection amount of 0.032 µg (1/25 of total DNA transfected) for Cas9-gHR$_C$4 and 0.16 µg (1/5 of total DNA transfected) for Cas9-gHR$_C$2 (Fig. 7b and Additional file 1: Fig. S7b,c). This suggests that local DNA replication driven by SV40 LT could collide with both I-SceI and SpCas9-sgRNA after DNA cleavage to dislodge I-SceI and Cas9-sgRNA from its cleaved target and restrict the engagement of c-NHEJ in repair of exposed DSBs.

By restricting c-NHEJ due to a collision with replication fork, DSB repair pathway choice would be biased towards HDR. To test this possibility, we used the HDR reporter to measure the bias between HDR and NHEJ in repair of the same DSB induced by

SpCas9-sgRNA that was tightly bound with its target and by SpCas9-sgRNA variants with weakened target interaction. In the HDR reporter, repair of the same Cas9-induced DSBs around the I-SceI site of *I-SceI-GFP* by HDR generates the "WT *GFP*," whereas NHEJ generates "mutant *GFP*" due to disruption of the I-SceI site (Fig. 7c). We can separate these two repair outcomes in mESC by nested PCR and evaluate the HDR bias (i.e., the ratio of HDR to total edited) by deep sequence analysis. After HDR and NHEJ induced by SpCas9-gHR$_C$4 in mESC, we found the HDR bias was nearly 3-fold lower with gHR$_C$4 variants (G1C and 17nt) than with gHR$_C$4 (Fig. 7d), indicating a reduced HDR preference when the interaction of SpCas9-sgRNA to its target is weakened. At the site targeted by gHR$_C$2, where the HDR stimulation by DNA-PKcs inhibition was fully abolished in U2OS cells by the expression of SV40 *LT* at a transfection amount of 0.16 μg (Fig. 7b), SV40 *LT* expression at the same transfection amount increased the HDR bias by nearly 2-fold (Fig. 7d), indicating a shift of the repair pathway from NHEJ to HDR. Therefore, for Cas9-sgRNA target sites where c-NHEJ is disfavored in repair of Cas9-induced DSBs, it is likely that Cas9-sgRNA at these sites may have a higher probability for collision with local DNA replication after DNA cleavage due to persistent target interaction. Cas9-induced replication-coupled DSBs are subsequently generated with particular end configurations in S phase and favor HDR over c-NHEJ for their repair.

### Palindromic fusion of sister chromatids arises from collision of Cas9-sgRNA at cleaved targets with DNA replication

While spontaneous dissociation of Cas9-sgRNA from cleaved DNA results in a conventional two-ended DSB, DNA replication that releases Cas9-sgRNA from its cleaved target may generate a three-ended DSB, with the leading strand likely forming a blunt end on one sister chromatid and the lagging strand a 3′-overhanging end with long ssDNA on the other sister chromatid (Fig. 7e). These two ends each can rejoin with the other blunt end of the DSB, or have a potential to directly ligate with each other, the latter generating a palindromic chromosome from sister chromatid fusion (SCF) and potentially promoting chromatid breakage-fusion-bridge (BFB) cycles [46–49] (Fig. 7e). Because neither DNA-PKcs nor Ku80 is engaged at Cas9-induced DSBs at the gHR$_C$4 target site for repair in the HDR reporter in mESC (Fig. 1b, c), it is likely that Cas9-gHR$_C$4 at this site may collide with a replication fork after DNA cleavage, generating a three-ended DSB and allowing subsequent fusion of two sister chromatids and production of a palindromic chromosome. Because the product contains palindromic DNA sequence surrounding the junctions, a single primer could in theory be annealed to both the leading strand template and the newly synthesized lagging strand in the repair product for PCR amplification. However, no PCR products were detected from repair of Cas9-induced DSBs at the gHR$_C$4 target site in the HDR reporter in mESC and U2OS cells with a single primer, e.g., TF1, TF2, or TF3 (data not shown), likely due to replication slippage of Taq DNA polymerase over a hairpin structure formed by palindromic DNA sequences in PCR amplification [50, 51]. We thus paired a distal primer to the break (TF2 or TF3) with the most proximal primer TF1 to minimize the length of palindromic DNA sequence in PCR amplification of repair products induced by SpCas9-gHR$_C$4 in the HDR reporter and detected PCR bands over 250 bp in mESC (Fig. 7f and Additional file 1: Fig. S8a). In U2OS cells, these PCR bands were detected only after expression of SV40 *LT*, suggesting

replication-coupled generation of three-ended DSBs and fusion of newly duplicated sister chromatids at the SpCas9-gHR$_C$4 cleavage site (Fig. 7f and Additional file 1: Fig. S8a). This is consistent with the observation that DNA-PKcs inhibition stimulates HDR induced by Cas9-gHR$_C$4 in U2OS cells, but neither in mESC nor in U2OS cells highly expressing SV40 *LT*.

To further confirm that the PCR bands for these repair products were indeed fusions of sister chromatids via end ligation of Cas9-induced DSBs, we first cloned PCR products into a plasmid for Sanger sequencing. Among 40 clones for PCR bands with TF1 and TF2, 17 were from mESC and the rest from U2OS cells. Among 31 clones for PCR bands with TF1 and TF3, 29 were from mESC and the rest from U2OS cells. Sanger sequencing revealed only two sequence variations in each PCR band: DL251R6 and DL268R1 for the PCR band with TF1 and TF2 and DL231R5 and DL386R45 for the PCR band with TF1 and TF3 (Fig. 7g and Additional file 1: Fig. S8b). They all contained some *GFP* sequences inverted around the break site but no palindromic *GFP* sequences, indicating that SCF occur but palindromic sequences may be lost during repair or may not be amplified by PCR [50–54] (Additional files 2, 3 and 4: Table S1-S3). The deletion length in each sequence was distinctly asymmetric surrounding the break point, long at 231bp, 251bp, 268bp, or 386bp at one direction and short at 1bp, 5bp, 6bp, or 45bp at the other direction (Fig. 7g and Additional file 1: Fig. S8b). Lack of palindromic sequences in these PCR products is consistent with previous studies [47, 52–54]. It is likely that the collision between DNA replication and Cas9-sgRNA could generate long ssDNA at the lagging strand end and little or no ssDNA overhang at the leading strand end. Long ssDNA could be easily degraded, generating long deletion and loss of palindromic sequences. It is also possible that one half of palindromic sequences is lost in the first-round PCR due to replication slippage of Taq DNA polymerase over a hairpin structure formed by palindromic sequences [50, 51]. PCR targeted amplicon sequencing also confirmed inverted *GFP* sequences with no palindromic fragments around Cas9-induced DSBs, but with more junction sequence variations (Additional file 1: Fig. S9a,b). Taken together, these results suggest that three-ended DSBs could be generated from release of Cas9-sgRNA at some cleaved targets upon encountering local DNA replication, resulting in inverted duplication via end-joining of sister chromatids.

## Discussions

In in vitro biochemical assays, the target binding affinity of Cas9-sgRNA is primarily determined by the interactions of Cas9-sgRNA with its target [2, 13, 55]. As Cas9-sgRNA binds DNA targets with varying affinities and remains bound for variable time even after DNA cleavage, one key issue often ignored in the development and application of CRISPR/Cas9 genome editing is possible effects of Cas9-sgRNA target residence on DSB repair pathway choice in repair of Cas9-induced DSBs. These effects could hamper our efforts in predicting and improving the efficiency and specificity of CRISPR/Cas9 genome editing. In this study, we demonstrate that target residence of Cas9-sgRNA modulates c-NHEJ involvement in repair of Cas9-induced DSBs, shaping the choices of repair pathway that differ among targets with varying strength of Cas9-sgRNA target interaction (Fig. 7h). It also helps explain why inactivation of c-NHEJ by chemical or genetic approaches enhance HDR-mediated CRISPR/Cas9 genome

editing at some sites [40–45], not at others [56, 57]. Even at a same target, due to different strength and persistence of Cas9-sgRNA target residence, Cas9-sgRNA could be dissociated from cleaved DNA either spontaneously or by local transcription or DNA replication (Fig. 7h), exposing Cas9-induced DSBs with different end configurations for specific repair pathways. Shorter target residence may permit more frequent, spontaneous dissociation or transcription-mediated dissociation of Cas9-sgRNA from its cleaved targets. DSBs exposed in this way can readily engage c-NHEJ. In contrast, stronger and more persistent target residence delays DSB exposure and increases the probability of a collision between Cas9-sgRNA and local replication forks, generating DSB ends that disfavor c-NHEJ and potentially inducing inverted ligation of sister chromatids (Fig. 7h). This may lead to extensive structural abnormalities in chromosomes. Therefore, this regulation of DSB repair pathway choice not only provides insight into how on-target gross chromosomal rearrangements is generated in CRISPR/Cas9 genome editing, but also is potentially a new source for the heterogeneity of mutation profiles in CRISPR/Cas9 genome editing (Fig. 7h).

Like any other types of DSBs, a pathway choice for repair of Cas9-induced DSBs is influenced by many factors such as cell cycle stage, nucleotide composition and configuration of DNA ends, surrounding chromatin structure, and local DNA metabolism [10]. Owing to the innate complexity of DSB repair pathways and the interplay of the factors that regulate DSB repair pathway choice, repair products in CRISPR/Cas9 genome editing are highly heterogeneous in mammalian cells, making it difficult to accurately predict mutation profiles or readily isolate favorable genomic edits in genome editing. Structural and biochemical studies have demonstrated that DSB induction by Cas9-sgRNA is distinct as compared to ionized radiation (IR), radiomimetic drugs, and other DNA endonucleases [2, 13–17, 58, 59]. Prior to DNA cleavage, Cas9-sgRNA binds to its target via the base pairing of sgRNA with target DNA strand and the interactions of Cas9 with both sgRNA and target DNA and initiates the R-loop formation. The R-loop formation in turn activates cleavage of target DNA strand and non-target DNA strand by Cas9. After DSB induction, Cas9-sgRNA remains bound to the cleaved DNA products, concealing the DSBs from access by the DDR and repair machineries [13–18]. In this case, exposure of DSBs is a prerequisite for DSB recognition and repair. Previous studies indicate that Cas9-sgRNA could be released from cleaved targets either spontaneously or by forces such as DNA replication, transcription, or chromatin remodeling in eukaryotic cells, exposing DSBs [11, 12, 19–22]. Because different forms of Cas9-sgRNA dissociation after DNA cleavage may modify Cas9-induced DSBs with different end configurations, it is possible that target residence of Cas9-sgRNA may modulate DSB repair pathway choices by influencing the residence duration of Cas9-sgRNA at cleaved DNA and dissociation of Cas9-sgRNA from it. This regulation may add a new layer of control over DSB repair pathway choices and additional complexity into generation and prediction of mutations in CRISPR/Cas9 genome editing.

It has been previously shown that dSpCas9-sgRNA could block translocating RNAP at some targets, thus repressing gene expression [60]. Further, sgRNAs targeting the non-template DNA strand of transcription generally demonstrate better gene silencing than sgRNAs targeting the template strand [21, 60]. This raises a possibility that Cas9-sgRNA bound to its cleaved target could encounter translocating RNAP and be removed

from cleaved DNA by a collision with transcription in a strand-biased manner. While the collision with transcription on template strand might facilitate genome editing more efficiently than on the non-template strand [21, 60], this bias was not apparent in our study possibly due to a limited number of Cas9-sgRNA target sites we tested and a different cell type we used. Also, considering different target binding affinities of Cas9-sgRNA at different target sites, it is still possible that target dissociation of Cas9-sgRNA by transcription is affected not only by the sgRNAs that anneal to either the template or non-template strand of transcription but also by Cas9-sgRNA target binding affinities. Nevertheless, it was proposed that target dissociation of Cas9-sgRNA at cleaved DNA by local transcription could expose Cas9-induced DSBs for c-NHEJ repair and facilitate recleavage of accurate NHEJ products by Cas9 as a multi-turnover enzyme to increase the level of editing [21]. Our data indicate that c-NHEJ involvement is not altered by the collision of Cas9-sgRNA with transcription no matter which strand the sgRNA is paired with. In particular, it appears that lack of c-NHEJ engagement in repair of Cas9-induced DSBs is not caused by target dissociation of Cas9-sgRNA by local transcription. It is likely that throughout the cell cycle, transcriptional collision, like spontaneous dissociation, may expose DSBs with clean ends that can be recognized and rejoined easily by c-NHEJ factors. Mechanical perturbations such as DNA torsion and DNA stretching imposed by chromatin remodeling may also destabilize the Cas9-sgRNA-DNA complex and dislodge Cas9-sgRNA from the cleaved DNA [61–63]. Assuming the cell cycle would not be altered by these mechanical perturbations, the end configurations of Cas9-induced DSBs exposed in these cases would remain unchanged and not alter DSB repair pathway choice.

However, replication-coupled dissociation of Cas9-sgRNA from cleaved DNA is restricted to the S phase of the cell cycle and generates three-ended DSBs, which appear to reject c-NHEJ for repair. In these three-ended DSBs, the staggered end with a long 3′-ssDNA overhang may not engage c-NHEJ factors such as Ku70/Ku80, and the availability of sister chromatids can further promote HDR, antagonizing c-NHEJ. Therefore, when using Cas9-induced DSBs at individual sites to study regulation of DSB repair pathway choices, we should avoid generalization unless it is taken into consideration how Cas9-sgRNA interacts with its target and is released from it after DNA cleavage. In addition, repair of the three-ended DSBs provides an opportunity for the DNA ends of two sister chromatids to rejoin, not only creating a palindromic chromosome with two centromeres or no centromere [48, 49], but also leaving the third end for potential translocation. Both dicentric and acentric palindromic chromosomes are unstable and serves as a potential source for chromothripsis and complex chromosomal rearrangements including large deletions and insertions at the target site [46–49, 64]. Therefore, this study identifies a potential mechanism underlying on-target chromosomal rearrangements previously detected in CRISPR/Cas9 genome editing [65–69] and suggests that Cas9-sgRNA variants with shorter post-cleavage target residence might be a strategy to reduce the occurrence of dangerous chromosomal rearrangements such as palindromic SCF. In contrast to forced dissociation by DNA replication in S phase, spontaneous dissociation might occur in different stages of the cell cycle, where the DSB repair pathway choice differs partly due to availability of repair factors or substrates for different repair pathways. However, in a population of asynchronous cells, the effect of the cell cycle

stage offsets one another among different cells and does not appear to be significant. Although it remains poorly understood how Cas9-sgRNA is spontaneously released from cleaved targets, the strength and duration of target residence of Cas9-sgRNA may affect spontaneous dissociation of Cas9-sgRNA from cleaved DNA and residence duration of Cas9-sgRNA. Taken together, target residence of Cas9-sgRNA should be integrated with a network of regulators into a decision point for final DSB repair pathway choices at different targets or even at a same target, generating different sets of repair products and contributing to the heterogeneity of mutation profiles in CRISPR genome editing.

Off-target effects are a serious problem in CRISPR/Cas9 genome editing and have greatly limited clinical use of this technology [36]. Due to single or multiple mismatches between sgRNA and off-target DNA, the interaction of Cas9-sgRNA with off-target sites is weaker than that at the on-target site. As a result, the DNA binding affinity of Cas9-sgRNA at an off-target site is much weaker in general, and the residence duration could be shorter [36]. Thus, Cas9-sgRNA at off-target sites, despite being less efficient in DNA cleavage, is dissociated from the cleaved DNA more frequently in a spontaneous manner, exposing DSBs that are more likely to engage c-NHEJ. In addition, because DNA recleavage occurs less at off-target sites than at on-target sites, c-NHEJ, which is innately accurate in repair of Cas9-induced DSBs, generates even less mutagenic repair events at off-target sites. Therefore, while inactivation of c-NHEJ by chemical or genetic approaches is often used to enhance HDR-mediated CRISPR/Cas9 genome editing [40–45], our study revealed that this strategy generate more off-target mutations and cause stronger off-target effects. However, this stimulation of off-target effect was often ignored in CRISPR/Cas9 genome editing [40–44], and should be addressed when we use the strategy to enhance HDR-mediated CRISPR/Cas9 genome editing.

Because the tight target binding of Cas9-sgRNA is excessive for genome editing at some sites, reducing this target binding to some degree may not affect on-target activity but help significantly mollify off-target effects [37]. Various strategies have been designed to remove the excessive target binding and improve the specificity of the modified Cas9-sgRNA [36]. These strategies include truncating 20-nt spacer of a sgRNA to 17-18 nt and mutating the Cas9 residues that are important for non-specific interactions of Cas9 with non-target strand of DNA and the RNA-DNA hybrid [37–39]. We found that when these truncated sgRNA or Cas9 variants such as SpCas9-HF1 and eSpCas9, as compared with wild-type SpCas9, are used in CRISPR/Cas9 genome editing, inactivation of c-NHEJ by chemical inhibitors or genetic modifications may enhance genome editing including NHEJ-mediated gene KO or HDR-mediated knock-in. However, these c-NHEJ inactivation approaches still exert their effects globally in genome editing mediated by these variants, thus increasing off-target activities. Therefore, in either case of Cas9-sgRNA or its high-fidelity variants, a better strategy is needed to locally inhibit c-NHEJ while causing no additional off-target effects in CRISPR/Cas9 genome editing.

## Conclusions

Herein, we demonstrated that target residence of Cas9-sgRNA is a new regulator of DSB repair pathway choice in CRISPR/Cas9 genome editing. Indeed, involvement of c-NHEJ varies in repair of Cas9-induced DSBs at different target sites in mESC. Weakening

target interaction of Cas9-sgRNA biases the repair pathway choice towards c-NHEJ. Thus, inactivation of c-NHEJ elicits more stimulatory effect on Cas9-induced HDR at a target where target interaction of Cas9-sgRNA is weakened. In addition, due to weaker binding at off-target sites, the off-target activity of Cas9-sgRNA is exacerbated by c-NHEJ inactivation, which is often used to promote HDR-based CRISPR genome editing. Our study also revealed a mechanism by which target interaction of Cas9-sgRNA influences DSB repair pathway choices at Cas9-induced DSBs. Weakened target interaction of Cas9-sgRNA increases spontaneous target dissociation and reduces target residence of Cas9-sgRNA at cleaved and uncleaved DNA in vitro, suggesting that target residence may control c-NHEJ engagement in repair of Cas9-induced DSBs in cells. In particular, at sites with stronger target interaction or by extension, faster target dissociation and longer target residence of Cas9-sgRNA, a collision of Cas9-sgRNA with local DNA replication would dislodge Cas9-sgRNA from cleaved DNA, generating three-ended DSBs unsuitable for c-NHEJ repair. During repair of these three-ended DSBs, palindromic ligation of sister chromatids could occur at the break site in mESC and human U2OS cells, potentially leading to on-target gross chromosomal rearrangements, an editing outcome that has been widely reported as a serious concern in applications of CRISPR/Cas9 genome editing [65–69]. Therefore, target residence of Cas9-sgRNA could be an important contributor to significant on-target and off-target mutation variations in CRISPR/Cas9 genome editing by modulating repair pathway choices in repair of Cas9-induced DSBs.

## Materials and methods

### Plasmids

The expression plasmids for truncated and mismatched sgRNAs were constructed as described [37], and the expression plasmids for SpCas9, SpCas9 variants eSpCas9, SpCas9-HF1 and xCas9-3.7, and d SpCas9 were constructed previously [38, 39, 70]. The sgRNA target sequences and respective mutations for SpCas9 and SaCas9 are listed in Additional file 5: Table S4. The HDR reporter plasmid was previously constructed [26, 71]. To generate the reporter plasmid GFP-P2A-FLuc for replication fork-SpCas9 collision assays, the *P2A-Firefly luciferase* (*FLuc*) gene was fused to C-terminal of GFP in the sGEJ reporter previously established [29]. Due to an SV40 replication origin originally present in the sGEJ reporter, DNA replication can be induced by expression of SV40 LT in the GFP-P2A-FLuc collision reporter.

### Cell lines

HEK293 cells were cultured in Dulbecco's modified Eagle's medium (DMEM) containing 10% fetal bovine serum, 1% penicillin-streptomycin, and 2 mM L-glutamine. mESC were cultured as described before [71]. Isogenic $Xrcc4^{+/+}$ and $Xrcc4^{-/-}$ mESC containing the NHEJ reporter and the HDR reporter were established previously [27, 29]. Using a different line of HDR reporter mESC [26], $DNA\text{-}PKcs^{-/-}$ and $Ku80^{-/-}$ HDR reporter mESC along with isogenic wild-type clones were generated by the paired Cas9-sgRNA method as previously described [6]. Briefly, $2 \times 10^5$ mESC were transfected with the expression plasmids for paired sgRNAs and Cas9 in a 24-well plate and were seeded on mouse embryonic fibroblast (MEF) feeder cells for single clones without any antibiotic

selection. Knock-out clones were verified by PCR along with Sanger sequencing and Western blot. The HDR reporter U2OS cells were generated as previously described [72]. To generate the *GFP*⁺ cell lines for *GFP* KO experiments, mESC harboring the NHEJ reporter were transfected with expression plasmids for SpCas9-gI-SceI site and *GFP*⁺ cells were cloned, expanded, and determined by fluorescence-activated cell sorting (FACS) using the Beckman Coulter CytoFLEX flow cytometer.

**Transfection and DSB repair reporter assays**

Transfection of mESC was done with Lipofectamine 2000 (Invitrogen) in 24-well plates as previously described [28, 71]. Total $2 \times 10^5$ mESC harboring the HDR/NHEJ reporter in 200 μL culture medium were transfected with the expression plasmids for Cas9 (0.25 μg) and sgRNA (0.25 μg) or for the pcDNA3β control (0.5 μg). Transfection efficiencies were measured by parallel transfection of 0.05 μg pcDNA3β-GFP expression vector together with 0.45 μg the empty vector pcDNA3β. We excluded experiments in poor transfection efficiencies that were less than 50%. The HDR/NHEJ frequencies were calculated after being corrected by background readings (the pcDNA3β control) and normalized by the transfection efficiencies (the pcDNA3β-GFP vector). Cells were analyzed for the frequencies of induced *GFP*⁺ cells at least 3 days post-transfection but the transfection efficiencies were determined at no more than 3 days post-transfection. For U2OS or HEK293 cells transfection, $1.0 \times 10^5$ cells were seeded on a 24-well plate and grown to 80–95% confluence. 0.8 μg total DNA were transfected by Lipofectamine 2000. Cells harboring the NHEJ or HDR reporter were transfected with pcDNA3β-I-SceI or the expression plasmids for SpCas9-sgRNA or SaCas9-sgRNA as previously described [28, 71].

In dSaCas9-sgRNA transcription blockage experiments, *GFP*⁺ mESC were transfected with the expression plasmids for SpCas9-sgRNA, together with the expression plasmids for dSaCas9-sgRNA. In replication fork-dSpCas9 collision experiments, cells were transfected with the expression plasmids for I-SceI or SpCas9-sgRNA and the SV40 LT, together with the GFP-P2A-FLuc reporter plasmid as needed. If necessary, cells were treated with DNA-PKcs inhibitor NU7441 (TopScience Cat# T6276) at 6 h post-transfection. NU7441 was replaced with a fresh addition of the drug the next day. *GFP*⁺ and *GFP*⁻ cells were determined by FACS at 72 h and 96 h respectively post-transfection. The frequencies of NHEJ, HDR, and genome editing were calculated after being corrected with background readings and normalized with transfection efficiencies as described before [28].

To evaluate the effect of Cas9 dosage on NHEJ, NHEJ reporter cells were transfected with a varying amount of Cas9-sgRNA each at 0.25 μg, 0.1 μg, 0.01 μg, 0.001 μg, and 0.0001 μg. Cells transfected were treated with 2.5 μM NU7441 and analyzed by FACS 3 days post-transfection.

**In vitro DNA cleavage reaction and EMSA**

The in vitro DNA cleavage and EMSA were performed as described previously [13, 14, 55]. SpCas9 nuclease (Z03386, 0.2 μg/μL) was purchased from GenScript Biotech. The dSpCas9 nuclease (PC1351, 0.5 μg/μL) was purchased from Inovogen Biotech. All sgRNAs used for SpCas9 and dSpCas9 were synthesized by GenScript Biotech

and transported in powder packages. The sgRNAs were dissolved in RNA-free water and diluted to 1 μM before use. The primers labeled with either 5′-DyLight-680 or 5′-DyLight-800 were purchased from Takara BioMed (Additional file 6: Table S5). PCR was performed to generate 600–700 bp fluorescence-labeled DNA fragments. For a standard DNA cleavage reaction, 0.5 pmol SpCas9 was mixed with 0.5 pmol sgRNA in Cas9 nuclease reaction buffer (GenScript) and was incubated for 10 min at 37 °C. Then 0.1 pmol target DNA fluorescently labeled was added subsequently, making a total volume of 10 μL with RNA-free water. The reaction was performed at 37 °C and quenched by the addition of 2 μL of denatured loading dye. The cleaved DNA was resolved by 2% agarose gel electrophoresis for fluorescence-imaging analysis.

Similarly, in the EMSA assays, the binding reaction with target DNA was performed with 0.1 pmol target DNA after the dSpCas9-sgRNA complex was assembled with 0.5 pmol dSpCas9 and sgRNA each for 1 h or 24 h. The samples were resolved on 4–20% SurePAGE non-denatured gel (GenScript) in 0.5×TBE buffer at 200 V for 150 min in 4 °C cooling water for fluorescence-imaging analysis. To analyze turnover of dSpCas9-sgRNA and its variants, we preassembled the dSpCas9-sgRNA-DNA complex by incubating 0.5 pmol dSpCas9, 0.5 pmol 20-nt sgRNA or sgRNA variants, and 0.1 pmol target dsDNA for 1 h. We then added 0.5 pmol SpCas9-20-nt sgRNA into the reaction solution to cleave unbound DNA or dissociated DNA from preassembled dSpCas9-sgRNA-DNA for 6 h and 24 h at 37 °C. The reaction was quenched by the addition of 2 μL of denatured loading dye and the cleaved DNA resolved by 2% agarose gel electrophoresis.

The fluorescence imaging of gel electrophoresis was captured by Licor Odyssey infrared scanner and quantified by ImageJ. As both ends of target DNA substrates were labeled, the percentages of cut DNA or unbound DNA were calculated as the ratios of the combined intensity of two cut DNA bands to the combined intensity of total uncut and cut DNA or the ratios of the intensity of unbound DNA bands to the combined intensity of total bound and unbound DNA, respectively.

### GFP fluorescence measurement for CRISPRi in mESC

$GFP^+$ reporter cells were transiently transfected with 0.25 μg each of dCas9 and sgRNA expression plasmids in 24-well plates. Cells were analyzed at 96 h post-transfection for GFP fluorescence intensity using Beckman Coulter CytExpert 2.0 normalized with mCherry transfection efficiency (TE). The GFP fluorescence intensity of cells transfected with each dCas9-sgRNA was calculated as below:

$$I\left(\text{sgRNA}\right) = \frac{I\left(\text{sgRNAmeasured}\right) - I\left(\text{CTRLmeasured}\right) \times (1 - \text{TE})}{\text{TE}}$$

where $I$ (sgRNA): GFP intensity of cells expressing dCas9-sgRNA; $I$ (sgRNAmeasured): GFP intensity of cells after transfecting with dCas9-sgRNA; and $I$ (CTRLmeasured): GFP intensity of cells after transfecting with dCas9-control sgRNA.

### Luciferase assay

HEK293 cells were transiently transfected with GFP-P2A-Luciferase-based NHEJ reporter plasmids together with the expression plasmids for I-SceI or Cas9-sgRNA. The reporter was supplied at 0.025 μg in each well of 24-well plates. At 48 h post-transfection,

cells were harvested and analyzed with the Dual Luciferase Reporter Assay system (Promega). All assays were done in triplicates and all values normalized for transfection efficiency against Renilla luciferase activities as internal control.

### PCR targeted amplicon sequencing

For analysis of targeted genome editing at endogenous genome loci, cells were collected after NHEJ was induced by Cas9-sgRNAs. Genomic DNA (gDNA) was isolated from these cells using a gDNA purification kit (Axygen). The targeted regions were PCR-amplified with Taq DNA polymerase (TsingKe Biological Technology). Respective primers for PCR are listed in Additional file 6: Table S5. The Illumina deep sequencing was performed at Novogene Co. Ltd, and subsequent data analysis was performed as previously described [28].

### Off-target analysis

Potential off-target sites were identified using the latest version of the CRISPR Off-Target prediction website (http://crispor.tefor.net/). All potential sites were ranked by an off-target hit score, and high-ranked potential sites were selected. Off-target sites were amplified by PCR with primers listed in Additional file 6: Table S5 after gDNA extraction from cells transfected with Cas9-sgRNA at 3 days post-transfection. Off-target editing efficiency was determined by Illumina deep sequencing. The off-target rate was determined as the ratio of off-target to on-target mutagenesis levels.

### Three-ended DSB repair analysis

HDR reporter mESC were transfected with Cas9-sgRNA and harvested 2 days post-transfection. For HDR reporter U2OS cells, 0.008 μg of the SV40 *LT* plasmid in 0.8 μg of total DNA was simultaneously transfected to initiate replication. gDNA was collected and the palindromic DNA sequences were amplified by touchdown PCR with primers listed in Additional file 6: Table S5. PCR amplicons were subcloned into CE Entry vector (Vazyme C114-02) and analyzed by Sanger sequencing. Deep sequencing of PCR amplicons was also performed, and their repair junctions were characterized by bioinformatics analysis.

### Statistical analysis

Two-tailed Student's paired or unpaired *t* test was used for statistical analysis of repair frequencies, i.e., the frequencies of Cas9- and I-SceI-induced $GFP^+$ cells, Cas9-induced $GFP^-$ cells or Cas9-induced indels. Two-tailed Student's unpaired *t* test also allowed statistical analysis of comparison between two groups of sgRNAs targeting template strand of transcription or non-template strand, respectively. One-way ANOVA with post hoc Dunnett's multiple comparison test was performed for fold change of NHEJ alteration and HDR stimulation by inactivation of c-NHEJ between Cas9-sgRNA variants and their respective SpCas9-sgRNA controls and for fold change of off-target effect between NU7441 and DMSO, and between $Xrcc4^{+/+}$ and $Xrcc4^{-/-}$ cells. Correlation between transcription silencing and the NHEJ increase was determined by linear regression analysis.

## Supplementary Information

---

Additional file 1: Figure S1. Generation of DNA-PKcs$^{-/-}$ and Ku80$^{-/-}$ HDR reporter mESC clones. Figure S2. The effect of DNA-PKcs inhibition on the patterns of Cas9-induced NHEJ analyzed by targeted PCR amplicon Illumina sequencing. Figure S3. The effect of DNA-PKcs inhibition on the patterns of Cas9-induced NHEJ at the *Cola1* and *Rosa26* locus. Figure S4. Effect of DNA-PKcs inactivation on I-SceI-induced mutNHEJ. Figure S5. Sequences of mismatched or truncated sgRNAs used for weakening target interaction of Cas9-sgRNA. Figure S6. Effect of DNA-PKcs inhibition on SpCas9-mediated gene knockout (KO) at GFP gene. Figure S7. Replication locally disengages c-NHEJ at Cas9-induced DSBs. Figure S8. Analysis of palindromic sister chromatid ligation. Figure S9. Junctions of palindromic sister chromatid NHEJ products.

Additional file 2: Table S1. Junction sequences of sister chromatid fusion by Sanger sequencing.

Additional file 3: Table S2. Junction sequences of sister chromatid fusion products in mESC by PCR targeted amplicon sequencing.

Additional file 4: Table S3. Junction sequences of sister chromatid fusion products in U2OS cells by PCR targeted amplicon sequencing.

Additional file 5: Table S4. List of sgRNAs used in this study.

Additional file 6: Table S5. List of PCR primers used in this study.

Additional file 7. Review history.

---

### Acknowledgements
We thank members of the Xie lab for helpful discussions and the Core Facilities at Hua Jia Chi Campus, Zhejiang University School of Medicine, for technical support. We thank J. Hu at Peking University for the gift of expression plasmids for Cas9 variants and J.T. Wu for critically reading the manuscript.

### Review history
The review history is available as Additional file 7.

### Peer review

### Authors' contributions
S-C.L., Y-L.F., and X-N.S. generated DNA constructs and cell lines, conducted repair reporter assays and genome editing experiments, and performed bioinformatics analysis. R-D.C. conducted replication collision experiments and data analysis. Q.L., J-J.X., J-N.Z., Z-C.H., J-F.X., G-Q.C., Y.Y., Z.C., S-M.X., and H.L. assisted with generation of DNA constructs and cell lines. C.L., H-D.L., and A-Y.X. assisted with bioinformatics analysis. A-Y.X. conceived and developed the project and supervised the study. S-C.L., Y-L.F., X-N.S., R-D.C., and A-Y.X. analyzed and discussed the data, and S-C.L., Y-L.F., and A-Y.X. wrote the manuscript. All author(s) read and approved the final manuscript.

### Funding
This work is funded by the National Natural Science Foundation of China (No. 31870806 and No. 31671385 to A.Y.X., and No. 32071439 to Y.L.F), the Natural Science Foundation of Zhejiang Province (LQ20C050004 to S.C.L), and Fundamental Research Funds for the Central Universities in China (2019QNA7031 to Y.L.F).

### Availability of data and materials
Deep sequencing raw data are available in the Sequence Read Archive (SRA) under accession number PRJNA726333 (https://www.ncbi.nlm.nih.gov/sra/ PRJNA726333) [73].

## Declarations

### Ethics approval and consent to participate
Not applicable.

### Consent for publication
Not applicable.

### Competing interests
The authors declare that they have no competing interests. Chao Lou and Hao-Dan Li are employees at Shurui Tech Ltd.

### Author details
[1]Innovation Center for Minimally Invasive Technique and Device, Department of General Surgery, Sir Run Run Shaw Hospital, Zhejiang University School of Medicine, Hangzhou, Zhejiang 310019, People's Republic of China. [2]Institute of Translational Medicine, Zhejiang University School of Medicine and Zhejiang University Cancer Center, Hangzhou, Zhejiang 310029, People's Republic of China. [3]Department of Biochemistry and Molecular Biology, Zhejiang University School of Medicine, Hangzhou, Zhejiang 310058, People's Republic of China. [4]The First affiliated Hospital, Zhejiang University School of Medicine, Hangzhou, Zhejiang 310019, People's Republic of China. [5]Institute of Hepatopancreatobiliary

Surgery, Chongqing General Hospital, University of Chinese Academy of Sciences, Chongqing 400013, People's Republic of China. [6]Shurui Tech Ltd, Hangzhou, Zhejiang 310005, People's Republic of China.

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

## 

