## [Additional file 7. Review history. · Genome Biology]

Review History

First round of review

Reviewer 1

Are you able to assess all statistics in the manuscript, including the appropriateness of statistical tests used? Yes, and I have assessed the statistics in my report.

Comments to author:

In the manuscript "Target interaction of Cas9-sgRNA influences DNA double strand break repair pathway choices in CRISPR/Cas9 genome editing", Liu et al. designed several assays to study the influence of Cas9-sgRNA post-cleavage residence on the choice of DSB repair pathways. The authors found that c-NHEJ is not even engaged at tested sites and blocking the core factors of c-NHEJ would change the mutagenic spectrum of Cas9 repair outcomes. Moreover, the authors found that DNA replication but not transcription can efficiently remove resident Cas9-sgRNA to influence DNA repair choice. The findings are interesting and help gain deep insight into the genome editing field. However, the manuscript is immature and needs to be well-controlled to arrive the final conclusions.

1. In the Introduction, the alternative NHEJ should be alternative end joining and a-EJ in short as in the ref 4. "Target interaction" is not clear, the authors may consider a better one instead.

2. In Fig. 1a and Fig. 1d, the tandem Cas9 target sites are distributed in a region around the I-SceI cut site. The authors need to swap the order of Cas9 target sites to re-analyze the GFP percentages to avoid potential position effects on these sites, e.g. placing gHRc4 and gHRc5 in the middle and so on. The authors may provide a better schematics to show the orientations and distances of these tandem Cas9 target sites.

3. In Fig. S1, the authors should employ southern blot or western blot to verify these knock out strains. Moreover, the mESCs have two alleles and the sequences of both alleles should be showed.

4. It's hard to distinguish the repair products of c-NHEJ and a-EJ through the GFP truncation assay in Fig. 1d and 1e. For example, 1-bp insertions are routinely found for NHEJ-mediated Cas9 outcomes and may switch between out-of-frame and in-frame (this point should be explained and discussed). To better distinguish these by-products, the authors should perform targeted sequencing and then micro-homology analysis to confirm their findings.

5. For the targeted sequencing data at mCola1 and mRosa26 sites, the analysis of micro-homology usage should be performed to get more information about the NHEJ or a-EJ products.

6. The inhibition of NHEJ leads to the change of both a-EJ and HR. The authors should check the change of both a-EJ and HR in Fig. 2b and 2d. And more repeats should be performed to verify the significance in Fig. 2c. Moreover, the numbers of cells detected should be indicated in the panels.

7. In Fig. 2f, the PK inhibitor and knock our strains showed different phenotypes, which requires explanation.

8. The authors may perform competition assays to confirm that the mutations reduce target interaction of Cas9-sgRNA in Fig. 3. Moreover, the right panels are over-processed and hard to catch the RELATIVE fold-change. Original fold changes need to be showed for each sample.

9. Different sites are used for Fig 3a-c vs 3d-e. The changes of HR and a-EJ after NHEJ inhibition should be performed at the same sites to get the full pictures of repair choice change after NHEJ inhibition.

10. The mutation sites C2A and T15A in Fig. 3 can also be considered as off-targets for gEJc5, however, the cutting efficiencies were decreased for these sites after XRCC4 knock-out, not very consistent with the findings in Fig. 4. The authors may consider to reclaim the conclusions that NHEJ inhibition has more impact on on-target cleavage than off-target cleavage.

11. In Fig.5, the blockage of transcription by dCas9 was not so strong, the authors may consider remove the pPGK promoter to suppress GFP transcription and perform the analysis in parallel with pPGK proficient cells.

12. Cas9-sgRNA are reported to stay at the sgRNA-bound ends, so the model in Fig. 6h should be re-drawn and DNA replication comes from the other ends in Fig. 6b should be performed as a control to draw the conclusion that DNA replication machinery remove resident Cas9-sgRNA to affect DNA repair choice.

13. The sequence lengths in Table S1 are not consistent with the PCR band size in Fig. 6f. In mESCs, LT is not required to form TF1-TF2 bands. More explanations are required.

Reviewer 2

Are you able to assess all statistics in the manuscript, including the appropriateness of statistical tests used? Yes, and I have assessed the statistics in my report.

Comments to author:

In this manuscript entitled "Target interaction of Cas9-sgRNA influences DNA double strand break repair pathway choices in CRISPR/Cas9 genome editing" Liu et al., aim to study how the CRISPR/Cas9-sgRNA combination can affects the repair choice of DNA double strands breaks. Previously the authors developed various DSB reporter constructs for both NHEJ and HDR as tests for I-SceI-induced DSBs. In this study these constructs are used to study CRISPR/Cas9-induced DSBs.

The hypothesis that the affinity of Cas9-sgRNA to its break site may influence the repair pathway usage is interesting to the field, since it could impact the design of genome editing experiments.

The manuscript is clearly written and the authors discuss various KO models of c-NHEJ and test multiple Cas9 and sgRNA modifications. However, my major concern of this study is that the experimental design is in my opinion not suitable to investigate the question the authors want to address. Several firm claims are made in this work that are not proven e.g. c-NHEJ is not engaged at some CRISPR-sgRNA target sites; different persistence of Cas9-sgRNA target interaction exposes Cas9-induced DSBs with different end configurations for specific repair pathways and presence of Cas9 recleavage due to the accurate repair by c-NHEJ.

Other conclusions are not original, as published studies have conducted multiple guide repair analysis with larger sets of guides that already showed the wide variety of repair at the different target sites (van Overbeek, et al. Mol Cell. 2016; Shen, et al.; Nature. 2018; Allen, et al. Nat Biotechnol. 2018; Chakrabarti, et al. Mol Cell. 2019; Chen, et al. Nucleic Acids Res. 2019).

The connection of repair pathway choice and Cas9 dissociation is interesting, but the authors do not show the actual Cas9 variation in dissociation themselves.

Lastly, the paper shows limited mechanistic insights. It is unclear why the authors suggest the three-ended

DSB mode since no palindromic fusions were detected. The finding of inverted GFP sequences is not sufficient to support this model. It is also unclear how often the inverted GFP arises during DNA repair. With additional experiments to support its conclusions, this study may be of some interest to a specialized journal.

Major comments:

1. The NHEJ reporter is not a suitable system to compare the various sgRNAs for their contribution of c-NHEJ. The authors write that in theory, a third of the indels can lead to GFP+ cells using this reporter, however, it is well known that CRISPR-induced breaks lead to very specific repair, which is this is sgRNA dependent (Brinkman et al. *Nucleic Acids Res.* 2014; van Overbeek, et al. *Mol Cell.* 2016; Shen, et al.; *Nature.* 2018; Allen, et al. *Nat Biotechnol.* 2018; Chakrabarti, et al. *Mol Cell.* 2019; Chen, et al. *Nucleic Acids Res.* 2019). Chakrabarti et al describes the existence of precise and non-precise guide, where the precise sgRNA can be repaired by mainly one type of indel. If this precise indel is not in the correct frame, the NHEJ reporter used in this paper will not pick up this repair. In that case the authors may incorrectly conclude that this guide has little involvement of c-NHEJ. Moreover, in the presence of NU7441, repair by a-NHEJ may occur. This type of repair can result in a different indel in another frame, which again may be not detected with the reporter used. Each sgRNA may result in a different distribution of repair over the 3 possible frames and they should all be considered. Therefore, the researchers should sequence the repair indels for each sgRNA and report which indels are observed in DMSO and NU7441 conditions.

2. In almost all figures the authors consider repair to be just binary e.g. In figure 1a-c the authors do not take into consideration repair by a-NHEJ (either with microhomology or without), even though these pathways are reported to play a role in CRISPR/Cas9-induced DSB repair. a-NHEJ repair even may primarily compete with the HDR repair since both pathway are dependent on resection. Repair by a-NHEJ can result that less HDR repair occurs at a particular DSB. Therefore the conclusion that there is little or no c-NHEJ involvement because usage of NU7441 did not increase GFP+ in the HDR reporter needs nuance.

In figure 1d-g, the authors do not consider the repair by HDR, but only c-NHEJ and a-NHEJ, which raises similar concerns.

In figure 1f-g, only total indels are reported, while it is known that c-NHEJ and a-NHEJ can generate different types of indels (van Overbeek, et al. *Mol Cell.* 2016; Shen, et al.; *Nature.* 2018; Allen, et al. *Nat Biotechnol.* 2018; Brinkman et al. *Mol Cell.* 2018; Chakrabarti, et al. *Mol Cell.* 2019; Chen, et al. *Nucleic Acids Res.* 2019). The total amount of indels does not say anything about the involvement of c-NHEJ repair alone. Since all repair pathways are present in the cell, repair by c-NHEJ, a-NHEJ and HDR can compete with each other and can take over when one repair pathway is inhibited (van Overbeek, et al. *Mol Cell.* 2016; Brinkman, et al. *Mol Cell.* 2018; Schep et al. *Mol Cell.* 2021). The authors should present their data more thoroughly, in view that a-NHEJ or HDR may influence their data and not immediately conclude that when the amount of indels or GFP+ cells in NU7441 condition is similar as in the DMSO conditions there is no involvement of c-NHEJ repair at this CRISPR-induced DSB.

3. The authors mention several times that CRISPR/Cas9-induced DSB are repaired accurately. This is still a matter of debate. High c-NHEJ precision was reported when DNA breaks were introduced by I-SceI (Guirouilh-Barbat et al, *Frontiers in Genetics*, 2014). Also, it has been reported in vitro and also in wt yeast that blunt breaks are preferentially joined imprecisely (Boulton, *Nucleic Acids Res.* 1996; Schär et al, *Genes and Development*, 1997). Kinetics experiments suggested that CRISPR/Cas9-induced breaks were repaired mainly mutagenic (Brinkman, *Mol Cell.* 2018). The fidelity of DSB repair upon induction of Cas9 in combination with two co-transfected sgRNAs that target adjacent sequences was shown to result in accurate re-joining in varying degrees (Guo et al. *Genome Biology.* 2018). But re-joining of single end breaks may be different from re-joining the ends of two different breaks.

The authors suggest a cycle of Cas9 cutting and perfect repair that terminates when a 'rare' imperfect

repair event occurs, this seems contradictory with the long residence time of Cas9 reported by others (Richardson et al., Nature biotechnology, 2016; Ma et al. The Journal of cell biology 2016). The presence of target recleavage by Cas9 is not convincingly shown from their data and the authors should not conclude that Cas9 recleavage occurs and that c-NHEJ is accurate without more evidence.

4. It is not clearly presented how well the transfection efficiency of the sgRNAs was in the various experiments. The method states that the figures are normalized for transfection efficiency, but how normalization was performed and what the average percentage is of the transfection efficiency is missing. This is especially important for figure 2b-f. In figure 2b, the authors claim that when reducing the amount of Cas9-sgRNA, the NU7441 inhibitor initiates a reverse effect on repair and begins to stimulate GFP production. The figure shows that this effect takes place when only ~0.2% to ~0.02% GFP+ cells are detected. This effect may only represent a handful of cells when considering 2×10^5 cells were transfected. Low DNA input for transfection results in a low number of events and my concern is that there are too few events to make data reliable. The authors should clarify their experimental conditions and whether this experiment represent a large enough pool of cells to draw solid conclusions, if not the experiment should be performed with more cells.

5. The method section is inadequately described. Several times is a referred to previous publications. In the referred publication, the reader is referred to yet another publication. In this treasure hunt, many parts of the experiments remained unclear. E.g. the sequence of the reporter plasmids, transfection conditions, the way the transfection efficiency was calculated and was used for normalization. An adequate reference to the basic paper(s) would be of great help for the interested reader. Also the experimental conditions should mentioned in the manuscript.

6. A shortcoming in the paper is that it does not state anything about cell viability after inducing DSBs, since the percentage of GFP can vary widely when a sgRNA affects cell viability. The authors discuss that they found a potentially source for chromothripsis and complex chromosomal rearrangements, events likely to influence the cell viability. Also in cells with KOs in components of the c-NHEJ pathway, the cell may not be able to cope with the DNA repair as well. Cell viability can be measured with various compounds followed by fluorescence and it would improve the outcome of the data.

7. In my opinion the authors provide no new mechanistic insight or explanations for possible differences in repair choice. The authors argue that Cas9 stickiness could influence repair choice. Unfortunately, the authors don't show this in their own system. The effects of the asymmetry of this stickiness on gene targeting was the major thrust of the Corn Nature Biotechnology paper. At least a simple in vitro assay could have been performed with some of their CRISPR guides, Cas9 and target sequence (e.g. PCR fragment). After incubation, the reaction products can be separated by non-denaturing agarose electrophoresis. If a sticky complex is present the DNA will appear as unbroken on the gel.

Minor points:

- The aim of the authors was to study how the varying affinities of Cas9-sgRNA complex after DNA cleavage influences repair and hamper the efforts to predict efficiency and specificity of CRISPR/Cas9 genome editing.

Could the authors elaborate in the discussion on how can we take target interaction along in sgRNA design, based on their research findings?

- In several graphs the y-axes state "induced relative c-NHEJ", which is an interpretation of the data. The axis should describe what you measure e.g. "relative induced GFP"

- How come figure 2b and 2f does not show the same results for gEJw6? The GFP+ cells seems to be much higher in figure 2f in the 3 replicates.

- Unexpectedly, the XRCC4+/+ mutant is affected (almost) as much as XRCC4-/- cells when transfected with gEJw4 and gEJw6 in figure 2f. Do the authors have an explanation for this?

- The counts of the lines should continue over the whole manuscript and not start recounting at each page. This makes it hard to refer to the correct line.

- Typo's. Word "at" too much in the last sentence of in the part 'Inactivation of c-NHEJ induces varying stimulation of Cas9-induced HDR among targets'.
"mollify" instead of "modify" in discussion page 27 line 21.

Reviewer 3

Are you able to assess all statistics in the manuscript, including the appropriateness of statistical tests used? No, I do not feel adequately qualified to assess the statistics.

Comments to author:

The manuscript by Liu, Feng, and co-authors adds to the recent literature in understanding DNA repair after genome editing. This is an important area of ongoing research with much uncertainty in the DNA repair pathway choices and competing repair pathways. This manuscript seeks to address if persistent binding by Cas9 to the target locus favors certain repair outcomes and if decreasing the energy of binding can alter these outcomes. To explore the hypothesis, the authors use a truncated GFP repair reporter with a panel of gRNAs to determine bias toward HDR and an out-of-frame GFP reporter to measure NHEJ. They created a series of genetically modified cells deficient in c-NHEJ repair proteins. The authors measured editing outcomes with mismatched gRNAs and reported high-specificity Cas9 variants to determine if lower binding energy affects gene editing outcomes. Finally, they attempt to determine if transcription or replication impact DNA repair outcomes.

Understanding the risk of on-target chromosome rearrangements is also important considering ongoing clinical translation of such technologies and the tendency of conventional PCR and NGS assays to exclude such results. The manuscript uses one figure to describe transcriptional control and the lack of measured effect on NHEJ. The final figure indicates the potential of chromosomal rearrangements. Overall, I think this manuscript warrants publication after concerns are addressed. It adds new observations to the published literature on 1) the complexity of DNA repair 2) indicates that promoting HDR may have the unintended consequence of increasing off-targets and 3) adds to growing concerns of on-target chromosome alterations.

The following are my major concerns and a few minor comments for consideration.

Major concerns:

1. Data presentation and irregularities in data normalization. Throughout the manuscript, data is presented as raw data with individual measured replicates on the left with corresponding normalized data on the right. Individual data points are appreciated. The normalization isn't straightforward and may invite criticism. The error bars do not always translate which may indicate the error is being lost in the normalization. The mean changes based on the normalization. For example: How is SpCas9-HF1 gEJc5 XRCC4—nearly 5 fold higher in the right graph when it appears 10-30% higher in the left graph? From what I gather, this may come from multiple normalization steps - e.g. (A16T-NU7441/A16T-DMSO)/(20-NU7441/20-DMSO). This should be clarified. Consider other means of presentation or normalization.

2. Negative controls to indicate reporter activity in the absence of DSB. Many reporters have some baseline 'leakiness'. This could be indicated on the graph (e.g. a dotted line). This would be more important in the dose limiting experiments.
3. Statistics: Looking at Figure 1F, is gC2 significant? Was this a simple t-test or were statistics adjusted for multiple comparisons? What happened to the error bars between the left and right graphs?
4. Conclusions based on HEK293 and mES clonal cell lines. The conclusions of this manuscript were derived from these two cell lines. This should be made clear in any conclusions derived. Clone-to-clone variation was not noted in the manuscript. This could mask trends or create them. For example, why in Figure 3d would we expect differences in HDR efficiency between DMSO treated, WT, and XRCC4++? These can vary wildly in their HDR efficiency and are treated as a denominator for normalization between left and right graphs
5. Figure 5 and resulting conclusions are not definitive. Considering the previously published literature (e.g Clarke et al, ref 18) it seems that transcriptionally driven Cas9 dissociation would have the same impact as reduced affinity. As the manuscript shows, this was not observed. The reasons could be 1) as described by the authors or 2) the method did not achieve sufficient transcriptional blocking to measure the intended affect. Gene silencing was variable but mostly very weak, typically below 50% with the exception of gGw5. Suggestions include revising the discussion on this finding OR performing lentiviral transduction to generate persistent silencing.

Minor considerations for the authors:

1. What is the frame of the I-SceI GFP reporter? As noted in this manuscript and elsewhere, DNA repair products of CRISPR are not random and therefore 1/3 of products wouldn't necessarily lead to GFP restoration. Also editing was several fold higher in CRISPR editing than I-SceI which may affect interpretation here.
2. From a presentation standpoint, having Figure 1E and 2F together would make sense.
3. Figure 3A, there is a lot of data crossing the break making it difficult to see. Consider log plot?

Manuscript number: GBIO-D-21-01216

Response to the Reviewer Comments

As mentioned in Cover Letter, we sincerely thank the Editors and Reviewers for their detailed and constructive comments and suggestions. Our point-to-point response to the reviewer comments is shown in *italics/blue* text below. Texts are marked in **red** in the manuscript that we have changed and added in response to the reviewer comments.

Reviewer #1

In the manuscript "Target interaction of Cas9-sgRNA influences DNA double strand break repair pathway choices in CRISPR/Cas9 genome editing", Liu et al. designed several assays to study the influence of Cas9-sgRNA post-cleavage residence on the choice of DSB repair pathways. The authors found that c-NHEJ is not even engaged at tested sites and blocking the core factors of c-NHEJ would change the mutagenic spectrum of Cas9 repair outcomes. Moreover, the authors found that DNA replication but not transcription can efficiently remove resident Cas9-sgRNA to influence DNA repair choice. The findings are interesting and help gain deep insight into the genome editing field. However, the manuscript is immature and needs to be well-controlled to arrive the final conclusions.

*We thank the reviewer for spending time reviewing our work and providing the constructive suggestions. To address the concerns raised by the reviewer, we have performed new experiments and additional data analysis, added new figures (e.g., new **Fig 2e,f**, new **Fig 4**, new **Fig S2b-e** and new **Fig S3a-c**), redrew some other figures, and modified some contents and claims accordingly in the revised manuscript, as suggested by the reviewer. We hope that the reviewer could find this revised manuscript more solid and better controlled.*

1. In the Introduction, the alternative NHEJ should be alternative end joining and a-EJ in short as in the ref 4. "Target interaction" is not clear, the authors may consider a better one instead.

Although the term "a-NHEJ" can also be found in the literatures (Deriano and Roth, 2013; Rai et al., 2019; Seol et al., 2018), we agree with the reviewer that "a-EJ" is more commonly used. We have thus changed "a-NHEJ" to "a-EJ" in the manuscript.

In the last version of the manuscript, while our data generally supported that target interaction of Cas9-sgRNA influences DSB repair pathway choices in CRISPR/Cas9 genome editing, we did not directly determine whether mismatched or truncated sgRNAs weaken Cas9-sgRNA target interaction, limit target residence or facilitate dissociation of Cas9-sgRNA from cleaved DNA. We thus used the term "target interaction" to be less specific. However, this may appear vague and unclear to the

reviewer. Possibly because of this, both Reviewer #1 and 2 suggested that we should perform *in vitro* binding assays and directly analyze target residence of Cas9 with mismatched or truncated sgRNAs. We did the experiments as suggested and found that mismatched or truncated sgRNAs indeed reduced the target binding and residence of Cas9-sgRNA and facilitated target dissociation (new **Fig 4**). Thus, to be more specific and direct, we have changed “target interaction” to “target residence” accordingly in the revised manuscript. We thank the reviewer for the suggestion.

2. In Fig. 1a and Fig. 1d, the tandem Cas9 target sites are distributed in a region around the I-SceI cut site. The authors need to swap the order of Cas9 target sites to re-analyze the GFP percentages to avoid potential position effects on these sites, e.g. placing gHRc4 and gHRc5 in the middle and so on. The authors may provide a better schematics to show the orientations and distances of these tandem Cas9 target sites.

*We agree that there are potential position effects on Cas9 target sites in analysis of DSB repair pathway choice in **Fig 1a** and **1d**. To exclude such position effects, two approaches could be used.*

*First, we could directly compare the effects of strong and weak Cas9 target binding or residence on DSB repair pathway choice in repair of Cas9-induced DSBs at the same site in the same context as did in **Fig 3** (i.e., 20-nt sgRNA vs. mismatched or truncated sgRNAs, and SpCas9 vs. eSpCas9, SpCas9-HF1 or xCas9-3.7 at the same given site). The different effects observed between strong and weak Cas9 target binding or residence at the given site would be position-independent. In other words, potential position effects on Cas9 target sites in analysis of DSB repair pathway choices would be avoided. In addition, we performed *in vitro* binding assays as suggested by the reviewer and found that mismatched and truncated sgRNA indeed reduced target binding and target residence of Cas9-sgRNA (new **Fig 4**). Therefore, in the revised manuscript, we did not draw this conclusion that “target residence influences DNA double strand break repair pathway choices in CRISPR/Cas9 genome editing” until we excluded position effects in new **Fig 4**.*

Secondly, we need to create an identical context for different Cas9 target sites in target DNA in the genome and then compared DSB repair pathway choice in repair of Cas9-induced DSBs between these different sites. One solution is to swap the order of Cas9 target sites as proposed by the reviewer. However, the swap also changes the flanking sequences for a given site. Several recent studies have indicated that the flanking sequences are important for Cas9 target binding and residence (Aldag et al., 2021; Zhang et al., 2019, 2020; Zhu et al., 2019). In addition, as discussed in this manuscript, two published review articles and references therein (Feng et al., 2021; Xue and Greene, 2021), post-cleavage Cas9 target residence at a given position is controlled by several factors including target sequence, adjacent sequences around the target, local transcription, DNA replication and chromatin remodeling, etc. Therefore, besides target sequence, changing the genomic position of a Cas9 target

could also alter post-cleavage Cas9 target residence and thus DSB repair pathway choices. However, whether DSB repair pathway choices are altered or not at a given Cas9 target site such as gHR4 or gHR5 after swapping the order of Cas9 target sites in the reporter, it does not tell us whether or how Cas9 target residence affects DSB repair pathway choices. We still need to take the first approach described above and compared the effects of strong and weak Cas9 target binding or residence at a given site in order to determine the effect of Cas9 target residence on DSB repair pathway choices. Nevertheless, in a different study, we inserted a 900-bp sequence upstream of the HDR reporter to slightly shift position of the HDR reporter along with the Cas9 target sites including the gHR4 and gHR5 sites downwards in the genome (**ResFig 1a**). We found that DNA-PKcs inhibition by NU7441 did not alter the frequency of Cas9-induced GFP⁺ cells at the gHR4 and gHR5 sites (**ResFig 1b**), but moderately increased the frequency of Cas9-induced GFP⁺ cells at gHR3 site (**ResFig 1b**).

ResFig 1. The position effect on c-NHEJ engagement. (a) Knock-in of a 900-bp sequence upstream of the HDR reporter in HDR reporter mESC to generate PEO-HDR reporter cells. (b) The effect of NU7441 on HDR in cells containing either the HDR reporter or the PEO-HDR reporter.

As suggested by the reviewer, we also added new schematics in Supplementary Fig S1a and S2a to show the orientations and distances of the tandem Cas9 target sites, which could not be included in Fig. 1a and 1d due to space limitations.

3. In Fig. S1, the authors should employ southern blot or western blot to verify these knock out strains. Moreover, the mESCs have two alleles and the sequences of both alleles should be showed.

As suggested by the reviewer, we have performed Western blot to confirm the absence

of Ku80 or DNA-PKcs proteins in selected KO clones (new **Fig S1b,c**). Targeted deletion including homozygous deletion of two alleles in KO clones was first verified by Sanger sequencing of targeted PCR amplicon. The sequences of both alleles were shown for wild-type and homozygous out-of-frame deletion for KO clones used in this study in **Fig S1b** and **S1c**. Paired Cas9-sgRNA strategy we previously developed was used to generate KO cells in this study (Guo et al., 2018), and the procedure was added into the Methods and Materials in the revised manuscript.

4. It's hard to distinguish the repair products of c-NHEJ and a-EJ through the GFP truncation assay in Fig. 1d and 1e. For example, 1-bp insertions are routinely found for NHEJ-mediated Cas9 outcomes and may switch between out-of-frame and in-frame (this point should be explained and discussed). To better distinguish these by-products, the authors should perform targeted sequencing and then micro-homology analysis to confirm their findings.

In the NHEJ reporter used in this study, there is a 34-bp sequence between “Koz-ATG” and ATG for EGFP. In repair of Cas9-induced DSBs at a site between “Koz-ATG” and the ATG-GFP coding region, both c-NHEJ and a-EJ can generate indels at the repair junction. In general, net addition of “3n+2” bp or net loss of “3n-1” bp in indels can change the 34-bp frame-shift to in-frame, leading to production of GFP⁺ cells. Thus, as 1-bp templated insertion (TIs) does not correct the 34-bp frame-shift, Cas9-induced GFP⁺ cells measured by flow cytometry should not contain this type of indels. We and others have found that 1-bp TIs are frequent in Cas9-induced NHEJ at some sites, but not all sites (Allen et al., 2019; Brinkman et al., 2018; Chakrabarti et al., 2019; Guo et al., 2018; Lemos et al., 2018; van Overbeek et al., 2016; Shen et al., 2018; Shou et al., 2018). The concern raised by the reviewer is that exclusion of these frequent 1-bp TIs may cause a favored or disfavored bias in our analysis of c-NHEJ engagement.

*We thus performed targeted sequencing and junction analysis on the c-NHEJ-independent site gEJw7 and the c-NHEJ-dependent site gEJc5 from **Fig. 1e**. Consistent with FACS-based NHEJ reporter assays, DNA-PKcs inhibition by NU7441 has no effect on the editing efficiency at the gEJw7 site, but reduces the editing efficiency at the gEJc5 site (new **Fig S2b**). About 25% of total NHEJ products were 1-bp TIs at gEJw7 but less than 0.5% at gEJc5 (new **Fig S2e**). NU7441 reduced the level of TIs at both sites (new **Fig S2e**). However, the patterns of deletion length and the MH usage were altered slightly at the gEJw7 site but strongly at the gEJc5 site by DNA-PKcs inhibition (new **Fig S2c,d**). Although NHEJ products with TIs were not picked up in FACS-based reporter assays, it appears that the rest of products detected by the FACS-based assays could reflect the change in c-NHEJ engagement by inactivation of c-NHEJ.*

It is important to note that we cannot distinguish a-EJ from c-NHEJ by individual repair products alone. Compared to c-NHEJ, there remains confusion in defining

a-EJ in the field. Early this year, Ramsden et al published a review in Nat Rev Mol Cell Biol stating “Unlike other DSB repair pathways, a-EJ has been difficult to define” and “The large number of alternative names for a-EJ reflects in part a lack of clarity in how best to define this pathway.”(Ramsden et al., 2022). They pointed out that “early work made it clear that an end-joining pathway exists that does not require Ku, XRCC4 or LIG4, thus being an ‘alternative’ to NHEJ — hence, a-EJ.” It is still useful and less confusing to define c-NHEJ and a-EJ by participating repair factors or lack of these factors, not by their products (Ramsden et al., 2022). For example, as both c-NHEJ and a-EJ can produce accurate end joining despite with different frequencies, it is difficult to tell whether an accurate repair product is generated by c-NHEJ or a-EJ without knowing the status of c-NHEJ factors. As regards 1-bp TIs, which are frequent in Cas9-induced NHEJ at some sites (not all sites), we also cannot assign them into either c-NHEJ or a-EJ. In contrast, by the original definition of a-EJ, while Cas9-induced NHEJ events, whether accurate or mutagenic, can be a mixture of c-NHEJ and a-EJ products in wild-type cells, they are certainly a-EJ products in cells lacking either one of core c-NHEJ factors (e.g., DNA-PKcs^{-/-}, Ku80^{-/-} or Xrcc4^{-/-} cells).

5. For the targeted sequencing data at mCola1 and mRosa26 sites, the analysis of micro-homology usage should be performed to get more information about the NHEJ or a-EJ products.

*We thank the reviewer for the suggestion. We have analyzed the MH usage in NHEJ products at the 4 sites of mCola1 and the 4 sites of mRosa26. At the gC1, gC4, gR1, gR2, gR3 and gR4 sites, NU7441 has little or moderate effect on patterns of MH usage. At the gC2 and gC3 sites, NU7441 significantly increased the usage of MH in NHEJ. These data were provided in new **Fig S3b**, and their description and interpretation were included in the revised manuscript.*

6. The inhibition of NHEJ leads to the change of both a-EJ and HR. The authors should check the change of both a-EJ and HR in Fig. 2b and 2d. And more repeats should be performed to verify the significance in Fig. 2c. Moreover, the numbers of cells detected should be indicated in the panels.

*In **Fig 2b** and **Fig 2d** (new **Fig 2b,c**), we used the NHEJ reporter to analyze the change of NHEJ after treatment of NU7441. However, the NHEJ reporter could not be used for simultaneous analysis of the HR change after treatment of NU7441. We have thus used the HR reporter to analyze HR induced by different concentration of Cas9-sgRNAs and determine the HR change by treatment of NU7441 in the revised manuscript (new **Fig 2e,f**). We found that reducing the concentration of Cas9-sgRNA transfected did not alter the state of c-NHEJ engagement at gHRc4 site, further indicating lack of c-NHEJ engagement at this site. In contrast, this concentration reduction enhanced NU7441-mediated stimulation of HDR at the gHRc2 site (new **Fig 2e,f**). This suggests that Cas9 recleavage of accurate NHEJ enrich both a-EJ*

products and HDR products. We have added this finding into the revised manuscript.

We also performed more repeats to **Fig 2c** (new **Fig 2b**) (five independent experiments in total, each in triplicates) as suggested by the reviewer. The change for the treatment with $0.0001+0.0001\mu\text{g}$ was found to be significant. We redrew the panels in **Fig 2b,c** (new **Fig 2b**). Because of 5 independent experiments, each in triplicates that generated too many raw numbers, listing these total numbers of cells analyzed by FACS and GFP⁺ cells detected in the panels of **Fig 2** would make the panels crowded. We have thus noted these numbers in average in the legends of **Fig 2**. The numbers in 2 representative independent experiments among 5, each in triplicates for **Fig 2c** (new **Fig 2b**) are listed in **ResFig 2** for reviewer's evaluation.

Exprt#1		DMSO			NU7441		
Triplicates		1	2	3	1	2	3
Control	Total	334671	400943	368137	148772	180273	141083
	GFP+	4	3	3	4	1	6
0.25ug	Total	465712	475990	444001	219111	203782	198473
	GFP+	42369	42839	42457	10671	11792	11890
0.1ug	Total	443063	414408	480319	227923	181328	182415
	GFP+	40098	40125	40156	13486	12085	11779
0.01ug	Total	479300	441374	424788	215669	190370	191586
	GFP+	14389	13964	14789	6638	6583	6553
0.001ug	Total	429658	416845	432394	254718	217388	227694
	GFP+	1060	1042	1041	792	786	780
0.0001ug	Total	743299	676885	707499	298832	291040	293372
	GFP+	127	127	139	105	98	126

Exprt#2		DMSO			NU7441		
Triplicates		1	2	3	1	2	3
Control	Total	395347	334361	274308	60586	56443	66940
	GFP+	5	15	9	8	2	2
0.25ug	Total	208350	312842	302651	65804	84186	54176
	GFP+	19855	28163	28126	4075	4332	4169
0.1ug	Total	198618	203082	236871	45138	47394	51156
	GFP+	17844	18830	18253	2550	2981	2828
0.01ug	Total	157647	143216	79124	36630	27517	25602
	GFP+	6848	6482	4342	1472	1423	1451
0.001ug	Total	148298	122626	141795	61216	61212	60602
	GFP+	413	279	355	201	229	215
0.0001ug	Total	1226753	1721493	1265483	466440	434589	435261
	GFP+	250	312	280	109	130	191

ResFig 2. The numbers of cells analyzed in 2 representative independent experiments among 5 for Fig 2c (new Fig. 2b).

7. In Fig. 2f, the PK inhibitor and knock our strains showed different phenotypes, which requires explanation.

In **Fig 2f** (new **Fig 2d**), NU7441 elicited weaker manifestation of c-NHEJ engagement than XRCC4 KO. This difference was also observed in **Fig 1b,c,e**. We do not know the

*exact reason behind this difference but one explanation was considered. Core NHEJ factors such as DNA-PKcs, Ku70/Ku80, XRCC4 and DNA ligase 4 act in different steps and perform different tasks in c-NHEJ. Disruption at different c-NHEJ steps could yield different substrates for repair by alternative repair pathways such as HDR and a-EJ, causing different changes in the efficiency and signature of alternative repair pathways. This might explain why inactivation of DNA-PKcs, Ku80 and XRCC4 stimulated Cas9-induced HDR to different extents (See new **Fig. 1b,c**). Even at the gHRc4 and gHRc5 sites, inactivation of XRCC4 modestly stimulated HDR but inactivation of DNA-PKcs or Ku80 did not. In addition, some core NHEJ factors may also participate in a-EJ when c-NHEJ is inactivated by lack of one core factor; however, the compensation may differ in a-EJ for different core factors that are lost. It appears that the lost function of DNA-PKcs in a-EJ can be more readily compensated than the lost function of XRCC4. Thus, DNA-PKcs inactivation and XRCC4 inactivation presented different manifestation of c-NHEJ engagement. We have incorporated this explanation into the revised manuscript. The other explanation is that chemical inhibition may cause different phenotypes as compared to genetic deletion, in particular in the case of DNA-PKcs (Menolfi and Zha, 2020). However, in our study, chemical inhibition of DNA-PKcs by NU7441 and KO of DNA-PKcs display similar effect on stimulation of Cas9-induced HR at various sites (See NU7441 treatment in new **Fig 1b** vs DNA-PKcs deletion in new **Fig 1c**).*

8. The authors may perform competition assays to confirm that the mutations reduce target interaction of Cas9-sgRNA in Fig. 3. Moreover, the right panels are over-processed and hard to catch the RELATIVE fold-change. Original fold changes need to be showed for each sample.

*We thank the reviewer for the suggestion. In previous version of the manuscript, we provided no direct evidence to confirm that the mutations of sgRNA reduce target interaction of Cas9-sgRNA in **Fig 3**, although many studies have previously demonstrated that mismatched or truncated sgRNAs reduce target interaction or residence time of Cas9-sgRNA and promote target dissociation of Cas9-sgRNA (Jiang and Doudna, 2017; Kim et al., 2019; references therein). In this revision, we performed in vitro Cas9-sgRNA cleavage assays, target residence assays and competition assays as the reviewer suggested and found that mismatched or truncated sgRNAs in **Fig 3** indeed reduced target residence of Cas9-sgRNA (new **Fig 4**). Compared to sgRNAs with perfectly matched 20-nt spacer (wild-type sgRNA), mismatched or truncated sgRNAs caused a slower rate of DNA cleavage by Cas9 although all target DNA were completely cleaved after 24-h reaction (new **Fig 4a**). After DNA cleavage, more cleaved DNA were released from the Cas9-sgRNA-DNA ternary complex with mismatched or truncated sgRNAs as compared to perfectly matched sgRNA (new **Fig 4b**). This suggests that post-cleavage Cas9-sgRNA target residence is reduced by mismatched or truncated sgRNAs. The target binding of dCas9-sgRNA was also less efficient when sgRNAs were mismatched or truncated (new **Fig 4c**). After target binding of dCas9-sgRNA or dCas9-sgRNA mutants, the*

competition by SpCas9-sgRNA (wild-type) revealed that target DNA in the dCas9-sgRNA-DNA ternary complex was released more from dCas9-sgRNA mutants than from wild-type dCas9-sgRNA (new Fig 4d). Together, these data confirmed that the mutations of sgRNA reduced target interaction and target residence of Cas9-sgRNA. We have added these results into the revised manuscript.

We have also changed the relative fold-change panels to the original fold-change panels in Fig 3 as suggested by the reviewer.

9. Different sites are used for Fig 3a-c vs 3d-e. The changes of HR and a-EJ after NHEJ inhibition should be performed at the same sites to get the full pictures of repair choice change after NHEJ inhibition.

As mentioned above in response to Q6, the GFP-based DSB repair reporters were originally designed to measure only HDR or NHEJ, not simultaneously both by flow cytometry. That's the reason why we used two types of reporters, one for HDR and the other for NHEJ, in this study. Using either the NHEJ reporter or the HDR reporter, we were still able to analyze the effect of c-NHEJ inactivation by chemical inhibition or genetic KO on HDR or NHEJ induced by Cas9-sgRNA and its variants at the same target sites and found that c-NHEJ engagement varies between sites. Reduced Cas9-sgRNA target interaction or residence promoted c-NHEJ engagement at the same Cas9 target sites. However, using the HDR reporter, we later sequenced the junction of repair products from the HDR reporter assays and distinguished NHEJ products from HDR products as shown in new Fig 7c and 7d. In this way, we could measure HDR and NHEJ simultaneously with the HDR reporter and determine the HDR bias at the same sites as shown in new Fig 7d. But this approach requires targeted deep sequencing of repair junction in the HDR reporter and is limited and not convenient to use.

10. The mutation sites C2A and T15A in Fig. 3 can also be considered as off-targets for gEJc5, however, the cutting efficiencies were decreased for these sites after XRCC4 knock-out, not very consistent with the findings in Fig. 4. The authors may consider to reclaim the conclusions that NHEJ inhibition has more impact on on-target cleavage than off-target cleavage.

We agree that the mutation sites C2A and T15A in Fig 3a can be considered as off-targets for gEJc5 and the absolute editing efficiencies were decreased for these sites in XRCC4-null cells. In fact, XRCC4 deletion reduced the off-target editing efficiencies from 5% to 3% for C2A (reduction by 40%) and from 2.4% to 1.8% for T15A (reduction by 25%). However, XRCC4 deletion decreased the on-target editing efficiency of gEJc5 from 15% to 3% (reduction by 75%). In order to compare the extent of the change in off-target effects between XRCC4-wt cells and XRCC4-null cells, we need to normalize the absolute off-target editing efficiencies of gEJc5 with its on-target editing efficiency. Because the on-target editing efficiency of gEJc5 was

decreased proportionally more than off-target editing efficiency due to XRCC4 deletion, the ratios of absolute off-target editing efficiencies to absolute on-target efficiency were higher in XRCC4-null cells than in XRCC4-wt cells. This indicated exacerbation of off-target effect by XRCC4 deletion. We applied similar calculation on the change of off-target effect by NU7441 in **Fig 4** (new **Fig 5** in the revised manuscript) and found that c-NHEJ inactivation exacerbated off-target effects. This observation was consistent with the results in **Fig 3** and **Fig 4** (new **Fig 5** in the revised manuscript) and indicated that c-NHEJ inactivation could cause exacerbation of off-target effects as stated in the revised manuscript.

However, as pointed out by the reviewer, XRCC4 deletion reduced “off-target” editing efficiencies for gEJc5, whereas NU7441 did not in **Fig 4** (new **Fig 5** in the revised manuscript). This apparent difference could be explained. First, as we described above in response to Q9, NU7441 treatment and XRCC4 knock-out could have different phenotypes. When we only compared the effect of NU7441 in **Fig 3a** and **Fig 4** (new **Fig 5**), the effect of NU7441 on “off-target” editing efficiencies for gEJc5 (C2A and T15A) in **Fig 3a** was not much different from the effect of NU7441 on off-target editing efficiencies in **Fig 4** (new **Fig 5**).

Secondly, in **Fig 1e**, we showed that XRCC4 deletion significantly reduced on-target editing efficiencies of gEJc5, gEJw4, gEJw5 and gEJw6, indicating c-NHEJ engagement at these sites. The data in new **Fig 2b-d** suggested that limiting Cas9 target re-cleavage could elicit the stimulatory effect of XRCC4 deletion on Cas9-induced indels at these sites, but to different extents at different sites. Because the stimulatory effect was much weaker at the gEJc5 and gEJw5 sites than at the gEJw4 and gEJw6 sites, limiting Cas9 target re-cleavage could only allow this simulation to reverse the frequencies of Cas9-induced indels at the gEJw4 and gEJw6 sites in XRCC4-null cells to a level higher than that in XRCC4-wt cells, but not allow to fully restore at the gEJc5 and gEJw5 sites. It is possible that single-nucleotide mismatch (e.g., C2A and T15A) in gEJc5 could increase c-NHEJ engagement but only slightly lower Cas9 target re-cleavage at the gEJc5 site. Consequently, the stimulatory effect of XRCC4 deletion on Cas9-induced indels was modest and only partially restore indel-based editing efficiencies at these “off-target” sites for gEJc5 in **Fig 3a**. In contrast, in **Fig 4** (new **Fig 5**), NU7441 only slightly reduced on-target editing efficiencies. Moreover, more mismatches in off-target sites would greatly limit target re-cleavage at these off-target sites; thus, NU7441 could block c-NHEJ, channeling DSB repair towards error-prone a-EJ from c-NHEJ that is largely accurate in repair of Cas9-induced DSBs. As a result, absolute efficiencies of off-target editing were increased. Because the ratios of absolute off-target editing efficiencies to absolute on-target efficiency were elevated by either XRCC4 deletion in **Fig 3a** or NU7441 in **Fig 4** (new **Fig 5**), this did not contradict the idea that c-NHEJ inactivation could cause exacerbation of off-target effects as stated in the revised manuscript.

11. In Fig.5, the blockage of transcription by dCas9 was not so strong, the authors may consider remove the pPGK promoter to suppress GFP transcription and perform the analysis in parallel with pPGK proficient cells.

After we found lack of c-NHEJ engagement in repair Cas9-induced DSBs at some sites in new Fig. 1-4, we tried to determine the underlying mechanisms. In Fig 5 (new Fig 6), we tried to address the question whether transcription-mediated target dissociation prevents c-NHEJ engagement. But we did not describe our question and rationale well for this part in the previous version of the manuscript and caused some confusion. We have made clarification in the revised manuscript. As explained, in order to test the hypothesis that the collision with local transcription machinery may prevent c-NHEJ engagement in repair of Cas9-induced DSBs at some sites, we needed: 1) active transcription that could collide with Cas9-induced DSBs and dislodge Cas9-sgRNA from cleaved DNA; 2) comparison of c-NHEJ engagement in repair of Cas9-induced DSBs at the same sites in the presence and absence of transcription-mediated dissociation of Cas9-sgRNA from cleaved DNA. Removal of the PGK promoter could reduce transcription and thereby the probability of a collision between local transcription and Cas9-sgRNA at the cleaved site, preventing us from doing the above analysis. In addition, we agree with the reviewer, the stronger the transcription blockage, the better for the assay. However, it remains a challenge to achieve a strong and satisfactory dCas9-based transcription blockage in mammalian cells (Dominguez et al., 2016). The blockage is about 50% in our assays using either dSaCas9-gSaGw1 or dSaCas9-gSaGw2 to block the GFP transcription. This transcription blockage was not as effective as in prokaryotic cells, but as strong as those previously reported for GFP gene silencing in mammalian cells (Qi et al., 2013).

12. Cas9-sgRNA are reported to stay at the sgRNA-bound ends, so the model in Fig. 6h should be re-drawn and DNA replication comes from the other ends in Fig. 6b should be performed as a control to draw the conclusion that DNA replication machinery remove resident Cas9-sgRNA to affect DNA repair choice.

There might be some misunderstanding here. It is widely reported that SpCas9-sgRNA are tightly bound with both PAM-proximal and PAM-distal ends of cleaved DNA at the targets after DNA cleavage (Richardson et al., 2016; Sternberg et al., 2014; Zhu et al., 2019). While Richardson et al revealed earlier release of the PAM-distal non-target DNA strand from the SpCas9-sgRNA-DNA ternary complex, they also found that Cas9-sgRNA remains tightly bound with the other three strands and Cas9-induced DSBs are exposed by symmetric dissociation of Cas9 from both DSB ends due to this persistent binding (Richardson et al., 2016). Therefore, DNA replication from either direction would dissociate SpCas9-sgRNA from target DNA, although some mechanistic details in this replication-coupled SpCas9 dissociation from its target might differ between two directions.

Differently, many studies have demonstrated that Cas12a (LbCas12a and AsCas12a) stays on at PAM-proximal ends but rapidly releases PAM-distal ends after DNA cleavage (Singh et al., 2018; Strohkendl et al., 2018; Swarts and Jinek, 2019; Swarts et al., 2017). It is not clear whether the residence of SaCas9 at cleaved DNA is similar to Cas12a or SpCas9. One study recently provided some evidence to indicate that SaCas9 stays bound only with PAM-proximal ends, not with PAM-distal ends after DNA cleavage (Zhang et al., 2020), but we have not seen additional research articles with similar findings thus far.

13. The sequence lengths in Table S1 are not consistent with the PCR band size in Fig. 6f. In mESCs, LT is not required to form TF1-TF2 bands. More explanations are required.

In Table S1, the sequences in gray shade were not detected and thus denoted as deleted sequences in the table legends although these sequences were initially expected in the palindromic sister chromatid NHEJ products. After the deleted sequences are removed from length calculation, the sequence length we detected in Table S1 is ~300 bp and consistent with the PCR band size in Fig 6f (new Fig 7f in the revised manuscript).

It is well established that SV40 LT does not work for replication of DNA harboring an SV40 origin in mESC due to the host species specificity (Murakami et al., 1986). Therefore, we did not use SV40 LT to drive DNA replication in mESC. However, due to fast cycling time of mESC (~12 hours per cell cycle), these cells are more susceptible for SpCas9 residing at the cleaved DNA to collide with local DNA replication, thus generating three-ended DSBs and forming TF1-TF2 bands after palindromic fusion of sister chromatid ends.

Reviewer #2:

In this manuscript entitled "Target interaction of Cas9-sgRNA influences DNA double strand break repair pathway choices in CRISPR/Cas9 genome editing" Liu et al., aim to study how the CRISPR/Cas9-sgRNA combination can affect the repair choice of DNA double strand breaks. Previously the authors developed various DSB reporter constructs for both NHEJ and HDR as tests for I-SceI-induced DSBs. In this study these constructs are used to study CRISPR/Cas9-induced DSBs.

The hypothesis that the affinity of Cas9-sgRNA to its break site may influence the repair pathway usage is interesting to the field, since it could impact the design of genome editing experiments.

The manuscript is clearly written and the authors discuss various KO models of c-NHEJ and test multiple Cas9 and sgRNA modifications. However, my major concern of this study is that the experimental design is in my opinion not suitable to investigate the question the authors want to address. Several firm claims are made in

this work that are not proven e.g. c-NHEJ is not engaged at some CRISPR-sgRNA target sites; different persistence of Cas9-sgRNA target interaction exposes Cas9-induced DSBs with different end configurations for specific repair pathways and presence of Cas9 recleavage due to the accurate repair by c-NHEJ.

Thanks for the comments but we respectfully disagree with the reviewer's views that "the experimental design is in my opinion not suitable to investigate the question the authors want to address". In this study, using the c-NHEJ inhibitor NU7441 and genetic inactivation of c-NHEJ, we first found that c-NHEJ involvement varied at different target sites in repair of Cas9-induced DSBs (Fig 1). It is however unexpected that the frequency of Cas9-induced indels was inhibited at many Cas9 target sites by inactivation of c-NHEJ (Fig 1e-g). We previously found that c-NHEJ repair of I-SceI- and Cas9-induced DSBs is inherently accurate with the directly ligatable ends of these DSBs (Guo et al., 2018; Xie et al., 2009). We and others also found that inactivation of c-NHEJ by chemical inhibition or genetic modification stimulated I-SceI-induced indels by channeling DSBs that are supposedly repaired by c-NHEJ towards error-prone a-EJ for repair (new Fig S4b,c) (Bindra et al., 2013; Feng et al., 2017). We thus hypothesized that Cas9-mediated re-cleavage of accurate NHEJ products amplify the mutagenicity of c-NHEJ in CRISPR/Cas9 genome editing and tested the hypothesis by limiting Cas9 re-cleavage and analyzing the effect on genome editing. Our data suggested that Cas9 re-cleavage of accurate NHEJ products occur at the sites where c-NHEJ is engaged, not at the sites where no c-NHEJ is engaged (Fig 2). We then asked why c-NHEJ involvement varied at different target sites in repair of Cas9-induced DSBs. We used mismatched sgRNAs, truncated sgRNAs, Cas9 variants and off-target sites to reduce Cas9 target interaction and identified the effect of this reduction on DSB repair pathway choices in repair of Cas9-induced DSBs (Fig 3). In vitro binding assays indicated that mismatched and truncated sgRNA indeed reduced target binding and target residence of Cas9-sgRNA (new Fig 4). These data together suggested that reduction of Cas9 target residence increased c-NHEJ involvement. Due to c-NHEJ involvement favored at off-target sites where Cas9 target residence is short or weak, inactivation of c-NHEJ by chemical inhibition or genetic modification increased the ratio of off-target editing to on-target editing, exacerbating off-target effect (new Fig 5). Therefore, as an important strategy for enhancing HDR-mediated genome editing, inactivation of c-NHEJ needs further improvement in order to avoid exacerbation of off-target effects. To further address why c-NHEJ is little engaged at some sites in repair of Cas9-induced DSBs, we proposed and tested the transcription/replication collision mechanisms that could explain the biased disengagement of c-NHEJ at these Cas9-sgRNA target sites (new Fig 6 and Fig 7). Our data suggested that replication collision may dislodge Cas9-sgRNA from cleaved DNA, thus generating three-ended DSBs and resulting in palindromic fusion of sister chromatids. Together, we concluded that Cas9-sgRNA target residence modulates DSB repair pathway choices likely through varying dissociation of Cas9-sgRNA from cleaved DNA. In summary, this work demonstrated the influence of Cas9 target residence on DSB repair pathway

choices and provided unique insights into the mechanisms underlying the heterogeneity of repair products in CRISPR/Cas9 genome editing.

In response to more specific comments by the reviewer, we have provided evidence in this revised manuscript, along with published work from us and others, to support: 1) c-NHEJ is little engaged at some Cas9 target sites; 2) Different persistence of Cas9-sgRNA target binding influence DSB repair pathway choices at least in part by exposing Cas9-induced DSBs with different end configurations; 3) Presence of Cas9 recleavage due to accurate repair by c-NHEJ.

*First, **Fig 1** demonstrated that chemical inhibition or genetic inactivation of c-NHEJ has little effect on Cas9-induced c-NHEJ and HDR at some Cas9 target sites, e.g., the sites targeted by gHRc4, gHRc5, gEJw3, gEJw7, gC1, gC4, gR1 and gR4. **Fig 2** further confirmed no change on c-NHEJ by chemical inhibition or genetic inactivation of c-NHEJ at the gEJw7 site even after excluding the effect of Cas9 target re-cleavage. Junction analysis of NHEJ products at some of these sites revealed that mutation signatures including the pattern of deletion length and the MH usage were little affected by inactivation of c-NHEJ (**Fig S2c,d** and **Fig S3a,b**). In contrast, significant changes were observed at sites with c-NHEJ engagement (**Fig S2c,d** and **Fig S3a,b**). **Fig 7** indicated replication-coupled Cas9 dissociation from cleaved DNA may help explain lack of c-NHEJ engagement at these sites.*

*Secondly, as commented by the reviewer, we have not directly demonstrated “different persistence of Cas9-sgRNA target interaction exposes Cas9-induced DSBs with different end configurations for specific repair pathways” in this study. This is an issue that we are also aware of but it is difficult to directly resolve. However, many studies have shown that mutations in Cas9 or sgRNA reduce persistent target interaction of Cas9-sgRNA and facilitate target dissociation (See the reviews by Jiang and Doudna, 2017; Kim et al., 2019; and reference therein). We thus used mismatched sgRNAs, truncated sgRNAs and Cas9 variants to reduce Cas9 target interaction or target residence and found that reduction of Cas9 target interaction or target residence increased c-NHEJ involvement (**Fig 3**). In new **Fig 4**, using in vitro binding or competition assays as suggested by the reviewer, we showed that mismatched or truncated sgRNAs indeed reduced the target binding and residence of Cas9-sgRNA. We also analyzed c-NHEJ involvement at off-target sites where Cas9 target interaction or target residence is reduced and found that c-NHEJ is more engaged at off-target editing (new **Fig 5**). Because reduced target interaction or target residence increased the probability of spontaneous dissociation of Cas9-sgRNA from cleaved DNA, these data suggested that two clean ends of Cas9-induced DSBs exposed by spontaneous target dissociation are a favorable substrate for c-NHEJ. This is consistent with previous studies showing that clean ends of DSBs readily engage c-NHEJ (See the review by Bétermier et al., 2014 and references therein). To explain why c-NHEJ is little engaged at some Cas9 target sites, we excluded the effect of transcription collision on lack of c-NHEJ engagement (new **Fig 6**) and proposed*

that Cas9-induced DSBs at these sites are mostly exposed by collision with DNA replication due to persistent target residence at cleaved DNA, generating three-ended DSBs unsuitable for c-NHEJ. Thus, in new **Fig 7a-d**, we initiated local DNA replication to dislodge Cas9-sgRNA and analyzed the effect of c-NHEJ inhibition on HDR of Cas9-induced DSBs exposed. We found that simulation of Cas9-induced HDR by c-NHEJ inhibition was reduced or even abolished by local DNA replication. We also detected inverted duplications at junctions of NHEJ products, as inverted junctions serve as an indication for sister chromatid fusion at the sites of Cas9-induced DSBs and for generation of three-ended DSBs (new **Fig 7e-g**). Cas9-induced sister chromatid fusion has also been reported in recent studies (Kagaya et al., 2020; Umbreit et al., 2020). Generation of inverted duplications was closely associated with local DNA replication (new **Fig 7e-g**). As it is known that end configurations influence DSB repair pathway choices (Chang et al., 2017), these data together supported the model that post-cleavage target residence of Cas9-sgRNA influences DSB repair pathways choices, likely by exposing Cas9-induced DSBs with different end configurations (new **Fig 7h**).

Thirdly, when we think of Cas9 target re-cleavage, we consider: 1) Cas9-sgRNA targets should be reused; 2) The Cas9-sgRNA enzyme complex should be reused. Reuse of Cas9-sgRNA targets requires accurate repair of Cas9-induced DSBs. Possibly because of highly efficient CRISPR/Cas9 genome editing (i.e., a high level of Cas9-induced indels), it was thought that NHEJ repair of Cas9-induced DSBs was mostly mutagenic. Since Cas9-induced DSBs have two clean ligatable ends, this seems contradictory to previous findings that directly ligatable ends can be accurately and efficiently repaired by c-NHEJ. In fact, even before several lines of evidence were available to support that c-NHEJ is inherently accurate in repair of Cas9-induced DSBs, Cas9-mediated recleavage had been proposed (Danner et al., 2017; White et al., 2016). Because accurate NHEJ of Cas9-induced DSBs is indistinguishable from un-cleaved targets, it is however impossible to address this contradiction by directly analyzing accurate NHEJ products in repair of Cas9-induced DSBs. We and others had thus designed several indirect methods to assess the accuracy in NHEJ of Cas9-induced DSBs. First, we had used paired Cas9-sgRNAs to demonstrate that Cas9-induced NHEJ are inherently accurate and the frequency of accurate NHEJ in repair of Cas9-induced DSBs is about 50% in average (Guo et al., 2018). Second, Song et al had used precise insertion of 34-bp double strand oligodeoxynucleotides (ODNs) into Cas9-induced DSBs to determine the efficiency of accurate NHEJ in repair of Cas9-induced DSBs (Song et al., 2021). Based on their data, they stated “the average value of NHEJ accuracy is approximately 75% in maximum in HEK 293T cells”. Recently, Yin et al has used Cas9-induced translocations to demonstrate accurate end joining at translocation junctions and deduce the occurrence of repeated cleavage of re-targetable junctions by Cas9-sgRNA (Yin et al., 2022). These publications were cited in the revised manuscript.

Reuse of Cas9-sgRNA also seems contradictory to the notion that SpCas9 is a

*single-turnover enzyme. However, in the study that defined SpCas9 as a single-turnover enzyme (Sternberg et al., 2014), Sternberg et al stated that “The finding that Cas9:RNA remains tightly bound to both ends of the cleaved DNA, suggests it acts as a single-turnover enzyme.” Therefore, this study did not exclude the possibility that if SpCas9-sgRNA is released from cleaved DNA, it can be reused to target DNA. We should not misinterpret this “single-turnover enzyme” statement as if SpCas9 can only cleave DNA once. Given that Cas9-induced DSBs can be accurately repaired, accurate repair products can be re-cleaved by Cas9-sgRNAs that either target for the first time or are re-used after dissociation from their targets. In cells, some of Cas9-sgRNAs are dislodged by external forces such as DNA replication (in this study), transcription and chromatin remodeling (Clarke et al., 2018; Wang et al., 2020); Cas9-sgRNA dissociated from its targets could serve as a multiple-turnover enzyme (Clarke et al., 2018; Wang et al., 2020). New **Fig 4** in our revised manuscript indicated that SpCas9-sgRNA variants could be spontaneously dissociated from cleaved DNA, providing an opportunity for these SpCas9-sgRNAs to cleave DNA again. In addition, in our manuscript, by reducing the concentration of Cas9-sgRNA in cells by serial dilution for transfection to limit re-cleavage of accurate NHEJ, we found that re-cleavage of accurate NHEJ products by Cas9 could amplify the mutagenicity of c-NHEJ in CRISPR/Cas9 genome editing (new **Fig 2a-d**).*

Other conclusions are not original, as published studies have conducted multiple guide repair analysis with larger sets of guides that already showed the wide variety of repair at the different target sites (van Overbeek, et al. Mol Cell. 2016; Shen, et al.; Nature. 2018; Allen, et al. Nat Biotechnol. 2018; Chakrabarti, et al. Mol Cell. 2019; Chen, et al. Nucleic Acids Res. 2019).

Thanks for the comment. If the focus of this manuscript were to document the significant heterogeneity of repair products at different targets, we agree it would not be new or original. However, this is clearly not the focus of the manuscript. On the contrary, under the background “the wide variety of repair at the different target sites”, this manuscript tried to address why “the wide variety of repair at the different target sites” occurs. The field may contribute several apparently well-recognized factors such as nucleotide composition of target sequences, the chromatin context, cell cycle and cell type to the control of DSB repair pathway choices and thereby the heterogeneity of repair products in CRISPR/Cas9 genome editing; but it appears that Cas9 target residence at cleaved DNA has thus far been ignored as an important contributor. As a unique property in Cas9-induced DNA breakage, Cas9 target residence at cleaved DNA could affect exposure of Cas9-induced DSBs and subsequent repair; and its effect on DSB repair should have been addressed. In this manuscript, we found that Cas9 target residence at cleaved DNA influences c-NHEJ engagement and identified a replication-coupled mechanism that may prevent c-NHEJ engagement at some target sites. This work provided unique insights into the mechanisms underlying the heterogeneity of repair products in CRISPR/Cas9 genome editing and may point a direction of CRISPR/Cas9 engineering by enhancing or

weakening target residence for different applications. This manuscript demonstrated that the influence of Cas9 target residence on DSB repair pathway choices is real and important.

The connection of repair pathway choice and Cas9 dissociation is interesting, but the authors do not show the actual Cas9 variation in dissociation themselves.

*Thanks for the comment. In previous version of the manuscript, we provided no direct evidence for target dissociation of Cas9. This is also a concern of Reviewer #1. As described above in response to Q8 by Reviewer #1, we performed in vitro Cas9-sgRNA cleavage assays, target residence assays and competition assays in this revision and found that sgRNA mismatch or truncation facilitated target dissociation of Cas9-sgRNA and reduced target residence (new **Fig 4**). Compared to sgRNA with perfectly matched 20-nt spacer (wild-type sgRNA), mismatched or truncated sgRNAs caused a slower rate of DNA cleavage by Cas9 although all target DNA were completely cleaved after 24-h reaction (new **Fig 4a**). After DNA cleavage, more cleaved DNA were exposed from the Cas9-sgRNA-DNA ternary complex for mismatched or truncated sgRNAs as compared to perfectly matched sgRNA (new **Fig 4b**). This suggests that post-cleavage Cas9-sgRNA target dissociation is increased by mismatched or truncated sgRNAs. The target binding of dCas9-sgRNA was also less efficient when sgRNAs were mismatched or truncated (new **Fig 4c**). After target binding of dCas9-sgRNA and dCas9-sgRNA mutants, the competition by SpCas9-sgRNA (wild-type) revealed that dCas9-sgRNA mutants were more released from their targets than dCas9-sgRNA (wild-type), exposing target DNA from the dCas9-sgRNA-DNA ternary complex for DNA cleavage by Cas9 (new **Fig 4d**). Together, these data confirmed that mismatched or truncated sgRNAs reduced target residence and facilitated target dissociation of Cas9-sgRNA or dCas9-sgRNA. We have added these results into the revised manuscript.*

Lastly, the paper shows limited mechanistic insights.

*Thanks for the comment. This comment shares some similarity in two previous comments made by the reviewer, i.e., “the experimental design is in my opinion not suitable to investigate the question the authors want to address” and “Other conclusions are not original”. As we mentioned above, Cas9 target residence at cleaved DNA is a unique property in Cas9-induced DNA breakage and would affect target dissociation of Cas9-sgRNA from cleaved DNA. The exposure of Cas9-induced DSBs by different means is expected to have effect on DSB repair pathway choices but this effect has not been addressed. In this manuscript, using the DNA-PKcs inhibitor NU7441 and genetic inactivation of c-NHEJ, we first found that c-NHEJ involvement varied at different target sites in repair of Cas9-induced DSBs even in the same cells (**Fig 1**). It is however unexpected that the frequency of Cas9-induced indels was inhibited at many Cas9 target sites by inactivation of c-NHEJ (**Fig 1e-g**). We previously found that c-NHEJ repair of I-SceI- and Cas9-induced DSBs is inherently*

accurate with the directly ligatable ends of these DSBs (Guo et al., 2018; Xie et al., 2009). We and others also found that inactivation of c-NHEJ by chemical inhibition or genetic modification stimulated I-SceI-induced indels by channeling DSBs that are supposedly repaired by c-NHEJ towards error-prone a-EJ for repair (Fig. S4b,c) (Bindra et al., 2013; Feng et al., 2017). We thus hypothesized that Cas9-mediated re-cleavage of accurate NHEJ products amplifies the mutagenicity of c-NHEJ in CRISPR/Cas9 genome editing and tested the hypothesis by limiting Cas9 re-cleavage and analyzing the effect on genome editing. Our data suggested that Cas9 re-cleavage of accurate NHEJ products occur at the sites where c-NHEJ is engaged, not at the sites where no c-NHEJ is engaged (Fig 2). We then asked why c-NHEJ involvement varied at different target sites in repair of Cas9-induced DSBs. We used mismatched sgRNAs, truncated sgRNAs, Cas9 variants and off-target sites to reduce Cas9 target interaction or target residence and identified the effect of this reduction on DSB repair pathway choices in repair of Cas9-induced DSBs (Fig 3). In vitro binding assays indicated that mismatched and truncated sgRNA indeed reduced target binding and target residence of Cas9-sgRNA (new Fig 4). These data together suggested that reduction of Cas9 target interaction or target residence increased c-NHEJ involvement. Due to c-NHEJ involvement favored at off-target sites where Cas9 target residence is short, inactivation of c-NHEJ by chemical inhibition or genetic modification increased the ratio of off-target editing to on-target editing, exacerbating off-target effect (new Fig 5). Therefore, as an important strategy for enhancing homology-directed repair, inactivation of c-NHEJ needs further improvement in order to avoid exacerbation of off-target effects. To further address why c-NHEJ is little engaged at some sites in repair of Cas9-induced DSBs, we proposed and tested the transcription/replication collision mechanisms that could explain the lack of c-NHEJ engagement at these Cas9-sgRNA target sites (new Fig 6 and Fig 7). Our data suggested that replication collision may dislodge Cas9-sgRNA from cleaved DNA, thus generating three-ended DSBs and resulting in palindromic fusion of sister chromatids. Together, although it is not completely clear about how Cas9-sgRNA target residence is regulated at each site and how this regulation affects DSB repair pathway choices in repair of Cas9-induced DSBs, the manuscript not only provided significant insights into the effect of Cas9 target residence on DSB repair pathway choices but also shed light on the mechanisms underlying the heterogeneity of repair products in CRISPR/Cas9 genome editing.

It is unclear why the authors suggest the three-ended DSB mode since no palindromic fusions were detected. The finding of inverted GFP sequences is not sufficient to support this model. It is also unclear how often the inverted GFP arises during DNA repair. With additional experiments to support its conclusions, this study may be of some interest to a specialized journal.

Thanks for the comment. When we tested the idea that palindromic sister chromatid fusion (SCF) may occur via NHEJ at the sites where local DNA replication collides with Cas9-sgRNA bound with cleaved DNA, it was also surprising to us: 1) PCR with single primer TF1 or TF2 did not generate a clear band for potential Cas9-induced palindromic SCF; 2) Using asymmetric primer pair TF1 and TF2 for PCR, we detected clear PCR bands and found inverted GFP sequence flanking the site of Cas9-induced DNA breakage after DNA sequencing but palindromic sequences appear to be deleted. In a recent work, PCR analysis of palindromic SCF products also showed large deletions and lack of palindromic sequences at the fusion junctions (See **Fig. S2** in Kagaya et al., 2020). In telomere end-to-end fusions that either occurred naturally or were induced by I-SceI or TALEN, the expected palindromic sequences at the repair junction were also absent after PCR (Capper et al., 2007; Liddiard et al., 2016; Lo et al., 2002). These authors attributed this lack of palindromic sequences to asymmetric end resection although no clear evidence was provided. However, it is known that a hairpin structure could cause replication slippage of thermostable DNA polymerases started in the first round of PCR, thus resulting in the deletion of the hairpin in final PCR products (Hommelsheim et al., 2014; Viguera et al., 2001). As a result, it remains a technical challenge in the field to directly detect palindromic sequences by PCR. This may explain why PCR with single primer such as TF1 or TF2 in this manuscript generated no PCR products. It is because the interval palindromic sequences between two inverted TF1 or two inverted TF2 primers could all be part of a hairpin structure (**ResFig 3a**). On the other hand, PCR with asymmetric primer pair TF1 and TF2 deleted the hairpin formed by palindromic GFP sequences but keep some inverted GFP sequences that are not part of the hairpin (**ResFig 3b**).

ResFig 3. The SCF hairpin structure that may cause replication slippage of Taq DNA polymerase in PCR. (a) A perfect hairpin with a single primer TF1. (b) A imperfect hairpin with an asymmetric primer pair TF1 and TF2.

How often does inverted GFP arises from Cas9-induced palindromic SCF? Using indirect analysis, a few studies have suggested that palindromic SCF arises from Cas9-induced DSBs with an extremely low frequency but could lead to chromothripsis (Kagaya et al., 2020; Leibowitz et al., 2021; Umbreit et al., 2020). Of note, these studies did not pay any attention to the effect of Cas9 target residence on Cas9-induced palindromic SCF. Due to this extremely low frequency, it is difficult to directly analyze of Cas9-induced palindromic SCF. In this recent work studying Cas9-induced palindromic SCF (Kagaya et al., 2020), the authors stated: “It was complicated to analyze the types and the number of fusions in a given cell without harvesting the cell, and the exact timing of the fusion events was also exceedingly difficult to discern.” They had to specifically design an artificial fluorescence-based SCF reporter and integrated this reporter into the subtelomere of chromosome X. The integration into the subtelomere region could potentially increase the frequency of SCF. Even with this improvement, the frequency of Cas9-induced SCF-related products quantified by flow cytometry remains low, about 1% with 100% lentiviral infection of Cas9-sgRNA. With no support of an SCF reporter, Cas9-induced SCF events could not be enriched for analysis and PCR is a major approach in detecting Cas9-induced palindromic SCF in our study despite the limitations discussed above. Nevertheless, as Cas9 target residence at cleaved DNA causes palindromic SCF and potentially chromothripsis, we are currently developing a reporter for palindromic SCF to quickly and sensitively detect Cas9-induced palindromic SCF and monitor the regulation and fate of palindromic SCF and chromothripsis in cells in a different

ResFig 4. Palindromic SCF reporter. The palindromic SCF reporter contains two artificial exons A and B but the exon A is inverted so that RFP is not generated (a). After Cas9 induces a site-specific DSB in the reporter integrated at the Rosa26 locus of mESC (Chr 6), collision with DNA replication would dislodge Cas9-sgRNA from cleaved DNA, generating a three-ended DSB (b). Palindromic sister chromatids could join together by NHEJ to generate palindromic SCF, leading to a correct A-B splicing frame (c). This A-B splicing generates active RFP, making cells RFP⁺. Thus, the level of Cas9-induced RFP⁺ cells represents the relative frequency of palindromic SCF and can be quantified by flow cytometry (d).

research project. Our preliminary data indicate that the frequency of Cas9-induced palindromic SCF is ~0.16% in our system (ResFig 4), but the SCF products and their fate are yet to be further characterized.

Major comments:

1. The NHEJ reporter is not a suitable system to compare the various sgRNAs for their contribution of c-NHEJ. The authors write that in theory, a third of the indels can lead to GFP⁺ cells using this reporter, however, it is well known that CRISPR-induced breaks lead to very specific repair, which is this is sgRNA dependent (Brinkman et al. Nucleic Acids Res. 2014; van Overbeek, et al. Mol Cell. 2016; Shen, et al.; Nature. 2018; Allen, et al. Nat Biotechnol. 2018; Chakrabarti, et al. Mol Cell. 2019; Chen, et al. Nucleic Acids Res. 2019). Chakrabarti et al describes the existence of precise and non-precise guide, where the precise sgRNA can be repaired by mainly one type of indel. If this precise indel is not in the correct frame, the NHEJ reporter used in this paper will not pick up this repair. In that case the authors may incorrectly conclude that this guide has little involvement of c-NHEJ.

Moreover, in the presence of NU7441, repair by a-NHEJ may occur. This type of repair can result in a different indel in another frame, which again may be not detected with the reporter used.

Each sgRNA may result in a different distribution of repair over the 3 possible frames and they should all be considered. Therefore, the researchers should sequence the repair indels for each sgRNA and report which indels are observed in DMSO and NU7441 conditions.

In this manuscript, we used an NHEJ reporter to measure the frequency of Cas9-induced GFP⁺ cells, compare the frequencies between c-NHEJ-proficient cells and c-NHEJ-deficient cells and determine c-NHEJ engagement in repair of Cas9-induced DSBs. As explained in response to Q4 by reviewer #1, there is a 34-bp sequence between “Koz-ATG” and ATG for EGFP in the NHEJ reporter. In repair of Cas9-induced DSBs at a site between “Koz-ATG” and the ATG-GFP coding region, both c-NHEJ and a-EJ can generate indels at the repair junction. In general, net addition of “3n+2” bp or net loss of “3n-1” bp in indels can reframe the 34-bp frame-shift to in-frame, leading to production of GFP⁺ cells. Given uniqueness in repair of Cas9-induced DSBs, previous statement in the manuscript “in theory, a third of the indels can lead to GFP⁺ cells” is not accurate. We thank the reviewer for pointing this out and have made a change accordingly.

Why could a change in the frequency of Cas9-induced GFP⁺ cells reflect the choice between c-NHEJ and a-EJ in NHEJ reporter cells? We have stated in the manuscript: “As c-NHEJ and a-EJ generate different proportions of accurate NHEJ (accNHEJ) products and indel-based mutagenic NHEJ (mutNHEJ) products, inactivation of c-NHEJ would channel more Cas9-induced DSBs towards error-prone a-EJ in addition to HDR, increasing the frequencies of mutNHEJ.” The increase of mutNHEJ upon c-NHEJ inactivation could proportionally increase the probability of the

mutNHEJ types that correct the frame in the NHEJ reporter, generating relatively more GFP⁺ cells. Thus, an increase in the frequency of Cas9-induced GFP⁺ cells could reflect more involvement of a-EJ in repair of Cas9-induced DSBs due to c-NHEJ inactivation.

We and others have found that 1-bp templated insertions (TIs) are frequent in Cas9-induced NHEJ at some sites, but not all sites (Allen et al., 2019; Brinkman et al., 2018; Chakrabarti et al., 2019; Guo et al., 2018; Lemos et al., 2018; van Overbeek et al., 2016; Shen et al., 2018; Shou et al., 2018). The concern raised by the reviewer is that exclusion of these frequent 1-bp TIs may cause a favored or disfavored bias in our analysis of c-NHEJ engagement. For example, we may draw an incorrect conclusion if NHEJ is completed by 1-bp TIs which could not be measured by flow cytometry due to incorrect frame for GFP expression. In addition, if a major type of indels such as 1-bp TIs happen to be products of a-EJ, we would not be able to detect them in the presence of NU7441 using the NHEJ reporter. Indeed, 1-bp TIs in Cas9-induced indels do not correct the frame in our NHEJ reporter to generate Cas9-induced GFP⁺ cells. However, 1-bp TIs undergo no end resection and are considered as accurate NHEJ products (Brinkman et al., 2018; Guo et al., 2018; Lemos et al., 2018; Shou et al., 2018). Therefore, although 1-bp TIs, like accurate NHEJ events, are not picked up in the NHEJ reporter assays for GFP⁺ cells, c-NHEJ inhibition by NU7441 could reduce accurate NHEJ including 1-bp TIs and increase the probability of a-EJ that generates more GFP⁺ cells. In other words, c-NHEJ engagement could be determined by measuring the frequency of Cas9-induced GFP⁺ cells regardless of how frequent 1-bp TIs are.

However, if 2-bp TIs or 1-bp deletions, like accurate NHEJ or 1-bp TIs, are among the major types of c-NHEJ, we would agree with the concern that “the authors may incorrectly conclude that this guide has little involvement of c-NHEJ.” In this case, as both 2-bp TIs or 1-bp deletions correct the frame and generate GFP⁺ cells, c-NHEJ inactivation could lower the frequency of Cas9-induced GFP⁺ cells by reducing 2-bp TIs or 1-bp deletions but increase the frequency of Cas9-induced GFP⁺ cells by reducing accurate NHEJ or 1-bp TIs. Because it is not clear how this offset shifts, we might draw an incorrect conclusion. However, even in studies mentioned by the reviewer, neither 2-bp TIs nor 1-bp deletions are dominant among types of Cas9-induced indels and their effect could be ignored.

Nevertheless, we used alternative approaches to validate our conclusions. First, if c-NHEJ is engaged, c-NHEJ inactivation would stimulate Cas9-induced HDR and this stimulation is independent of NHEJ indel types. Using this strategy, we confirmed that c-NHEJ engagement varies in repair of Cas9-induced DSBs between different target sites. Secondly, as suggested by the reviewer, we performed targeted sequencing and junction analysis on the c-NHEJ-independent site gEJw7 and the c-NHEJ-dependent site gEJc5 from Fig. 1e. More than 25% of total NHEJ products were 1-bp TIs at gEJw7 but less than 0.5% at gEJc5. In either case, consistent with

NHEJ reporter assays, DNA-PKcs inhibition by NU7441 has no effect on the editing efficiency at the gEJw7 site, but reduces the editing efficiency at the gEJc5 site (new Fig S2b). The deletion length and the MH usage were altered slightly at the gEJw7 site but strongly at the gEJc5 site by DNA-PKcs inhibition. In contrast, the deletion length became slightly longer but MH usage changed little after DNA-PKcs inhibition. Therefore, c-NHEJ is indeed engaged to repair Cas9-induced DSBs at the gEJc5 site but little at the gEJw7 site. We have added the data into the revised manuscript (new Fig S2b-e). Thirdly, targeted sequencing and junction analysis of 8 sites at two natural loci also revealed that c-NHEJ engagement varies in repair of Cas9-induced DSBs between different target sites (new Fig S3a-c). The effect of one particular dominant indel type was excluded. Lastly, Fig 3 further demonstrated that mismatched or truncated sgRNAs and SpCas9 variants promotes a bias towards c-NHEJ in repair of Cas9-induced DSBs. This supports our conclusion that Cas9-sgRNA target interaction or target residence influence c-NHEJ engagement.

2. In almost all figures the authors consider repair to be just binary e.g. In figure 1a-c the authors do not take into consideration repair by a-NHEJ (either with microhomology or without), even though these pathways are reported to play a role in CRISPR/Cas9-induced DSB repair. a-NHEJ repair even may primarily compete with the HDR repair since both pathway are dependent on resection. Repair by a-NHEJ can result that less HDR repair occurs at a particular DSB. Therefore the conclusion that there is little or no c-NHEJ involvement because usage of NU7441 did not increase GFP+ in the HDR reporter needs nuance.

In figure 1d-g, the authors do not consider the repair by HDR, but only c-NHEJ and a-NHEJ, which raises similar concerns.

Thanks for the comment. These three pathways c-NHEJ, a-EJ and HDR compete in repair of Cas9-induced DSBs at a given site. Given the dominant nature of c-NHEJ, it is well documented that c-NHEJ inactivation would promote both HDR and a-EJ (Allen et al., 2002; Chang et al., 2017; Delacôte et al., 2002; Pierce et al., 2001). If c-NHEJ inactivation does not promote either HDR or a-EJ at a given site, we can infer that c-NHEJ is little engaged at this site. If we understand it correctly, the reviewer was concerned that the competition between HDR and a-EJ in the absence of c-NHEJ may reverse the stimulation of either HDR or a-EJ, leading to the misinterpretation on lack of c-NHEJ engagement. However, if c-NHEJ is engaged, c-NHEJ inactivation would channel unrepaired DSBs that are supposedly repaired by c-NHEJ to alternative pathways for repair, some for HDR and some for a-EJ. It is impossible that this channeling is restricted to a single pathway. In particular, HDR and many sub-pathways of a-EJ do not directly compete. For example, a-EJ sub-pathways in G1 phase do not compete with HDR. Even if some sub-pathways of a-EJ, e.g., polymerase theta-mediated end-joining (TMEJ) in the S phase, may compete with HDR (Ramsden et al., 2022), the stimulation of HDR or a-EJ by c-NHEJ inactivation would be maintained, at least partially if not fully. In addition, a-EJ was originally defined as the NHEJ pathway operating without either one of

core NHEJ factors, not by their products; this definition is still useful and less confusing (Ramsden et al., 2022). It is difficult to distinguish a-EJ from c-NHEJ by the MH usage or deletion length in individual repair products alone without knowing the status of c-NHEJ factors. Therefore, in this manuscript, we distinguished a-EJ from c-NHEJ only when c-NHEJ is inactivated regardless of indel types. The effect of c-NHEJ inactivation on the efficiency of HDR or a-EJ at a given site would indicate whether or not c-NHEJ is engaged at this site.

*Certainly, it would be ideal that we could simultaneously measure HDR and a-EJ at a same site in the absence of c-NHEJ. However, our reporters do not have such capacity. With the HDR reporter, we can alternatively sequence the junction of repair products from the HDR reporter assays and distinguish NHEJ products from HDR products at the same site as shown in new **Fig 7c** and **7d**. In this way, we measured HDR and NHEJ simultaneously at the same sites in the presence or absence of c-NHEJ and confirmed the HDR bias induced by c-NHEJ inactivation as shown in **Fig 7d**. But this approach requires targeted deep sequencing of repair junction in the HDR reporter and is not convenient to use.*

In figure 1f-g, only total indels are reported, while it is known that c-NHEJ and a-NHEJ can generate different types of indels (van Overbeek, et al. Mol Cell. 2016; Shen, et al.; Nature. 2018; Allen, et al. Nat Biotechnol. 2018; Brinkman et al. Mol Cell. 2018; Chakrabarti, et al. Mol Cell. 2019; Chen, et al. Nucleic Acids Res. 2019). The total amount of indels does not say anything about the involvement of c-NHEJ repair alone. Since all repair pathways are present in the cell, repair by c-NHEJ, a-NHEJ and HDR can compete with each other and can take over when one repair pathway is inhibited (van Overbeek, et al. Mol Cell. 2016; Brinkman, et al. Mol Cell. 2018; Schep et al. Mol Cell. 2021). The authors should present their data more thoroughly, in view that a-NHEJ or HDR may influence their data and not immediately conclude that when the amount of indels or GFP+ cells in NU7441 condition is similar as in the DMSO conditions there is no involvement of c-NHEJ repair at this CRISPR-induced DSB.

*Thanks for the comment. As explained above, we can reasonably infer that c-NHEJ is little engaged if c-NHEJ inactivation does not promote either HDR or a-EJ at a given site. We have additionally analyzed the deletion length and the MH usage at repair junctions of Cas9-induced NHEJ at two sites of the NHEJ reporter and 4 sites each of Rosa26 and Col1 loci (new **Fig S2c,d** and **Fig S3a,b**). At the gC2, gC3, gR3 and gEJc5 sites, NU7441 significantly altered the pattern of the deletion length and the pattern of the MH usage, indicating c-NHEJ engagement. At the gC1, gC4, gR1, gR2, gR4 and gEJw7 sites, the alteration is much milder, suggesting little or moderate engagement of c-NHEJ. These results are consistent with the reporter assays.*

We agree that the total amount of indels does not reveal details of c-NHEJ, a-EJ and HDR. But we cannot distinguish a-EJ from c-NHEJ by individual repair products

alone as explained in response to Q4 of Reviewer #1. It is still useful and less confusing to use the original definition of a-EJ by defining c-NHEJ and a-EJ by participating repair factors or lack of these factors, not by their products (Ramsden et al., 2022). For example, as both c-NHEJ and a-EJ can produce accurate end joining despite with different frequencies, it is difficult to tell whether an accurate repair product is generated by c-NHEJ or a-EJ without knowing the status of c-NHEJ factors. Therefore, in this manuscript, we distinguished a-EJ from c-NHEJ only when c-NHEJ is inactivated regardless of indel types. The effect of c-NHEJ inactivation on the efficiency of HDR or a-EJ at a given site would indicate whether or not c-NHEJ is engaged at this site.

3. The authors mention several times that CRISPR/Cas9-induced DSB are repaired accurately. This is still a matter of debate. High c-NHEJ precision was reported when DNA breaks were introduced by I-SceI (Guirouilh-Barbat et al, *Frontiers in Genetics*, 2014). Also, it has been reported in vitro and also in wt yeast that blunt breaks are preferentially joined imprecisely (Boulton, *Nucleic Acids Res*, 1996; Schär et al, *Genes and Development*, 1997). Kinetics experiments suggested that CRISPR/Cas9-induced breaks were repaired mainly mutagenic (Brinkman, *Mol Cell*. 2018). The fidelity of DSB repair upon induction of Cas9 in combination with two co-transfected sgRNAs that target adjacent sequences was shown to result in accurate re-joining in varying degrees (Guo et al. *Genome Biology*. 2018). But re-joining of single end breaks may be different from re-joining the ends of two different breaks.

The authors suggest a cycle of Cas9 cutting and perfect repair that terminates when a 'rare' imperfect repair event occurs, this seems contradictory with the long residence time of Cas9 reported by others (Richardson et al., *Nature biotechnology*, 2016; Ma et al. *The Journal of cell biology* 2016). The presence of target recleavage by Cas9 is not convincingly shown from their data and the authors should not conclude that Cas9 recleavage occurs and that c-NHEJ is accurate without more evidence.

A similar concern was raised in the beginning by the reviewer and addressed accordingly above.

As described, when we think of Cas9 target re-cleavage, we consider: 1) Cas9-sgRNA targets can be reused; 2) The Cas9-sgRNA enzyme complex can be reused. Reuse of Cas9-sgRNA targets requires accurate repair of Cas9-induced DSBs. Since Cas9-induced DSBs have two clean ligatable ends, it has been proposed that c-NHEJ repair of Cas9-induced DSBs is largely accurate and Cas9-mediated recleavage of accurate NHEJ products amplifies the mutagenicity of c-NHEJ (Danner et al., 2017; White et al., 2016). Because it is impossible to directly distinguish accurate NHEJ products from un-cleaved targets in repair of Cas9-induced DSBs, we and others had employed several indirect methods to analyze accurate NHEJ. First, we had used paired Cas9-sgRNAs to demonstrate that Cas9-induced NHEJ are inherently accurate (Guo et al., 2018). Secondly, Song et al had used precise insertion of 34-bp double

strand oligodeoxynucleotides (ODNs) into Cas9-induced DSBs to determine the efficiency of accurate NHEJ in repair of Cas9-induced DSBs (Song et al., 2021). Based on their data, they stated “the average value of NHEJ accuracy is approximately 75% in maximum in HEK 293T cells”. Recently, Yin et al has used Cas9-induced translocations to demonstrate accurate end joining at translocation junctions and deduce the occurrence of repeated cleavage of re-targetable junctions by Cas9-sgRNA (Yin et al., 2022). These publications were cited in the revised manuscript.

*The notion that SpCas9 is a single-turnover enzyme may lead to the belief that SpCas9 can only be used once. However, in this in vitro study that defined SpCas9 as a single-turnover enzyme (Sternberg et al., 2014), Sternberg et al stated that “The finding that Cas9:RNA remains tightly bound to both ends of the cleaved DNA, suggests it acts as a single-turnover enzyme.” Therefore, this study did not exclude the possibility that it can be reused to target DNA if SpCas9-sgRNA is released from cleaved DNA. In particular, this in vitro study has not taken the cellular context into account yet. Moreover, either in this work by Sternberg et al or in the studies mentioned by the reviewer (Ma et al., 2016; Richardson et al., 2016), only a very limited number of sites were used to indicate this extraordinary long residence. Given the variability in Cas9 target interaction between different sites and local chromatin context for Cas9-induced DSBs, it is expected that target dissociation and by extension target residence would differ between sites. In cells, some of Cas9-sgRNA are dislodged by external forces such as DNA replication (in this study), transcription and chromatin remodeling (Clarke et al., 2018; Wang et al., 2020). Given that Cas9-induced DSBs can be accurately repaired, accurate repair products can be re-cleaved by Cas9-sgRNAs that either target for the first time or are re-used after dissociation from their targets. In cells, Cas9-sgRNAs dissociated from cleaved DNA by transcription or chromatin remodeling could serve as a multi-turnover enzyme (Clarke et al., 2018; Wang et al., 2020). New **Fig. 4** in our revised manuscript indicated that SpCas9-sgRNA variants could be spontaneously dissociated from cleaved DNA, possibly providing an opportunity for SpCas9 to cleave DNA again. In addition, in our manuscript, by reducing the concentration of Cas9-sgRNA in cells by serial dilution for transfection to limit re-cleavage of accurate NHEJ, we found that re-cleavage of accurate NHEJ products by Cas9 could amplify the mutagenicity of c-NHEJ in CRISPR/Cas9 genome editing (new **Fig 2a-d**). Therefore, with new **Fig 4** and the support of additional published work, it has now become more convincing that c-NHEJ repair of Cas9-induced DSBs is inherently accurate and accurate NHEJ products can be repeatedly cleaved by Cas9 until mutations are introduced.*

4. It is not clearly presented how well the transfection efficiency of the sgRNAs was in the various experiments. The method states that the figures are normalized for transfection efficiency, but how normalization was performed and what the average percentage is of the transfection efficiency is missing. This is especially important for figure 2b-f. In figure 2b, the authors claim that when reducing the amount of

Cas9-sgRNA, the NU7441 inhibitor initiates a reverse effect on repair and begins to stimulate GFP production. The figure shows that this effect takes place when only ~0.2% to ~0.02% GFP+ cells are detected. This effect may only represent a handful of cells when considering 2×10^5 cells were transfected. Low DNA input for transfection results in a low number of events and my concern is that there are too few events to make data reliable. The authors should clarify their experimental conditions and whether this experiment represent a large enough pool of cells to draw solid conclusions, if not the experiment should be performed with more cells.

We thank the reviewers for pointing out this issue. We have now provided more detailed information on experimental procedures and calculations including normalization with transfection efficiencies (TEs) in the Methods and Materials in the revised manuscript. In Fig 2, transfection was performed in a 24-well plate with 2×10^5 cells. TEs in this work were mostly over 50% and indicated in the figure legends in the revised manuscript. The experiments with TEs lower than 50% were excluded for analysis. Given that the doubling time for mESC is around 12 hours, the number of cells could easily reach over 1×10^6 cells at 3 days post transfection. Because of the concern that too few GFP+ cells may hurt the reliability of the

Exprt#1		DMSO			NU7441		
Triplicates		1	2	3	1	2	3
Control	Total	334671	400943	368137	148772	180273	141083
	GFP+	4	3	3	4	1	6
0.25ug	Total	465712	475990	444001	219111	203782	198473
	GFP+	42369	42839	42457	10671	11792	11890
0.1ug	Total	443063	414408	480319	227923	181328	182415
	GFP+	40098	40125	40156	13486	12085	11779
0.01ug	Total	479300	441374	424788	215669	190370	191586
	GFP+	14389	13964	14789	6638	6583	6553
0.001ug	Total	429658	416845	432394	254718	217388	227694
	GFP+	1060	1042	1041	792	786	780
0.0001ug	Total	743299	676885	707499	298832	291040	293372
	GFP+	127	127	139	105	98	126

Exprt#2		DMSO			NU7441		
Triplicates		1	2	3	1	2	3
Control	Total	395347	334361	274308	60586	56443	66940
	GFP+	5	15	9	8	2	2
0.25ug	Total	208350	312842	302651	65804	84186	54176
	GFP+	19855	28163	28126	4075	4332	4169
0.1ug	Total	198618	203082	236871	45138	47394	51156
	GFP+	17844	18830	18253	2550	2981	2828
0.01ug	Total	157647	143216	79124	36630	27517	25602
	GFP+	6848	6482	4342	1472	1423	1451
0.001ug	Total	148298	122626	141795	61216	61212	60602
	GFP+	413	279	355	201	229	215
0.0001ug	Total	1226753	1721493	1265483	466440	434589	435261
	GFP+	250	312	280	109	130	191

ResFig 5. The raw data of flow cytometry in 2 representative independent experiments among 5, each in triplicates for Fig 2b.

measurement, the total number of cells analyzed by flow cytometry at 3 days post-transfection in new **Fig 2b** was mostly over 2×10^5 viable cells to yield more than 100 Cas9-induced GFP⁺ cells for the frequency of GFP⁺ cells at ~0.2% to ~0.02%. For other figures, total number of cells analyzed by FACS was usually around 2×10^5 cells. The raw number of each flow cytometry measurement in 2 representative experiments among 5, each in triplicates for new **Fig 2b** are listed in **ResFig 5** for the reviewer's evaluation.

5. The method section is inadequately described. Several times is a referred to previous publications. In the referred publication, the reader is referred to yet another publication. In this treasure hunt, many parts of the experiments remained unclear. E.g. the sequence of the reporter plasmids, transfection conditions, the way the transfection efficiency was calculated and was used for normalization. An adequate reference to the basic paper(s) would be of great help for the interested reader. Also the experimental conditions should mentioned in the manuscript.

*We thank the reviewer for the comment and suggestion. We have now provided more detailed information on experimental procedures including transfection, measurement of transfection efficiencies (TEs) and normalization with TEs in the Methods and Materials in the revised manuscript. The target sequences in the NHEJ and HDR reporters were also showed in **Fig S1a** and **S2a**. The reporter plasmids and their sequence information would be provided upon request. The publications improperly cited for unique methods and constructs have now been replaced with the proper, original publications.*

6. A shortcoming in the paper is that it does not state anything about cell viability after inducing DSBs, since the percentage of GFP can vary widely when a sgRNA affects cell viability. The authors discuss that they found a potentially source for chromothripsis and complex chromosomal rearrangements, events likely to influence the cell viability. Also in cells with KOs in components of the c-NHEJ pathway, the cell may not be able to cope with the DNA repair as well. Cell viability can be measured with various compounds followed by fluorescence and it would improve the outcome of the data.

Because mammalian cells have evolved elegant DSB repair machineries to efficiently repair DSBs, a limited number of DSBs can be repaired timely and properly and do not cause cell death. In this manuscript, we used Cas9-sgRNA to introduce a single DSB in the cells and it is expected that the effect on cell viability would not be significant. In particular, mESC used in this study proliferate rapidly and are associated with a high level of replication stress. It is likely that a single DSB induced in these cells would be well tolerated. In addition, Cas9-induced chromothripsis or gross rearrangements occurs with a very low frequency (Kagaya et al., 2020; Leibowitz et al., 2021; Umbreit et al., 2020). Its effect on cell death is restricted to very few cells and hard to be detected among a large pool of cells. Nevertheless, we

have analyzed proliferation of mESC including wild-type cells and KO cells transfected with Cas9-sgRNAs. Cell proliferation is similar between cells transfected with control vector and with Cas9-sgRNA or dCas9-sgRNA, suggesting little effect of Cas9-sgRNA on the viability of mESC (ResFig 6).

ResFig 6. The effect of Cas9-sgRNA on proliferation of isogenic DNA-PKcs^{+/+} mESC (a) and DNA-PKcs^{-/-} mESC (b) transfected with control empty vector (EV), dCas9-sgRNA and Cas9-sgRNA.

7. In my opinion the authors provide no new mechanistic insight or explanations for possible differences in repair choice. The authors argue that Cas9 stickiness could influence repair choice. Unfortunately, the authors don't show this in their own system. The effects of the asymmetry of this stickiness on gene targeting was the major thrust of the Corn Nature Biotechnology paper. At least a simple in vitro assay could have been performed with some of their CRISPR guides, Cas9 and target sequence (e.g. PCR fragment). After incubation, the reaction products can be separated by non-denaturing agarose electrophoresis. If a sticky complex is present the DNA will appear as unbroken on the gel.

We thank the reviewer for the comment and suggestion. Cas9 target residence or dissociation analyzed in this study is quite different from the Corn paper showing earlier release of the PAM-distal non-target DNA strand but persistent binding with the other three strands (Richardson et al., 2016). In fact, due to Cas9-sgRNA residence at both PAM-proximal and PAM-distal ends of cleaved DNA, Cas9-induced DSBs are exposed by symmetric dissociation of Cas9 from both DSB ends (Richardson et al., 2016; Sternberg et al., 2014; Zhu et al., 2019). To provide some mechanistic insight into the mechanism by which target residence or dissociation of Cas9-sgRNA from cleaved DNA influence c-NHEJ engagement, we performed in vitro Cas9-sgRNA cleavage assays, target residence assays and competition assays as suggested by the reviewer. We found that sgRNA mismatch or truncation in Fig 3 indeed reduced target residence of Cas9-sgRNA (new Fig 4). Compared to sgRNA with perfectly matched 20-nt spacer (wild-type sgRNA), mismatched or truncated sgRNAs caused a slower rate of DNA cleavage by Cas9 although all target DNA were completely cleaved after 24-h reaction (new Fig 4a). After DNA cleavage, more cleaved DNA were released from the Cas9-sgRNA-DNA ternary complex for

mismatched or truncated sgRNAs as compared to perfectly matched sgRNA (new Fig 4b). This suggests that post-cleavage Cas9-sgRNA target residence is reduced by mismatched or truncated sgRNAs. The target binding of dCas9-sgRNA was also less efficient when sgRNAs were mismatched or truncated (new Fig 4c). After target binding of dCas9-sgRNA and dCas9-sgRNA mutants, the competition by SpCas9-sgRNA (wild-type) revealed that target DNA in the dCas9-sgRNA-DNA ternary complex were released more from dCas9-sgRNA mutants than from dCas9-sgRNA (wild-type) (new Fig 4d). Together, these data confirmed that the mutations of sgRNA reduced target interaction and target residence of Cas9-sgRNA. We have added these results into the revised manuscript.

Minor points:

The aim of the authors was to study how the varying affinities of Cas9-sgRNA complex after DNA cleavage influences repair and hamper the efforts to predict efficiency and specificity of CRISPR/Cas9 genome editing.

Could the authors elaborate in the discussion on how can we take target interaction along in sgRNA design, based on their research findings?

Thanks for the comment. In the revised manuscript, we proposed several potential applications possibly by harnessing target interaction or target residence of Cas9-sgRNA or dCas9-sgRNA. First, Cas9-sgRNA variants with shorter target residence allows more spontaneous target dissociation from cleaved DNA and increases c-NHEJ engagement, thus reducing the occurrence of dangerous chromosomal rearrangements such as palindromic SCF. Secondly, Cas9-sgRNA variants with enhanced binding affinity or with a long resident duration at their DNA targets could increase activities of dCas9-based effectors and improve dCas9-based applications. Thirdly, because global c-NHEJ inactivation may exacerbate off-target effects due to the favored involvement of c-NHEJ at off-target sites, the strategy of local c-NHEJ inhibition is needed to improve Cas9-mediated HDR while not exacerbating off-target effects. However, we do not expect that our manuscript would help the detailed design of Cas9-sgRNA variants. It only points enhancing or weakening target residence for different CRISPR applications as a potential direction for CRISPR/Cas9 engineering.

In several graphs the y-axes state "induced relative c-NHEJ", which is an interpretation of the data. The axis should describe what you measure e.g. "relative induced GFP"

We apologize for this confusion. In this manuscript, we first used the frequency of induced GFP⁺ cells to indicate the efficiency of Cas9-induced NHEJ or HDR and then calculated relative NHEJ or HDR (versus DMSO mock treatment or isogenic wild-type control where the frequency is normalized to 1) to demonstrate the relative change of Cas9-induced NHEJ or HDR by chemical inhibition or genetic inactivation of c-NHEJ. The first is what we measured and the second is the comparison analysis

of what we measured.

How come figure 2b and 2f does not show the same results for gEJw6? The GFP+ cells seems to be much higher in figure 2f in the 3 replicates.

In our reporter assays, we performed at least three independent experiments, each in triplicates. Because independent experiments were usually done in different time, the readout of induced GFP⁺ cells may vary between independent experiments due to variations in cell passage, cell state and transfection etc. (Rouet et al., 1994; Xie et al., 2004, 2009). This is probably why many publications presented only relative values. However, these experimental variations were mostly ignorable in each independent experiment with the same cell passage, cell state and transfection. Therefore, despite the variations between independent experiments, it is important to compare the effect of treatments such as chemical inhibition (i.e., NU7441) or genetic inactivation (i.e., XRCC4 deletion) in each independent experiment and determine whether the effect of treatment is consistent between at least three independent experiments. Nevertheless, the more serious these experimental variations between independent experiments, the bigger difference in the readout of induced GFP⁺ cells. Thus, these variations should be better controlled and reduced.

Unexpectedly, the XRCC4^{+/+} mutant is affected (almost) as much as XRCC4^{-/-} cells when transfected with gEJw4 and gEJw6 in figure 2f. Do the authors have an explanation for this?

If we understand this comment correctly, the reviewer wondered why the frequency of Cas9-induced GFP⁺ cells at the gEJw4 and gEJw6 sites was reduced in XRCC4^{+/+} cells as much as in XRCC4^{-/-} cells compared to DMSO or NU7441. The wild-type cells for treatment with DMSO or NU7441, XRCC4^{+/+} cells and XRCC4^{-/-} cells were derived from the same NHEJ reporter mESC; however, the experiments with DMSO and NU7441 were done independently from those with XRCC4^{+/+} and XRCC4^{-/-} cells. As mentioned above, the readout of induced GFP⁺ cells may vary between independent reporter assay experiments in different time due to experimental variations in cell passage, cell state and transfection etc. (Rouet et al., 1994; Xie et al., 2004, 2009). This caused the variations the reviewer pointed out. However, in each independent experiment, treatment with NU7441 was done in parallel with DMSO and XRCC4^{-/-} in parallel with XRCC4^{+/+}. The effect of each paired treatment was consistent between independent experiments.

The counts of the lines should continue over the whole manuscript and not start recounting at each page. This makes it hard to refer to the correct line.

Thanks for the suggestion. We made the change accordingly.

Typo's. Word "at" too much in the last sentence of in the part 'Inactivation of c-NHEJ

induces varying stimulation of Cas9-induced HDR among targets'. "mollify" instead of "modify" in discussion page 27 line 21.

Thanks for the comment. We originally used "not...at all at the targets" in the sentence. We now removed "at all". The meaning of "mollify" is "reduce the severity of". "Modify" does not have such meaning. We thus used "mollify off-target effect" to express the meaning "reduce off-target effect".

Reviewer #3: The manuscript by Liu, Feng, and co-authors adds to the recent literature in understanding DNA repair after genome editing. This is an important area of ongoing research with much uncertainty in the DNA repair pathway choices and competing repair pathways. This manuscript seeks to address if persistent binding by Cas9 to the target locus favors certain repair outcomes and if decreasing the energy of binding can alter these outcomes. To explore the hypothesis, the authors use a truncated GFP repair reporter with a panel of gRNAs to determine bias toward HDR and an out-of-frame GFP reporter to measure NHEJ. They created a series of genetically modified cells deficient in c-NHEJ repair proteins. The authors measured editing outcomes with mismatched gRNAs and reported high-specificity Cas9 variants to determine if lower binding energy affects gene editing outcomes. Finally, they attempt to determine if transcription or replication impact DNA repair outcomes.

Understanding the risk of on-target chromosome rearrangements is also important considering ongoing clinical translation of such technologies and the tendency of conventional PCR and NGS assays to exclude such results. The manuscript uses one figure to describe transcriptional control and the lack of measured effect on NHEJ. The final figure indicates the potential of chromosomal rearrangements.

Overall, I think this manuscript warrants publication after concerns are addressed. It adds new observations to the published literature on 1) the complexity of DNA repair 2) indicates that promoting HDR may have the unintended consequence of increasing off-targets and 3) adds to growing concerns of on-target chromosome alterations.

We thank the reviewer very much for spending time reviewing our work.

The following are my major concerns and a few minor comments for consideration.

Major concerns:

1. Data presentation and irregularities in data normalization. Throughout the manuscript, data is presented as raw data with individual measured replicates on the left with corresponding normalized data on the right. Individual data points are appreciated. The normalization isn't straightforward and may invite criticism. The error bars do not always translate which may indicate the error is being lost in the normalization. The mean changes based on the normalization. For example: How is SpCas9-HF1 gEJc5 XRCC4—nearly 5 fold higher in the right graph when it appears

10-30% higher in the left graph? From what I gather, this may come from multiple normalization steps - e.g. (A16T-NU7441/A16T-DMSO)/(20-NU7441/20-DMSO). This should be clarified. Consider other means of presentation or normalization.

We thank the reviewer for the comment and suggestion. We changed our presentation and normalization to be more straightforward in the revised manuscript. In the previous version, as the review correctly understood, we did multiple normalization steps, e.g., (A16T-NU7441/A16T-DMSO)/(20-NU7441/20-DMSO) or (SpCas9-HF1-XRCC4^{-/-}/ SpCas9-HF1-XRCC4^{+/+})/(SpCas9 20-nt-XRCC4^{-/-}/SpCas9 20-nt-XRCC4^{+/+}). This was intended to show the effect of reducing target interaction on c-NHEJ engagement by comparison with SpCas9 20-nt DMSO treatment or SpCas9 20-nt in XRCC4^{+/+} cells, but indeed not straightforward. We thus reduced normalization steps in the revised manuscript and used 'Relative Cas9-induced NHEJ or HDR' on the right to indicate the original fold change of the frequency of Cas9-induced GFP⁺ cells by chemical inhibition or genetic inactivation c-NHEJ (i.e., versus DMSO treatment or XRCC4^{+/+} cells where the frequency of Cas9-induced GFP⁺ cells are normalized to 1).

2. Negative controls to indicate reporter activity in the absence of DSB. Many reporters have some baseline 'leakiness'. This could be indicated on the graph (e.g. a dotted line). This would be more important in the dose limiting experiments.

The leakage of the NHEJ and HDR reporters was rare and reported at ~0.01% or less previously (Xie et al., 2004, 2009). In this study, the background levels were also low at 0.01% or less for both reporters (ResFig 7) and noted in the revised manuscript. Also as described in the Methods and Materials, in each time of experiments, we measured the baseline frequencies of GFP⁺ cells, the frequencies of GFP⁺ cells after transfection with Cas9-sgRNA, and transfection efficiencies together. The frequencies of Cas9-induced GFP⁺ cells were subsequently calculated first by correcting with the background frequencies and then by normalizing with transfection efficiencies.

ResFig 7. The background level of the NHEJ reporter (a) and the HDR reporter (b) in mESC transfected with control empty vector. Total numbers of cells analyzed were no less than 200,000 and the frequency of GFP⁺ cells is no more than 0.01%.

3. Statistics: Looking at Figure 1F, is gC2 significant? Was this a simple t-test or were statistics adjusted for multiple comparisons? What happened to the error bars between the left and right graphs?

In Fig 1f, we performed 4 independent experiments, and two-tailed Student's paired t-test was used for statistical analysis ($p < 0.05$ for gC2). Because independent experiments were usually done in different time, the readout of induced GFP⁺ cells may vary between independent experiments due to variations in cell passage, cell state and transfection etc. (Rouet et al., 1994; Xie et al., 2004, 2009). In this case, the experimental variations between independent experiments were not small. However, despite these variations, we were still able to compare the effect of NU7441 treatment on Cas9-induced NHEJ at each site in each independent experiment and determine whether the effect was consistent between these four independent experiments. Relative SpCas9-induced NHEJ on the right was calculated by normalizing the editing efficiency with DMSO treatment to 1.0 in each independent experiment and the experimental variations between independent experiments were cancelled out to some extent. Thus, the error bars on the right graphs were much smaller. We checked the values and calculation of the error bars again in this figure and found no mistakes. Nevertheless, the more serious these experimental variations between independent experiments, the bigger difference in the readout of induced GFP⁺ cells. Thus, these variations should have been controlled better and minimized.

4. Conclusions based on HEK293 and mES clonal cell lines. The conclusions of this manuscript were derived from these two cell lines. This should be made clear in any conclusions derived. Clone-to-clone variation was not noted in the manuscript. This could mask trends or create them. For example, why in Figure 3d would we expect differences in HDR efficiency between DMSO treated, WT, and XRCC4⁺⁺? These can vary wildly in their HDR efficiency and are treated as a denominator for normalization between left and right graphs

Thanks for the suggestion. We have specified the cell types for our conclusions in the revised manuscript. The clone variation was examined in the beginning of the study and also noted in the revised manuscript. We used the HDR reporter mESC to generate isogenic DNA-PKcs^{-/-} and Ku80^{-/-} cells as well as wild-type control cells by CRISPR/Cas9 and analyzed 2 wild-type clones, 3 DNA-PKcs^{-/-} clones and 2 Ku80^{-/-} clones in Fig S1d. The clone variation between isogenic XRCC4^{+/+} and XRCC4^{-/-} HDR reporter clones used in this study was evaluated in our previous study (Xie et al., 2009). The XRCC4^{+/+} HDR reporter mESC clone was also used for treatment with DMSO and NU7441. The parental HDR reporter mESC for generation of DNA-PKcs^{-/-} and Ku80^{-/-} clones and the parental HDR reporter mESC for XRCC4^{+/+} and XRCC4^{-/-} clones were from different origins (Yan et al., 2006; Zha et al., 2008). Therefore, the WT control for DNA-PKcs^{-/-} clones and Ku80^{-/-} clones and the XRCC4^{+/+} cell control for XRCC4^{-/-} cells could produce variation in the frequencies

of Cas9-induced GFP⁺ cells. In addition, the readout of induced GFP⁺ cells may vary between independent reporter assay experiments in different time due to experimental variations in cell passage, cell state and transfection etc. (Rouet et al., 1994; Xie et al., 2004, 2009). As the experiments with DMSO and NU7441 were done independently from those experiments with XRCC4^{+/+} and XRCC4^{-/-} cells, experimental variations could also be generated with the wild-type XRCC4^{+/+} cells. As we mentioned above, although the effect of each paired treatment was consistent between independent experiments, the experimental variations should have been controlled better to minimize.

5. Figure 5 and resulting conclusions are not definitive. Considering the previously published literature (e.g Clarke et al, ref 18) it seems that transcriptionally driven Cas9 dissociation would have the same impact as reduced affinity. As the manuscript shows, this was not observed. The reasons could be 1) as described by the authors or 2) the method did not achieve sufficient transcriptional blocking to measure the intended affect. Gene silencing was variable but mostly very weak, typically below 50% with the exception of gGw5. Suggestions include revising the discussion on this finding OR performing lentiviral transduction to generate persistent silencing.

*We thank the reviewer for the comment and suggestion. We added some new discussion on the effect of transcription collision on c-NHEJ engagement in the revised manuscript. After we found that c-NHEJ was little engaged in repair of Cas9-induced DSBs at some sites, we hypothesized that target dissociation of Cas9-sgRNA at cleaved DNA by collision with local transcription at these sites may prevent c-NHEJ engagement. We provided evidence to show that lack of c-NHEJ engagement was not caused by transcriptional collision at Cas9-induced DSBs. Based on the published report by Clarke et al (Clarke et al., 2018), target dissociation of Cas9-sgRNA at cleaved DNA by local transcription would instead expose Cas9-induced DSBs for c-NHEJ repair and facilitate re-cleavage of accurate NHEJ products by Cas9 as a multi-turnover enzyme to increase the level of editing. In this regard, our data did not contradict the Clarke study although different questions were addressed. New **Fig 6c** indicated that c-NHEJ engagement was little affected by the annealing of sgRNAs to either the template or non-template strand of transcription. New **Fig 5d** found no significant correlation between c-NHEJ engagement and dCas9-based transcription silencing. New **Fig 5f** demonstrated that transcription activities did not change the state of c-NHEJ engagement at the sites where c-NHEJ is little engaged. In the previous version of the manuscript, we did not describe our question and rationale well and caused some confusions. We have made clarification in the revised manuscript accordingly.*

One inconsistency between our work and the Clarke study is that we did not find increased mutagenesis by Cas9 with the sgRNAs that anneal to the template strand of EGFP transcription. We do not know the exact reason but it may be due to different sites, a smaller number of sites and different cell types we used. Also, considering

different target binding affinities of Cas9-sgRNA at different target sites, it is still possible that target dissociation of Cas9-sgRNA by transcription is affected not only by the sgRNAs that anneal to either the template or non-template strand of transcription but also by Cas9-sgRNA target binding affinities. That is a possible reason behind the variation in dCas9-based gene silencing at different sites even with the sgRNAs annealing to the same template or non-template strand in both our study and the Clarke study (Clarke et al., 2018). In addition, we agree with the reviewer, the stronger the transcription blockage, the better for the assay. However, it remains a challenge to achieve a strong and satisfactory dCas9-based transcription blockage in mammalian cells (Dominguez et al., 2016). The blockage is about 50% in our assays using either dSaCas9-gSaGw1 or dSaCas9-gSaGw2 to block the GFP transcription (Fig 5e,f). This transcription blockage was not as effective as in prokaryotic cells, but as strong as those previously reported for GFP gene silencing in mammalian cells (Qi et al., 2013).

Minor considerations for the authors:

1. What is the frame of the I-SceI GFP reporter? As noted in this manuscript and elsewhere, DNA repair products of CRISPR are not random and therefore 1/3 of products wouldn't necessarily lead to GFP restoration. Also editing was several fold higher in CRISPR editing than I-SceI which may affect interpretation here.

We thank the reviewer for pointing out this issue. In the NHEJ reporter, there is a 34-bp interval between “Koz-ATG” and ATG for EGFP. In repair of Cas9-induced DSBs at a site between “Koz-ATG” and the ATG-GFP coding region, both c-NHEJ and a-EJ can generate indels at the repair junction. In general, net addition of “3n+2” bp or net loss of “3n-1” bp in indels can reframe the 34-bp frame-shift to in-frame, leading to production of GFP⁺ cells. Given uniqueness in repair of Cas9-induced DSBs, previous statement in the manuscript “in theory, a third of the indels can lead to GFP⁺ cells” is not accurate. We made a change in the revised manuscript accordingly. We additionally used targeted PCR amplicon deep sequencing to analyze NHEJ repair junctions of the reporter and endogenous sites. The data support the findings from FACS-based reporter assays (Fig S2b-e and Fig S3a-c).

2. From a presentation standpoint, having Figure 1E and 2F together would make sense.

We agree with the reviewer that Fig 1e and 2f (new Fig 2d) are connected more closely. But the low dosage of Cas9-sgRNA (0.001ug) was chosen after the dose-dependent assay of Fig 2b-e (new Fig 2b,c) were completed. It would be abrupt to have Fig 1e and 2f (new Fig 2d) together in the beginning.

3. Figure 3A, there is a lot of data crossing the break making it difficult to see. Consider log plot?

We redrew Fig 3a and it should be much easier to read now. Thanks for the suggestion.

References

Aldag, P., Welzel, F., Jakob, L., Schmidbauer, A., Rutkauskas, M., Fettes, F., Grohmann, D., and Seidel, R. (2021). Probing the stability of the SpCas9-DNA complex after cleavage. *Nucleic Acids Res* *49*, 12411–12421. <https://doi.org/10.1093/nar/gkab1072>.

Allen, C., Kurimasa, A., Brenneman, M.A., Chen, D.J., and Nickoloff, J.A. (2002). DNA-dependent protein kinase suppresses double-strand break-induced and spontaneous homologous recombination. *Proc Natl Acad Sci U S A* *99*, 3758–3763. <https://doi.org/10.1073/pnas.052545899>.

Allen, F., Crepaldi, L., Alsinet, C., Strong, A.J., Kleshchevnikov, V., De Angeli, P., Páleníková, P., Khodak, A., Kiselev, V., Kosicki, M., et al. (2019). Predicting the mutations generated by repair of Cas9-induced double-strand breaks. *Nat. Biotechnol.* *37*, 64–72. <https://doi.org/10.1038/nbt.4317>.

Bétermier, M., Bertrand, P., and Lopez, B.S. (2014). Is non-homologous end-joining really an inherently error-prone process? *PLoS Genet.* *10*, e1004086. <https://doi.org/10.1371/journal.pgen.1004086>.

Bindra, R.S., Goglia, A.G., Jasin, M., and Powell, S.N. (2013). Development of an assay to measure mutagenic non-homologous end-joining repair activity in mammalian cells. *Nucleic Acids Res* *41*, e115. <https://doi.org/10.1093/nar/gkt255>.

Brinkman, E.K., Chen, T., de Haas, M., Holland, H.A., Akhtar, W., and van Steensel, B. (2018). Kinetics and fidelity of the repair of Cas9-induced double-strand DNA breaks. *Mol Cell* *70*, 801-813.e6. <https://doi.org/10.1016/j.molcel.2018.04.016>.

Capper, R., Britt-Compton, B., Tankimanova, M., Rowson, J., Letsolo, B., Man, S., Haughton, M., and Baird, D.M. (2007). The nature of telomere fusion and a definition of the critical telomere length in human cells. *Genes Dev* *21*, 2495–2508. <https://doi.org/10.1101/gad.439107>.

Chakrabarti, A.M., Henser-Brownhill, T., Monserrat, J., Poetsch, A.R., Luscombe, N.M., and Scaffidi, P. (2019). Target-specific precision of CRISPR-mediated genome editing. *Mol. Cell* *73*, 699-713.e6. <https://doi.org/10.1016/j.molcel.2018.11.031>.

Chang, H.H.Y., Pannunzio, N.R., Adachi, N., and Lieber, M.R. (2017). Non-homologous DNA end joining and alternative pathways to double-strand break repair. *Nat Rev Mol Cell Biol* *18*, 495–506. <https://doi.org/10.1038/nrm.2017.48>.

Clarke, R., Heler, R., MacDougall, M.S., Yeo, N.C., Chavez, A., Regan, M., Hanakahi, L., Church, G.M., Marraffini, L.A., and Merrill, B.J. (2018). Enhanced bacterial immunity and

mammalian genome editing via RNA-polymerase-mediated dislodging of Cas9 from double-strand DNA breaks. *Mol. Cell* 71, 42–55.e8. <https://doi.org/10.1016/j.molcel.2018.06.005>.

Danner, E., Bashir, S., Yumlu, S., Wurst, W., Wefers, B., and Kühn, R. (2017). Control of gene editing by manipulation of DNA repair mechanisms. *Mamm Genome* 28, 262–274. <https://doi.org/10.1007/s00335-017-9688-5>.

Delacôte, F., Han, M., Stamato, T.D., Jasin, M., and Lopez, B.S. (2002). An *xrcc4* defect or Wortmannin stimulates homologous recombination specifically induced by double-strand breaks in mammalian cells. *Nucleic Acids Res* 30, 3454–3463. <https://doi.org/10.1093/nar/gkf452>.

Deriano, L., and Roth, D.B. (2013). Modernizing the nonhomologous end-joining repertoire: alternative and classical NHEJ share the stage. *Annu Rev Genet* 47, 433–455. <https://doi.org/10.1146/annurev-genet-110711-155540>.

Dominguez, A.A., Lim, W.A., and Qi, L.S. (2016). Beyond editing: repurposing CRISPR-Cas9 for precision genome regulation and interrogation. *Nat. Rev. Mol. Cell Biol.* 17, 5–15. <https://doi.org/10.1038/nrm.2015.2>.

Feng, Y., Liu, S., Chen, R., and Xie, A. (2021). Target binding and residence: a new determinant of DNA double-strand break repair pathway choice in CRISPR/Cas9 genome editing. *J Zhejiang Univ Sci B* 22, 73–86. <https://doi.org/10.1631/jzus.B2000282>.

Feng, Y.-L., Xiang, J.-F., Liu, S.-C., Guo, T., Yan, G.-F., Feng, Y., Kong, N., Li, H.-D., Huang, Y., Lin, H., et al. (2017). H2AX facilitates classical non-homologous end joining at the expense of limited nucleotide loss at repair junctions. *Nucleic Acids Res.* 45, 10614–10633. <https://doi.org/10.1093/nar/gkx715>.

Guo, T., Feng, Y.-L., Xiao, J.-J., Liu, Q., Sun, X.-N., Xiang, J.-F., Kong, N., Liu, S.-C., Chen, G.-Q., Wang, Y., et al. (2018). Harnessing accurate non-homologous end joining for efficient precise deletion in CRISPR/Cas9-mediated genome editing. *Genome Biol.* 19, 170. <https://doi.org/10.1186/s13059-018-1518-x>.

Hommelsheim, C.M., Frantzeskakis, L., Huang, M., and Ülker, B. (2014). PCR amplification of repetitive DNA: a limitation to genome editing technologies and many other applications. *Sci Rep* 4, 5052. <https://doi.org/10.1038/srep05052>.

Jiang, F., and Doudna, J.A. (2017). CRISPR–Cas9 structures and mechanisms. *Annu Rev Biophys* 46, 505–529. <https://doi.org/10.1146/annurev-biophys-062215-010822>.

Kagaya, K., Noma-Takayasu, N., Yamamoto, I., Tashiro, S., Ishikawa, F., and Hayashi, M.T. (2020). Chromosome instability induced by a single defined sister chromatid fusion. *Life Sci Alliance* 3, e202000911. <https://doi.org/10.26508/lsa.202000911>.

Kim, D., Luk, K., Wolfe, S.A., and Kim, J.-S. (2019). Evaluating and enhancing target

specificity of gene-editing nucleases and deaminases. *Annu. Rev. Biochem.* 88, 191–220. <https://doi.org/10.1146/annurev-biochem-013118-111730>.

Leibowitz, M.L., Papathanasiou, S., Doerfler, P.A., Blaine, L.J., Sun, L., Yao, Y., Zhang, C.-Z., Weiss, M.J., and Pellman, D. (2021). Chromothripsis as an on-target consequence of CRISPR-Cas9 genome editing. *Nat Genet* 53, 895–905. <https://doi.org/10.1038/s41588-021-00838-7>.

Lemos, B.R., Kaplan, A.C., Bae, J.E., Ferrazzoli, A.E., Kuo, J., Anand, R.P., Waterman, D.P., and Haber, J.E. (2018). CRISPR/Cas9 cleavages in budding yeast reveal templated insertions and strand-specific insertion/deletion profiles. *Proc. Natl. Acad. Sci. U.S.A.* 115, E2040–E2047. <https://doi.org/10.1073/pnas.1716855115>.

Liddiard, K., Ruis, B., Takasugi, T., Harvey, A., Ashelford, K.E., Hendrickson, E.A., and Baird, D.M. (2016). Sister chromatid telomere fusions, but not NHEJ-mediated inter-chromosomal telomere fusions, occur independently of DNA ligases 3 and 4. *Genome Res* 26, 588–600. <https://doi.org/10.1101/gr.200840.115>.

Lo, A.W.I., Sprung, C.N., Fouladi, B., Pedram, M., Sabatier, L., Ricoul, M., Reynolds, G.E., and Murnane, J.P. (2002). Chromosome instability as a result of double-strand breaks near telomeres in mouse embryonic stem cells. *Mol Cell Biol* 22, 4836–4850. <https://doi.org/10.1128/MCB.22.13.4836-4850.2002>.

Ma, H., Tu, L.-C., Naseri, A., Huisman, M., Zhang, S., Grunwald, D., and Pederson, T. (2016). CRISPR-Cas9 nuclear dynamics and target recognition in living cells. *J. Cell Biol.* 214, 529–537. <https://doi.org/10.1083/jcb.201604115>.

Menolfi, D., and Zha, S. (2020). ATM, ATR and DNA-PKcs kinases—the lessons from the mouse models: inhibition \neq deletion. *Cell Biosci* 10, 8. <https://doi.org/10.1186/s13578-020-0376-x>.

Murakami, Y., Eki, T., Yamada, M., Prives, C., and Hurwitz, J. (1986). Species-specific in vitro synthesis of DNA containing the polyoma virus origin of replication. *Proc Natl Acad Sci U S A* 83, 6347–6351. <https://doi.org/10.1073/pnas.83.17.6347>.

van Overbeek, M., Capurso, D., Carter, M.M., Thompson, M.S., Frias, E., Russ, C., Reece-Hoyes, J.S., Nye, C., Gradia, S., Vidal, B., et al. (2016). DNA repair profiling reveals nonrandom outcomes at Cas9-mediated breaks. *Mol. Cell* 63, 633–646. <https://doi.org/10.1016/j.molcel.2016.06.037>.

Pierce, A.J., Hu, P., Han, M., Ellis, N., and Jasin, M. (2001). Ku DNA end-binding protein modulates homologous repair of double-strand breaks in mammalian cells. *Genes Dev.* 15, 3237–3242. <https://doi.org/10.1101/gad.946401>.

Qi, L.S., Larson, M.H., Gilbert, L.A., Doudna, J.A., Weissman, J.S., Arkin, A.P., and Lim, W.A. (2013). Repurposing CRISPR as an RNA-guided platform for sequence-specific control of gene expression. *Cell* 152, 1173–1183. <https://doi.org/10.1016/j.cell.2013.02.022>.

- Rai, R., Gu, P., Broton, C., Kumar-Sinha, C., Chen, Y., and Chang, S. (2019). The replisome mediates a-NHEJ repair of telomeres lacking POT1-TPP1 independently of MRN function. *Cell Rep* 29, 3708-3725.e5. <https://doi.org/10.1016/j.celrep.2019.11.012>.
- Ramsden, D.A., Carvajal-Garcia, J., and Gupta, G.P. (2022). Mechanism, cellular functions and cancer roles of polymerase-theta-mediated DNA end joining. *Nat Rev Mol Cell Biol* 23, 125–140. <https://doi.org/10.1038/s41580-021-00405-2>.
- Richardson, C.D., Ray, G.J., DeWitt, M.A., Curie, G.L., and Corn, J.E. (2016). Enhancing homology-directed genome editing by catalytically active and inactive CRISPR-Cas9 using asymmetric donor DNA. *Nat. Biotechnol.* 34, 339–344. <https://doi.org/10.1038/nbt.3481>.
- Rouet, P., Smih, F., and Jasin, M. (1994). Introduction of double-strand breaks into the genome of mouse cells by expression of a rare-cutting endonuclease. *Mol. Cell. Biol.* 14, 8096–8106. .
- Seol, J.-H., Shim, E.Y., and Lee, S.E. (2018). Microhomology-mediated end joining: Good, bad and ugly. *Mutat. Res.* 809, 81–87. <https://doi.org/10.1016/j.mrfmmm.2017.07.002>.
- Shen, M.W., Arbab, M., Hsu, J.Y., Worstell, D., Culbertson, S.J., Krabbe, O., Cassa, C.A., Liu, D.R., Gifford, D.K., and Sherwood, R.I. (2018). Predictable and precise template-free CRISPR editing of pathogenic variants. *Nature* 563, 646–651. <https://doi.org/10.1038/s41586-018-0686-x>.
- Shou, J., Li, J., Liu, Y., and Wu, Q. (2018). Precise and predictable CRISPR chromosomal rearrangements reveal principles of Cas9-mediated nucleotide insertion. *Mol. Cell* 71, 498-509.e4. <https://doi.org/10.1016/j.molcel.2018.06.021>.
- Singh, D., Mallon, J., Poddar, A., Wang, Y., Tippana, R., Yang, O., Bailey, S., and Ha, T. (2018). Real-time observation of DNA target interrogation and product release by the RNA-guided endonuclease CRISPR Cpf1 (Cas12a). *Proc. Natl. Acad. Sci. U.S.A.* 115, 5444–5449. <https://doi.org/10.1073/pnas.1718686115>.
- Song, B., Yang, S., Hwang, G.-H., Yu, J., and Bae, S. (2021). Analysis of NHEJ-based DNA repair after CRISPR-mediated DNA cleavage. *Int J Mol Sci* 22, 6397. <https://doi.org/10.3390/ijms22126397>.
- Sternberg, S.H., Redding, S., Jinek, M., Greene, E.C., and Doudna, J.A. (2014). DNA interrogation by the CRISPR RNA-guided endonuclease Cas9. *Nature* 507, 62–67. <https://doi.org/10.1038/nature13011>.
- Strohkendl, I., Saifuddin, F.A., Rybarski, J.R., Finkelstein, I.J., and Russell, R. (2018). Kinetic basis for DNA target specificity of CRISPR-Cas12a. *Mol. Cell* 71, 816-824.e3. <https://doi.org/10.1016/j.molcel.2018.06.043>.
- Swarts, D.C., and Jinek, M. (2019). Mechanistic insights into the cis- and trans-acting DNase activities of Cas12a. *Mol Cell* 73, 589-600.e4. <https://doi.org/10.1016/j.molcel.2018.11.021>.

Swarts, D.C., van der Oost, J., and Jinek, M. (2017). Structural basis for guide RNA processing and seed-dependent DNA targeting by CRISPR-Cas12a. *Mol. Cell* 66, 221-233.e4. <https://doi.org/10.1016/j.molcel.2017.03.016>.

Umbreit, N.T., Zhang, C.-Z., Lynch, L.D., Blaine, L.J., Cheng, A.M., Tourdot, R., Sun, L., Almubarak, H.F., Judge, K., Mitchell, T.J., et al. (2020). Mechanisms generating cancer genome complexity from a single cell division error. *Science* 368, eaba0712. <https://doi.org/10.1126/science.aba0712>.

Viguera, E., Canceill, D., and Ehrlich, S.D. (2001). In vitro replication slippage by DNA polymerases from thermophilic organisms. *J Mol Biol* 312, 323–333. <https://doi.org/10.1006/jmbi.2001.4943>.

Wang, A.S., Chen, L.C., Wu, R.A., Hao, Y., McSwiggen, D.T., Heckert, A.B., Richardson, C.D., Gowen, B.G., Kazane, K.R., Vu, J.T., et al. (2020). The histone chaperone FACT induces Cas9 multi-turnover behavior and modifies genome manipulation in human cells. *Mol Cell* 79, 221-233.e5. <https://doi.org/10.1016/j.molcel.2020.06.014>.

White, M.K., Hu, W., and Khalili, K. (2016). Gene editing approaches against viral infections and strategy to prevent occurrence of viral escape. *PLoS Pathog* 12, e1005953. <https://doi.org/10.1371/journal.ppat.1005953>.

Xie, A., Puget, N., Shim, I., Odate, S., Jarzyna, I., Bassing, C.H., Alt, F.W., and Scully, R. (2004). Control of sister chromatid recombination by histone H2AX. *Mol. Cell* 16, 1017–1025. <https://doi.org/10.1016/j.molcel.2004.12.007>.

Xie, A., Kwok, A., and Scully, R. (2009). Role of mammalian Mre11 in classical and alternative nonhomologous end joining. *Nat. Struct. Mol. Biol.* 16, 814–818. <https://doi.org/10.1038/nsmb.1640>.

Xue, C., and Greene, E.C. (2021). DNA repair pathway choices in CRISPR-Cas9-mediated genome editing. *Trends Genet* 37, 639–656. <https://doi.org/10.1016/j.tig.2021.02.008>.

Yan, C.T., Kaushal, D., Murphy, M., Zhang, Y., Datta, A., Chen, C., Monroe, B., Mostoslavsky, G., Coakley, K., Gao, Y., et al. (2006). XRCC4 suppresses medulloblastomas with recurrent translocations in p53-deficient mice. *Proc Natl Acad Sci U S A* 103, 7378–7383. <https://doi.org/10.1073/pnas.0601938103>.

Yin, J., Lu, R., Xin, C., Wang, Y., Ling, X., Li, D., Zhang, W., Liu, M., Xie, W., Kong, L., et al. (2022). Cas9 exo-endonuclease eliminates chromosomal translocations during genome editing. *Nat Commun* 13, 1204. <https://doi.org/10.1038/s41467-022-28900-w>.

Zha, S., Sekiguchi, J., Brush, J.W., Bassing, C.H., and Alt, F.W. (2008). Complementary functions of ATM and H2AX in development and suppression of genomic instability. *Proc Natl Acad Sci U S A* 105, 9302–9306. <https://doi.org/10.1073/pnas.0803520105>.

Zhang, Q., Wen, F., Zhang, S., Jin, J., Bi, L., Lu, Y., Li, M., Xi, X.-G., Huang, X., Shen, B., et

al. (2019). The post-PAM interaction of RNA-guided spCas9 with DNA dictates its target binding and dissociation. *Sci Adv* 5, eaaw9807. <https://doi.org/10.1126/sciadv.aaw9807>.

Zhang, S., Zhang, Q., Hou, X.-M., Guo, L., Wang, F., Bi, L., Zhang, X., Li, H.-H., Wen, F., Xi, X.-G., et al. (2020). Dynamics of *Staphylococcus aureus* Cas9 in DNA target association and dissociation. *EMBO Rep* 21, e50184. <https://doi.org/10.15252/embr.202050184>.

Zhu, X., Clarke, R., Puppala, A.K., Chittori, S., Merk, A., Merrill, B.J., Simonović, M., and Subramaniam, S. (2019). Cryo-EM structures reveal coordinated domain motions that govern DNA cleavage by Cas9. *Nat. Struct. Mol. Biol.* 26, 679–685. <https://doi.org/10.1038/s41594-019-0258-2>.

Second round of review

Reviewer 1

The authors have generated new data to address my concerns. I recommend the publication of this manuscript in Genome Biology.

Reviewer 3

The authors have addressed my concerns and I have no further comments. I defer to the expert reviewers on DNA repair for the remaining concerns.